# Plausible Token Amplification for Improving Accuracy of Differentially Private In-Context Learning Based on Implicit Bayesian Inference

**Yusuke Yamasaki** [1]   **Kenta Niwa** [2]   **Daiki Chijiwa** [3]   **Takumi Fukami** [1]   **Takayuki Miura** [1]

## Abstract

We propose Plausible Token Amplification (PTA)[1] to improve the accuracy of Differentially Private In-Context Learning (DP-ICL) using DP synthetic demonstrations. While Tang et al. empirically improved the accuracy of DP-ICL by limiting vocabulary space during DP synthetic demonstration generation, its theoretical basis remains unexplored. By interpreting ICL as implicit Bayesian inference on a concept underlying demonstrations, we not only provide theoretical evidence supporting Tang et al.'s empirical method but also introduce PTA, a refined method for modifying next-token probability distribution. Through the modification, PTA highlights tokens that distinctly represent the ground-truth concept underlying the original demonstrations. As a result, generated DP synthetic demonstrations guide the Large Language Model to successfully infer the ground-truth concept, which improves the accuracy of DP-ICL. Experimental evaluations on both synthetic and real-world text-classification datasets validated the effectiveness of PTA.

## 1. Introduction

Large Language Models (LLMs) exhibit an impressive capability known as In-Context Learning (ICL) (Brown et al., 2020), where a few pairs of data and their labels (demonstrations) and new data (query) are provided together as a prompt. These demonstrations help LLM infer a concept—namely, a latent rule that connects data to their labels—which governs the token transitions in demonstrations. ICL aligns the LLM's prediction for the query with the inferred concept accordingly, enabling accurate responses. Despite ICL's success across various tasks (Mathur et al., 2023; Hegselmann et al., 2023), embedding sensitive information in demonstrations, such as personal health records, risks leaking sensitive information from responses (Wang et al., 2023a; Duan et al., 2023). This necessitates privacy-aware ICL methods that mitigate leakage risks while ensuring responses are aligned with the concept underlying demonstrations.

To mitigate leakage risks in ICL, Differential Privacy (DP) (Dwork et al., 2006a), which offers privacy guarantees against leakage risks, has been incorporated into ICL, often referred to as DP-ICL (Tang et al., 2024; Wu et al., 2024; Hong et al., 2024). Notably, (Tang et al., 2024) studied generating synthetic demonstrations by adding variance-tuned noise to the next-token probability obtained from an LLM prompted with the original demonstrations. The added noise ensures that the generated synthetic demonstrations satisfy DP, reducing the leakage risks of the originals by providing these synthetic ones in the prompt. While increasing noise variance lowers leakage risks, it also makes the synthetic demonstrations deviate more from the originals, thereby degrading task accuracy. To recover the degraded accuracy, (Tang et al., 2024) empirically limited the vocabulary space using public information during noise addition. While numerical experiments showed its effectiveness in improving accuracy, its theoretical basis remains unexplored.

For further credibility in DP-ICL, we begin with its theoretical analysis based on the Bayesian analysis (see Section 3) inspired by existing work on standard ICL (without employing DP) (Xie et al., 2022; Wang et al., 2023b), which interprets ICL as implicit Bayesian inference on a *ground-truth concept* underlying the original demonstrations. Extending this framework to DP-ICL, we establish Theorem 2 explaining how the added noise to ensure DP affects the LLM's ability to infer the ground-truth concept. Specifically, we show that the LLM successfully infers the ground-truth concept if the expected divergence, which depends on the next-token probability distribution, surpasses a certain threshold. This divergence measures how much the distribution under the ground-truth concept differs from that under any other

[1]NTT Social Informatics Laboratories [2]NTT Communication Science Laboratories [3]NTT Computer and Data Science Laboratories. Correspondence to: Yusuke Yamasaki <yusuke.yamasaki@ntt.com>.

*Proceedings of the 42nd International Conference on Machine Learning*, Vancouver, Canada. PMLR 267, 2025. Copyright 2025 by the author(s).

[1]The code will be available at `https://github.com/Yusuke-Yamasaki/pta`

concept, while the threshold reflects the negative impact of noise, primarily influenced by the noise variance and the vocabulary size. Hence, the added noise hinders the accurate inference of the ground-truth concept in DP-ICL, explaining why DP-ICL often suffers from accuracy degradation.

To address this accuracy degradation, we derive two insights from our theory: (i) Reducing the vocabulary size lowers the noise-dependent threshold, providing theoretical support for Tang et al.'s empirical method of limiting vocabulary space. (ii) Increasing the divergence between concepts by employing another next-token probability distribution that enlarges the gap between the ground-truth and any other concept. The second insight highlights a promising approach for designing the next-token probability distribution to increase the divergence, while previous work only focused on (i).

Motivated by (ii), we propose Plausible Token Amplification (PTA) to increase the divergence between the ground-truth and any other concept by amplifying tokens that distinctly represent the ground-truth concept (see Section 4). Formally, PTA solves an optimization that maximizes this divergence while remaining close to the original next-token probability distribution, ensuring the contextual coherence of the resulting synthetic demonstrations. As a result, PTA preserves alignment with the ground-truth concept, effectively recovering the degraded accuracy due to the added noise in DP-ICL. Our key contributions are summarized as follows:

**Bayesian analysis of DP-ICL (Section 3):** We introduce Bayesian analysis into DP-ICL and derive Theorem 2, revealing two key insights: (i) Reducing the vocabulary size mitigates the negative impact of noise on the ground-truth concept inference, theoretically supporting Tang et al.'s empirical method. (ii) Increasing the divergence between the ground-truth and any other concept helps guide the LLM toward accurate inference on the ground-truth concept. Since Tang et al.'s method does not fully exploit this second insight, our analysis implies that addressing this oversight can improve the accuracy of DP-ICL.

**PTA for accurate DP-ICL (Section 4):** Motivated by our Theorem 2, we propose PTA to increase the divergence between the ground-truth and any other concept by highlighting the distinctive tokens in the ground-truth concept. To achieve this, PTA compares next-token probability distribution conditioned on private demonstrations and public information, ensuring no additional leakage risks beyond Tang et al.'s method. Hence, PTA aligns generated DP synthetic demonstrations with the ground-truth concept while maintaining contextual coherence and improving the accuracy of DP-ICL.

**Experimental validations (Section 5):** We validated the effectiveness of PTA through experiments on synthetic and real-world text-classification tasks across various settings, demonstrating improved accuracy DP-ICL benchmark tests.

## 2. Preliminaries

We first review existing DP-ICL methods in Section 2.1. Section 2.2 presents a theoretical analysis of standard ICL (without employing DP).

### 2.1. DP-ICL

$(\varepsilon, \delta)$-Differential Privacy (DP) provides statistical privacy guarantees for datasets used in queries (formal definition is given in Appendix D). Here, $\varepsilon \, (> 0)$ measures the maximum allowable bound in outputs between neighboring datasets, while $\delta \in (0, 1)$ is the maximum failure probability of exceeding the bound $\varepsilon$. Smaller values of $(\varepsilon, \delta)$ imply lower leakage risks of the used dataset. The Gaussian mechanism, widely used to achieve $(\varepsilon, \delta)$-DP, adds the independent noise that follows the Gaussian distribution $\mathcal{N}(0, \sigma^2)$ to the output of the query. The Gaussian mechanism satisfies $(\varepsilon, \delta)$-DP (Dwork & Roth, 2014) when $\sigma^2$ is chosen appropriately on the basis of $(\varepsilon, \delta)$, and the $\ell_2$-sensitivity — defined as the maximum change in the query outputs between neighboring datasets measured by $\|\cdot\|_2$.

Integrating mechanisms to ensure DP into standard ICL (DP-ICL) can be used to mitigate leakage risks from query responses. With smaller $(\varepsilon, \delta)$ values, the LLM's responses are less likely to leak any individual demonstrations provided in the prompt. To achieve this, generating DP synthetic demonstrations (Tang et al., 2024) is a particularly promising approach, when handling an unpredictable number of queries, as noise variance remains stable regardless of the number of queries, due to DP's post-processing immunity (Dwork et al., 2006b). For more on related methods, see (Edemacu & Wu, 2025) and Appendix B.

Notably, (Tang et al., 2024) studied generating DP synthetic demonstrations by adding noise to a next-token probability distribution instead of model parameters, ensuring flexibility across LLMs and datasets. Noise is added to the next-token probability distribution generated by the LLM in parallel, conditioned on a prompt that contains a task instruction and a disjoint subset of the original private demonstrations where each demonstration is formatted as a label-data pair. To improve accuracy, the vocabulary space is limited before noise addition, using public information such as the task instruction. Specifically, two types of prompts are used: $S_{\text{pub}}$, which only includes the task instruction and the tokens generated thus far, and $S_{\text{priv}}^{(i)}$, which additionally contains the $i$-th disjoint subset obtained by partitioning the original demonstrations $\mathcal{D}_{\text{priv}}$ into $M$ disjoint subsets. $S_{\text{pub}}$ helps in limiting away irrelevant vocabularies, while $S_{\text{priv}}$ provides task-specific information. The token generation process satisfying DP is outlined as follows:

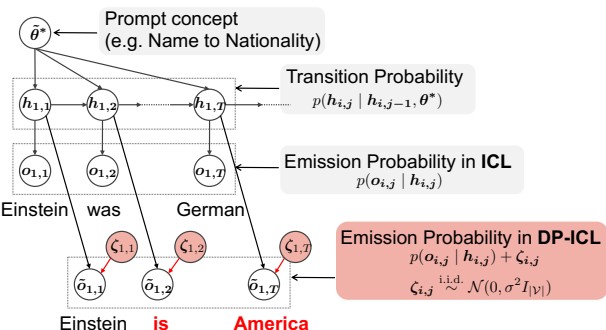

Figure 1: Illustration of HMM-based model for ICL introduced in (Xie et al., 2022) and the proposed one for DP-ICL introduced in this paper. The red arrows highlight the new dependencies, showing how the generated tokens in DP-ICL are influenced by noise added to ensure DP.

$$\mathcal{V}_{\text{pub}} = \arg\max_{\mathcal{V}' \subset \mathcal{V}} \sum_{v \in \mathcal{V}'} p(o^{\text{LLM}} = v | S_{\text{pub}}), \text{ s.t. } |\mathcal{V}_{\text{pub}}| = k, \quad (1)$$

$$p_v \leftarrow \frac{1}{M} \sum_{i=1}^{M} p(o^{\text{LLM}} = v | S_{\text{priv}}^{(i)}), \ \hat{p}_v \leftarrow p_v \Big/ \big( \sum_{v' \in \mathcal{V}_{\text{pub}}} p_{v'} \big), \quad (2)$$

$$v_{\text{syn}} = \arg\max_{v \in \mathcal{V}_{\text{pub}}} \{\hat{p}_v + \zeta_v\}, \quad \zeta_v \sim \mathcal{N}(0, 2\sigma^2), \quad (3)$$

where $\mathcal{V}$ is the LLM's token vocabulary, $o^{\text{LLM}}$ is a random variable such that $p(o^{\text{LLM}} = v)$ represents the next-token probability of the LLM, and $k \leq |\mathcal{V}|$ is the number of the vocabulary selected. (1) limits the vocabulary space $\mathcal{V}$ to a subset $\mathcal{V}_{\text{pub}}$ on the basis of public information $S_{\text{pub}}$, filtering out low-likelihood tokens. In (2), token probabilities conditioned on private prompts $S_{\text{priv}}^{(1)}, \ldots, S_{\text{priv}}^{(M)}$ are averaged and normalized within $\mathcal{V}_{\text{pub}}$. Finally, (3) adds noise to these probability distributions to ensure $(\varepsilon, \delta)$-DP, then yielding the synthetic token $v_{\text{syn}}$. The generated tokens are appended to both prompts $S_{\text{pub}}$ and $S_{\text{priv}}$, recursively constructing a DP synthetic demonstration. Empirical results showed that limiting the vocabulary space improved the stability and the accuracy of DP-ICL.

Tang et al.'s DP-ICL faces two main challenges. First, their method of limiting the vocabulary space has only been empirically evaluated, lacking theoretical evidence on its ability to improve the accuracy while satisfying DP. Secondly, limiting vocabulary space to $\mathcal{V}_{\text{pub}}$ may lead to degrading accuracy degrading particularly when $p(o^{\text{LLM}} | S_{\text{pub}})$ deviates from $p(o^{\text{LLM}} | S_{\text{priv}}^{(i)})$, as $\mathcal{V}_{\text{pub}}$ may include the low-likelihood tokens conditioned on $S_{\text{priv}}^{(i)}$ and the generated demonstration including them due to the added noise.

## 2.2. Bayesian analysis of ICL

Limited to standard ICL (without employing DP), its accuracy has been theoretically investigated on the basis of

Bayesian analysis (Xie et al., 2022; Wang et al., 2023b; Ling et al., 2024; Reizinger et al., 2024). From a Bayesian perspective, ICL infers concepts underlying the demonstrations provided in a prompt, guiding the LLM to generate tokens aligned with the inferred concept. To clarify the mechanism of this implicit Bayesian inference, we briefly review the problem setting and key results from (Xie et al., 2022), which offers fundamental theoretical insights into the implicit Bayesian inference in ICL.

**Problem setting in (Xie et al., 2022)** Token generation in LLMs is modeled using a Hidden Markov Model (HMM), as depicted in Figure 1. Figure 1 schematically illustrates how an HMM governs the token transitions where each token is emitted on the basis of a corresponding hidden state. In this model, each demonstration token $o_{i,j}$ is emitted from a hidden state $h_{i,j}$ in accordance with the emission probability $p(o_{i,j}|h_{i,j})$, which is independent of the shared concept $\theta^*$. In contrast, the transition probability $p(h_{i,j}|h_{i,j-1}, \theta^*)$ depends on $\theta^*$ (e.g., linking a name to its nationality).

On the basis of this framework, (Xie et al., 2022) construct a prompt by concatenating $n$ independent demonstrations that follow HMM where each is separated by a delimiter token and then appending a query $x_{\text{query}}$. Formally, we represent the prompt distribution as follows:

$$[S_n, x_{\text{query}}] = [O_1, o_1^{\text{delim}}, \ldots, O_n, o_n^{\text{delim}}, x_{\text{query}}] \sim p_{\text{prompt}},$$

where $S_n$ denotes the concatenation of the $n$ demonstrations $O_1, \ldots, O_n$ and $o_i^{\text{delim}}$ is a delimiter token. Each demonstration is independently generated in accordance with the shared concept $\theta^*$, ensuring the transition probability $p(h_{i,j}|h_{i,j-1}, \theta^*)$ remains consistent across all demonstrations while the $j$-th token of the $i$-th demonstration $o_{i,j}$ is sampled from $\mathcal{V}$ on the basis of its emission probability $p(o_{i,j}|h_{i,j})$. The formal prompt formulation using the HMM, for a structured framework to analyze token dependencies in ICL, is provided in Appendix C.

**Interpretation of ICL as implicit Bayesian inference.** To analyze the next-token probability of an LLM conditioned on the prompt $S_n, x_{\text{query}}$, we apply Bayes's rule and transform it as follows:

$$p(y|S_n, x_{\text{query}}) = \int_\theta p(y|S_n, x_{\text{query}}, \theta) p(\theta|S_n, x_{\text{query}}) d\theta$$
$$\propto \int_\theta p(y|S_n, x_{\text{query}}, \theta) \frac{p(S_n, x_{\text{query}}|\theta)}{p(S_n, x_{\text{query}}|\theta^*)} p(\theta) d\theta.$$

If the posterior $p(\theta|S_n, x_{\text{query}})$ concentrates on the ground-truth concept $\theta^*$ underlying the demonstrations, the LLM will generate tokens that align closely with $\theta^*$. To evaluate how closely $p(\theta|S_n, x_{\text{query}})$ concentrates on $\theta^*$, the following quantity is introduced:

$$r_n(\theta) = \frac{1}{n} \log \frac{p(S_n, x_{\text{query}}|\theta)}{p(S_n, x_{\text{query}}|\theta^*)}, \quad (4)$$

where $r_n(\theta)$ represents the average log-likelihood ratio of the prompt, comparing the likelihood conditioned on any other concept $\theta \in \Theta$ and the ground-truth concept $\theta^*$, letting $\Theta$ be a set of concepts. By using $r_n(\theta)$, the likelihood ratio $p(S_n, x_{\text{query}}|\theta)/p(S_n, x_{\text{query}}|\theta^*)$ can be expressed as $\exp(n \cdot r_n(\theta))$. If $r_n(\theta)$ converges to a negative constant for $\theta \neq \theta^*$, then $\exp(n \cdot r_n(\theta)) \to 0$ as $n$ increases, while $\exp(n \cdot r_n(\theta)) = 1$ when $\theta = \theta^*$. This convergence of $r_n(\theta)$ indicates that the posterior probability $p(\theta|S_n, x_{\text{query}})$ will increasingly favor the ground-truth concept $\theta^*$ over any other concept, as more demonstrations are provided in the prompt. In (Xie et al., 2022), a the condition for the convergence of $r_n(\theta)$ is theoretically analyzed under certain assumptions on HMM used in the prompt formulation. Due to space limitations, we provide a formal description of Assumptions 1-4 used in (Xie et al., 2022) in Appendix C.

Under these assumptions, (Xie et al., 2022) provide a key condition ensuring that the posterior distribution concentrates on $\theta^*$ as the number of demonstrations grows, i.e. $\lim_{n\to\infty} \exp(n \cdot r_n(\theta)) = 0$. Below, we restate their Theorem 1 in a transformed form for consistent expression in this paper; the full original statement and the derivation of transformation appear in Appendix C.

**Theorem 1** (Distinguishability of the prompt concept (Xie et al., 2022)). *Suppose Assumptions 1,2,3, and 4 in Appendix C hold. Then, we have $\lim_{n\to\infty} \exp(n \cdot r_n(\theta)) = 0$, if $\theta^*$ is the ground-truth concept of the prompt and satisfies the following condition for any other concept $\theta \in \Theta$:*

$$\mathbb{E}_{O \sim p_{prompt}} \left[ \log \frac{p(O|\theta^*)}{p(O|\theta)} \right] > C^{delim}, \qquad (5)$$

*where $p_{prompt}$ and $p$ represent the prompt distribution and pre-training distribution, respectively. The expectation is taken over $O \sim p_{prompt}$. $C^{delim}$ is a bounded positive value that depends on $\theta$ and $\theta^*$.*

Left-Hand Side (LHS) in (5) measures how much the distribution under $\theta^*$ deviates from that under any other concept $\theta$, while $C^{\text{delim}}$ accounts for the extra complexity introduced by delimiter tokens in the prompt. According to Theorem 1, if this divergence exceeds $C^{\text{delim}}$, the LLM accurately infers the ground-truth concept $\theta^*$, enabling Bayes-optimal predictions for queries (Xie et al., 2022). Consequently, designing demonstrations that yield a substantial divergence between $p(O|\theta^*)$ and $p(O|\theta)$ surpassing $C^{\text{delim}}$ is an effective way to improve the accuracy of ICL.

In the next section, we extend this theoretical foundation to investigate DP-ICL, incorporating modifications to account for the noise added to ensure DP.

## 3. Bayesian analysis of DP-ICL

In this section, we theoretically explore DP-ICL (Tang et al., 2024) on the basis of implicit Bayesian inference. Following the methodology in (Xie et al., 2022) outlined in Section 2.2, our analysis focuses on how adding noise negatively impacts the LLM's ability to infer the ground-truth concept $\theta^*$ underlying the demonstrations. First, we introduce (i) the modeling of the DP synthetic prompt distribution, particularly the noise addition mechanism in (3), while setting aside factors like vocabulary space limitation (1) and normalization (2). This simplification allows us more clearly analyze how adding noise impacts implicit Bayesian inference on concepts. We then proceed to assess (ii) the effects of vocabulary space limitation and normalization.

**(i) Modeling DP synthetic prompt distribution.** To analyze the negative impact of noise addition, we extend the HMM-based model presented in Appendix C to model the DP synthetic prompt distribution. We incorporate noise addition into the model:

$$\tilde{o}_{i,j} \sim p(\tilde{o}_{i,j}|h_{i,j}) := p(o_{i,j}|h_{i,j}) + \zeta_{i,j}, \qquad (6)$$

where $\zeta_{i,j} \sim \mathcal{N}(0, \sigma^2 I_{|\mathcal{V}|})$ represents $|\mathcal{V}|$-dimensional independently and identically distributed (IID) Gaussian noise. Our HMM formulations involve the modification solely on the emission probability compared to those in (Xie et al., 2022), as shown in Figure 1. Note that adding IID Gaussian noise to each token's emission probability is consistent with the standard method for adding Gaussian noise to the next-token probability for generating DP synthetic demonstration as discussed in Section 2.1. While the noise perturbs the output probability of token sequences to ensure DP, the hidden states transition will remain aligned with the ground-truth concept $\theta^*$ underlying the original demonstrations, as shown in Figure 1.

Next, we examine how closely the concept inferred from the DP synthetic prompt aligns with the ground-truth concept $\theta^*$. To do this, we define the quantity, analogous to (4):

$$\tilde{r}_n(\theta) := \frac{1}{n} \log \frac{p(\tilde{S}_n, x_{\text{query}}|\theta)}{p(\tilde{S}_n, x_{\text{query}}|\theta^*)}, \qquad (7)$$

where $\tilde{S}_n$ represents the DP synthetic prompt comprising $n$ DP synthetic demonstrations. This measure assesses how the noise addition affects the inference of the ground-truth concept compared to any other concepts. Intuitively, a larger noise variance $\sigma^2$ can hinder the accurate inference of the ground-truth concept $\theta^*$ because the noise may increase the likelihood conditioned on any other concepts $\theta \neq \theta^*$ in the numerator of (7) while decreasing the likelihood conditioned on the ground-truth concept $\theta^*$. To assess (7), assumptions used in (Xie et al., 2022) are applicable. Associated with $\tilde{r}_n(\theta)$, we derive the following lemma, as a DP-ICL counterpart to Lemma 8 in (Xie et al., 2022):

**Lemma 1.** *Suppose Assumptions 1,2,3, and 4 hold. Then, we then have*

$$\tilde{r}_n(\theta) \leq -\frac{1}{n}\sum_{i=1}^{n}\log\frac{p(\tilde{O}_i|\theta^*)}{p(\tilde{O}_i|\theta)} + C^{delim} + O(n^{-1}), \quad (8)$$

*where $C^{delim}$ is a bounded positive value that depends on $\theta$ and $\theta^*$.*

The proof is provided in Appendix C, utilizing the factorization of the complex prompt $\tilde{S}_n$ into the individual demonstrations $\tilde{O}_i$ on the basis of our HMM-based prompt formulation presented in (14a)-(14d) and (6) in Appendix C. Following the proof strategy in Theorem 1, we aim to clarify the condition ensuring that $\tilde{r}_n(\theta)$ converges to a negative constant as $n$ increases, signifying accurate inference of the ground-truth concept $\theta^*$.

To derive the condition for the convergence of $\tilde{r}_n(\theta)$, we focus on the first term in (8), which represents the average log-likelihood ratio of DP synthetic demonstrations. By reformulating this term to isolate two additional terms: (i) noise error and (ii) estimation error in DP-ICL, while preserving as many terms appearing in Theorem 1 as possible, the following theorem is derived.

**Theorem 2** (Distinguishabiliy of the prompt concept on DP-ICL). *Suppose Assumptions 1,2,3, and 4 in Appendix C hold. Then, we have $\lim_{n\to\infty}\exp(n\cdot\tilde{r}_n(\theta)) = 0$ with $(1-\gamma)(1-\gamma'), \gamma, \gamma' \in (0,1)$ probability, if $\theta^*$ is the underlying concept of the prompt and satisfies the following condition for any other competing concept $\theta \in \Theta$:*

$$\mathbb{E}_{O\sim p_{prompt}}\left[\log\frac{p(O|\theta^*)}{p(O|\theta)}\right] > C^{delim} + \hat{C}$$

$$+\underbrace{\Omega\left(T^2 G(\sigma)\log|\mathcal{V}|\right)}_{Noise\ error}+\underbrace{\Omega\left(T\sqrt{\frac{1}{n}\log\frac{2}{\min(\gamma,\gamma')}}\right)}_{Estimation\ error}, \quad (9)$$

*where $T$ is the length of each demonstration, $G(\sigma) \in [0,1]$ is a continuous function of noise variance $\sigma^2$ such that $\lim_{\sigma\to 0} G(\sigma) = 0$, and depends on the LLM's next-token probability distribution. $C^{delim}$ and $\hat{C}$ are bounded positive values that depend on $\theta$ and $\theta^*$.*

The detailed proof and formal statement is provided in Appendix C, and here we offer an interpretation of Theorem 2. We consider that Theorem 2 is a natural extension of Theorem 1, since the threshold in the Right-Hand Side (RHS) introduces additional noise and estimation errors in the accurate inference of the ground-truth concept while LHS represents the divergence between $p(O|\theta^*)$ and $p(O|\theta)$ the same as in Theorem 1. Both Theorems 1 and 2 almost converge in the noise-free setting except for $\hat{C}$, as the noise and the estimation errors vanish when $\sigma \to 0$ and $n \to \infty$, respectively. To mitigate the impact of these terms, we can

adjust the vocabulary size $|\mathcal{V}|$. Reducing the vocabulary size helps minimize the noise error, since the noise-dependent term $G(\sigma)$, which quantifies the deviation between $\tilde{p}_{prompt}$ and $p_{prompt}$ when $\sigma > 0$, is primarily influenced by $|\mathcal{V}|$. This helps LLM accurately infer the ground-truth concept, leading to a more accurate DP-ICL. This insight partially supports the empirical success of limiting the vocabulary space using publicly available information used in (Tang et al., 2024), suggesting that Tang et al.'s empirical method is broadly correct. However, the derived condition still overlooks the combined effects of vocabulary space limitation (1) and normalization (2) on the divergence term in LHS. Therefore, Theorem 2 needs to be further refined to better align with the baseline DP-ICL practice in (Tang et al., 2024).

**(ii) Effects of vocabulary space limitation and normalization.** To analyze the effects of vocabulary space limitation (1) and normalization (2), we reformulate the expectation of the log-likelihood ratio of demonstrations, which quantifies the divergence between the ground-truth and any other concept, as represented in the LHS of Theorem 2. Consider a scenario where irrelevant tokens are wrongly included and emphasized through (1)–(3). Then, generated tokens would be less aligned with the ground-truth concept. This misalignment significantly decreases the divergence between the ground-truth and any other concept, thus hindering the accurate inference of the ground-truth concept. To illustrate this, we reformulate the LHS of Theorem 2 as follows:

$$\mathbb{E}_O\left[\log\frac{p(O|\theta^*)}{p(O|\theta)}\right]=\sum_{\boldsymbol{v}\in\mathcal{V}^T}\underbrace{p_{prompt}(O=\boldsymbol{v})}_{\substack{\text{Token probability}\\\text{sampled from}\\\text{all vocabulary space}}}\underbrace{\log\frac{p(O=\boldsymbol{v}|\theta^*)}{p(O=\boldsymbol{v}|\theta)}}_{\text{Token likelihood ratio}}.$$

Since vocabulary space limitation can be reflected by altering $\mathcal{V}$ to its subset $\mathcal{V}_{pub}$ determined using publicly available information, the probability distribution computed over the limited vocabulary space is given by $\hat{p}_{prompt}$, and the LHS of Theorem 2 can be expressed by

$$\sum_{\boldsymbol{v}\in\mathcal{V}_{pub}^T}\underbrace{\hat{p}_{prompt}(O=\boldsymbol{v})}_{\substack{\text{Normalized token probability}\\\text{sampled from limited vocabulary space}}}\log\frac{p(O=\boldsymbol{v}|\theta^*)}{p(O=\boldsymbol{v}|\theta)}. \quad (10)$$

From (10), we observe that the obtained probability distribution does not prioritize maximizing the LHS of Theorem 2 for a given vocabulary set $\mathcal{V}_{pub}$, as it ignores the likelihood ratio term in (10). We consider Tang et al.'s method suboptimally modifying the next-token probability distribution, leading to a decreased LHS value of Theorem 2 and failing to fully capitalize on the reduction of the noise error benefits of limiting vocabulary space.

In the following section, we propose a method to modify next-token probability aiming to maximize the LHS of Theorem 2.

# 4. Proposed method

We propose Plausible Token Amplification (PTA) to maximize the divergence between the generated demonstrations under the ground-truth concept and any other concept, as formalized in (10), by modifying next-token probability distribution. PTA aligns generated demonstrations with the ground-truth concept while maintaining their natural flow. Section 4.1 outlines the problem motivating PTA, and Section 4.2 details its derivation as the solution.

## 4.1. Problem formulation

Based on insights from Section 3, our objective is to enable an LLM to accurately infer the ground-truth concept from the generated demonstrations. To achieve this, we want to modify the next-token probability so that it maximizes the divergence between $p(O|\theta^*)$ and $p(O|\theta)$ computed over the limited vocabulary space. However, if we focus solely on maximizing this divergence, the modified next-token probability may ignore natural token transitions. This leads to abrupt changes in token transitions that degrade the likelihood of the overall generated sequence and hinders the LLM's ability to accurately infer the ground-truth concept.

To address this, we introduce a regularization that keeps the modified distribution close to a baseline probability, which reflects the natural flow of tokens already generated. Specifically, for the $j$-th token, we formulate the following optimization problem to find a probability distribution $\hat{p} := (\hat{p}_v)_{v \in \mathcal{V}_{\text{pub}}}$ over the limited vocabulary space $\mathcal{V}_{\text{pub}}$:

$$\max_{\hat{p}} \sum_{v \in \mathcal{V}_{\text{pub}}} \hat{p}_v \log \frac{p(o^{\text{LLM}}=v|\theta^*)}{p(o^{\text{LLM}}=v|\theta)} - \frac{1}{\alpha} D_{\text{KL}}\left(\hat{p}\|p_{\text{base}}\right), \quad (11)$$

where $D_{\text{KL}}$ is the Kullback-Leibler (KL) divergence measures how much $\hat{p}_v$ deviates from the baseline probability $p_{\text{base}}(o^{\text{LLM}}) := p(o^{\text{LLM}}|\tilde{O}_{1:j-1})$. The hyperparameter $\alpha$ controls how strongly we penalize divergence from this baseline probability $p_{\text{base}}$. By incorporating the KL term, we ensure that $\hat{p}_v$ remain close to $p_{\text{base}}$, preserving the natural transition of tokens. Consequently, the resulting solution from (11) highlights tokens that distinctly represent the task while preserving the natural flow of demonstration.

## 4.2. Plausible Token Amplification (PTA)

The objective function (11) requires evaluating the divergence between $p(O|\theta^*)$ and $p(O|\theta)$. Since computing this divergence depends on both the ground-truth concept $\theta^*$ and any other concept $\theta$ that are not observable, (11) is intractable to solve directly. Instead, a practical alternative would be estimating the target divergence comparing likelihood conditioned on the concepts $\theta^*$ and $\theta$ by prompting the LLM with demonstrations that are closely aligned with these concepts, as using such demonstrations will condi-

---

**Algorithm 1** PTA for DP synthetic demonstrations

**Input:** instruction, $\mathcal{D}_{\text{priv}}, \tilde{y}, T, M, N, k, p, \sigma, \alpha$.
**Subroutine:** Algorithm 2, Algorithm 3
1: **Initialize:** Set $\tilde{O} \leftarrow []$.
2: **for** $j = 1$ to $T$ **do**
3: $\quad S_{\text{pub}}, S_{\text{priv}}^{(1)}, \ldots, S_{\text{priv}}^{(M)}$
$\quad\quad \leftarrow \text{GenPrompt}(\mathcal{D}_{\text{priv}}, M, N, \tilde{y}, \tilde{O})$
4: $\quad$ **for** $i = 1$ to $M$ **do**
5: $\quad\quad p_v^{(i)} \leftarrow p(o^{\text{LLM}} = v|\tilde{O})\left(\frac{p(o^{\text{LLM}}=v|S_{\text{priv}}^{(i)})}{p(o^{\text{LLM}}=v|S_{\text{pub}})}\right)^{\alpha}$
6: $\quad\quad p_v^{(i)} \leftarrow p_v^{(i)} / \sum_{v' \in \mathcal{V}} p_{v'}^{(i)}$
7: $\quad$ **end for**
8: $\quad \hat{p}_v \leftarrow \frac{1}{M} \sum_{i=1}^{M} \{p_v^{(i)} + \mathcal{N}(0, 2\sigma^2)\}$
9: $\quad \mathcal{V}_{\text{pub}} \leftarrow \text{VocabSpaceLimit}\left(\mathcal{V}, k, p, p(o^{\text{LLM}}|S_{\text{pub}})\right)$
10: $\quad v_{\text{syn}} = \arg\max_{v \in \mathcal{V}_{\text{pub}}} \frac{\max\{\hat{p}_v, 0\}}{\sum_{v' \in \mathcal{V}_{\text{pub}}} \max\{\hat{p}_{v'}, 0\}}$
11: $\quad \tilde{O} \leftarrow \tilde{O} + [v_{\text{syn}}]$
12: **end for**
13: **return** $\tilde{O}$

---

tion LLM on the concept underlying them as a result of an implicit Bayesian inference of ICL (Xie et al., 2022).

However, using demonstrations for any other concepts incurs additional privacy costs. For example, in a document classification task, accurately estimating the log-likelihood ratio requires evaluating combinations of concepts $(\theta^*, \theta)$. If demonstrations for any other concepts $\theta \neq \theta^*$ involve private information, additional noise must be introduced to ensure DP, potentially degrading the accuracy of DP-ICL.

To address this, we propose PTA for solving (11) without incurring additional privacy costs. PTA uses public prompts—composed of task instructions and tokens already generated, excluding private demonstrations—as substitutes for prompts representing any other concepts. Leveraging DP's post-processing immunity, public prompts incur no additional privacy costs when combined with generated tokens. This method enables the likelihood ratio to be estimated by comparing next-token probability distribution conditioned on private and public prompts:

$$\log \frac{p(o^{\text{LLM}}=v|\theta^*)}{p(o^{\text{LLM}}=v|\theta)} \approx \log \frac{p(o^{\text{LLM}}=v|S_{\text{priv}}^{(i)})}{p(o^{\text{LLM}}=v|S_{\text{pub}})}. \quad (12)$$

Since the likelihood ratio on the LHS of (12) are intractable to compute, the estimation on the RHS of (12) relies on the tractable next-token probability distribution conditioned on an observable private prompt $S_{\text{priv}}^{(i)}$ for the ground-truth concept and comparing them to an observable public prompt. This estimation effectively highlights the distinctive tokens without incurring additional privacy costs. The rationale and implication of having this estimation are further discussed in Appendix E.

By leveraging the estimated likelihood, PTA modifies the next-token probability distribution as follows:

$$p_v^{(i)} \propto \underbrace{p_{\text{base}}(o^{\text{LLM}} = v)}_{\text{base probability}} \underbrace{\left( \frac{p(o^{\text{LLM}} = v | S_{\text{priv}}^{(i)})}{p(o^{\text{LLM}} = v | S_{\text{pub}})} \right)^{\alpha}}_{\text{private to public ratio}}. \quad (13)$$

The derivation of this modification, which solves (11), is deferred to Appendix E. The public-to-private ratio amplifies tokens distinctive to the ground-truth concept, while the baseline probability $p_{\text{base}}(o^{\text{LLM}} = v)$ ensures coherence by reflecting the natural flow of tokens already generated. This prevents excessive likelihood shifts and avoids disrupting fluency when tokens are more likely under the public prompt than the private prompt. Further, the pre-tuned Gaussian noise is added to the modified probability to ensure DP.

The algorithm implementation of PTA is detailed in Algorithm 1. Subroutines used in Algorithm 1 are presented in Algorithms 2 and 3 in Appendix E. During each token generation step, the public prompt $S_{\text{pub}}$ and multiple private prompts $S_{\text{priv}}$ are generated in Line 3. The next-token probability distributions are then modified using PTA, as detailed in Line 4–7, to highlight tokens that distinctly represent the task. To ensure DP, the Gaussian mechanism is applied to the modified probability distribution, as described in Line 8. Finally, fixed and adaptive thresholds dynamically limit the vocabulary space, as implemented in Line 9, to filter out low-likelihood tokens and stabilize performance. This iterative process continues until the generated demonstrations reach the maximum sequence length $T$.

The DP guarantee associated with Algorithm 1 is identical to that described by (Tang et al., 2024) when the noise variance $\sigma^2$ is equivalent, indicating that the empirical privacy leakage is mitigated similarly to Tang et al.'s method. Although details are presented in Appendix D, this equivalence arises because both algorithms apply the Gaussian mechanism to next-token probabilities, which are projected onto probability simplex (i.e. these $\ell_2$-sensitivity is $\sqrt{2}$). PTA defined in (13), unique to this work, leverages public prompts, avoiding an increase in $\ell_2$-sensitivity or the need for additional noise. Moreover, the other algorithmic components are identical, enabling the privacy guarantee to be computed using the same numerical approach as outlined by (Gopi et al., 2021) and adopted by (Tang et al., 2024).

## 5. Experiments

### 5.1. Experimental setups

**Datasets.** We evaluated the proposed PTA against the method by (Tang et al., 2024) using one synthetic classification task, GINC, introduced by (Xie et al., 2022), and three real-world text-classification tasks: AGNews (4-way news classification) (Zhang et al., 2015), DBPedia (multi-class classification of Wikipedia articles) (Zhang et al., 2015), and TREC (6-way question classification) (Voorhees & Tice, 2000). Specifically, GINC is designed to align with the prompt setting in Section 2.2, enabling precise evaluation of the likelihood of generated DP synthetic demonstrations conditioned on specific concepts. This supports evaluating the theoretical condition in Lemma 1, which underpin the theoretical foundation of the proposed PTA. Further details about the datasets are available in Appendix F.2.

**Models and DP synthetic demonstration generation.** To generate DP synthetic demonstrations, we used three variants of the pre-trained GPT-2 model (Radford et al., 2018) with (MA1) 4-layer, (MA2) 12-layer, and (MA3) 16-layer. For the other text-classification tasks, we utilized three models: (MB1) Llama2-7B-GPTQ, (MB2) Llama2-7B (Touvron et al., 2023), and (MB3) Mistral-7B (Jiang et al., 2023)).

Using these models, we generated four DP synthetic demonstrations per task without replacement from the set of possible labels in the task. The next-token probabilities required for generating these demonstrations were obtained using a text-generation-inference platform[2]. For the GINC dataset, next-token probabilities were computed over a vocabulary size of $|\mathcal{V}| = 150$, following (Xie et al., 2022), whereas for other text-classification tasks, $|\mathcal{V}| = 32,000$ was used. To ensure a fair comparison, we adopted the same prompt format as (Tang et al., 2024). Further details on the pre-training of the GPT-2 model for GINC and the prompt format are provided in Appendices F.3 and F.9.

**DP parameters.** To evaluate robustness across different DP settings, we set $\varepsilon = \{1, 2, 4, 8, \infty\}$ with a fixed $\delta = 1/|\mathcal{D}_{\text{priv}}|$. The Gaussian noise variance $\sigma^2$ was determined by using a numerical method (Gopi et al., 2021), implemented with the authors' code[3], as detailed in Appendix D. For $\varepsilon = \infty$, to show the accuracy of ICL without using DP, the original private demonstrations were randomly sampled from the private dataset.

**Comparing methods.** We summarize the compared methods and their key features in Table 1. Our primary focus is on comparing the baseline method (B1) (Tang et al., 2024) with our proposed (P1) PTA. Additionally, we conducted an ablation study to examine the impact of individual features. (PA2) and (BA2) were designed to assess the impact of the adaptive threshold for limiting the vocabulary space, which balances token diversity and plausibility. Further, (PA3) was introduced to evaluate the role of the regularization term in (11), which aims to enhance natural token flow. We also

---

[2](Tang et al., 2024) used OpenAI's API to obtain next-token probabilities; however, the current API limits predictions to the top 20 tokens. To address this limitation, we used text-generation-inference, which allows access to full token spaces.

[3]`https://github.com/microsoft/prv_accountant`

Table 1: Comparison of methods and key features. Amplification represents whether the estimates of log-likelihood ratio in (12) are used, Base prob. represents whether base probability in (13) is applied, and Top-$p$ represents Line 2 in Algorithm 2.

| Method | Amplification | Base Prob. | Top-$p$ |
|---|---|---|---|
| (R0) 4-shot ICL using $\mathcal{D}_{\text{priv}}$ | | | |
| (B1) Baseline | | | |
| (BA2) Baseline w. Top-$p$ | | | ✓ |
| (P1) PTA | ✓ | ✓ | |
| (PA2) PTA w. Top-$p$ | ✓ | ✓ | ✓ |
| (PA3) PTA w.o. Base Prob. | ✓ | ✓ | |

Table 2: 4-shot DP-ICL accuracy across four datasets using the best-performing model: (MA3) for GINC and (MB3) for the other datasets, averaged over five different seeds. The highest accuracy is bolded, and the second-highest is underlined. Full results are available in Table 8.

| $\varepsilon$ | Method | GINC | AGNews | DBPedia | TREC |
|---|---|---|---|---|---|
| | (P1) | $\mathbf{93.99_{\pm 1.35}}$ | $\mathbf{87.88_{\pm 1.11}}$ | $82.06_{\pm 3.61}$ | $\mathbf{84.80_{\pm 1.66}}$ |
| | (PA2) | $90.06_{\pm 1.63}$ | $87.00_{\pm 1.16}$ | $84.72_{\pm 3.54}$ | $\underline{84.72_{\pm 2.50}}$ |
| 1 | (PA3) | $90.14_{\pm 0.95}$ | $83.24_{\pm 2.94}$ | $\mathbf{86.56_{\pm 1.75}}$ | $84.34_{\pm 1.28}$ |
| | (B1) | $\underline{90.80_{\pm 2.79}}$ | $83.74_{\pm 1.90}$ | $85.72_{\pm 1.44}$ | $83.56_{\pm 5.09}$ |
| | (BA2) | $89.34_{\pm 0.98}$ | $84.86_{\pm 2.83}$ | $\underline{86.10_{\pm 1.83}}$ | $81.00_{\pm 2.57}$ |
| | (P1) | $\mathbf{96.61_{\pm 1.40}}$ | $\mathbf{86.48_{\pm 1.67}}$ | $84.78_{\pm 1.55}$ | $\underline{84.24_{\pm 2.33}}$ |
| | (PA2) | $91.15_{\pm 1.24}$ | $84.24_{\pm 2.20}$ | $84.12_{\pm 3.47}$ | $83.52_{\pm 1.02}$ |
| 8 | (PA3) | $90.79_{\pm 1.16}$ | $\underline{84.92_{\pm 1.75}}$ | $\mathbf{85.56_{\pm 2.34}}$ | $\mathbf{84.54_{\pm 1.93}}$ |
| | (B1) | $\underline{94.63_{\pm 0.55}}$ | $83.62_{\pm 3.08}$ | $84.40_{\pm 2.39}$ | $82.64_{\pm 2.79}$ |
| | (BA2) | $90.61_{\pm 1.23}$ | $82.96_{\pm 2.32}$ | $\underline{84.86_{\pm 1.38}}$ | $81.96_{\pm 1.59}$ |
| $\infty$ | (R0) | $\mathbf{99.02_{\pm 0.28}}$ | $\mathbf{87.82_{\pm 1.22}}$ | $\mathbf{87.38_{\pm 1.30}}$ | $\mathbf{82.24_{\pm 1.70}}$ |

include (R0) as a non-private reference that used the original private demonstrations. To ensure fair comparisons across all configurations, a comprehensive hyperparameter search was conducted, as detailed in Appendix F.5.

## 5.2. Main results

**Accuracy comparison.** Table 2 presents the accuracy of DP-ICL for selected configurations and privacy parameters $\varepsilon = \{1, 8, \infty\}$ and the best-performing models (MA3) and (MB3). The full results, including additional configurations and privacy parameters, are provided in Appendix F.5. Particularly, (P1) consistently achieves superior accuracy to baselines (B1, BA2), except in the DBPedia at $\varepsilon = 1$, thereby validating its effectiveness across diverse privacy parameters and datasets. However, on DBPedia at $\varepsilon = 1$, (PA3) performs best. Since (PA3) amplifies the distinctive tokens but ignores the KL regularization term of (11) used in (P1), this result suggests that the KL term may act as a constraint that limits useful adaptation to the target distribution in certain settings. Nevertheless, by effectively amplifying the distinctive tokens in the task before adding noise, (P1) and its variant ensure that the highest-probability tokens are

Table 3: Proportion of generated DP synthetic demonstrations for GINC, using model (MA3), where the log-likelihood ratio exceeds the derived threshold, appearing in the RHS of Lemma 1.

| $\varepsilon$ | (P1) | (PA2) | (PA3) | (B1) | (BA2) |
|---|---|---|---|---|---|
| 1 | $\mathbf{49.33}$ | 40.89 | 38.22 | 44.89 | $\underline{42.22}$ |
| 8 | $\mathbf{53.33}$ | 41.33 | 42.67 | $\underline{52.44}$ | 44.45 |

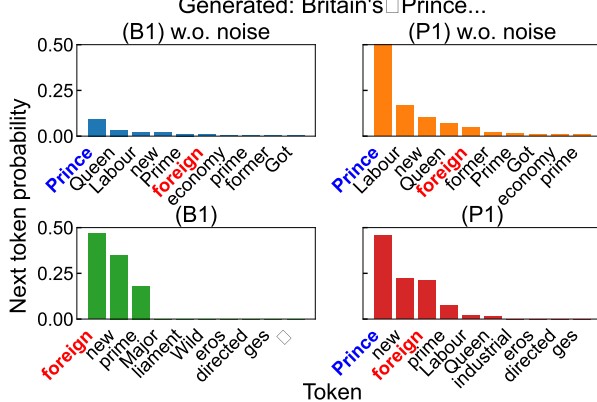

Figure 2: Next-token probabilities comparing (Top Left) (B1) w.o. noise addition, (Top Right) (P1) w.o. noise addition, (Bottom Left) (B1), and (Bottom Right) (P1).

sampled correctly, even under larger noise variance (e.g., AGNews with $\varepsilon = 1$).

**Distinguishability of generated demonstrations on GINC.** To demonstrate the positive effect of PTA on accurate concept inference, we empirically evaluate the log-likelihood ratio $\log(p(\tilde{O}|\theta^*)/p(\tilde{O}|\theta))$ appearing in Lemma 1 using GINC. The unique controlled setup of GINC enables us to numerically compute the likelihood ratio, as shown in Appendix F.4. According to the theoretical results established in Lemma 1 and Theorem 2, a larger value of the log-likelihood ratio indicates a more accurate inference of the ground-truth concept. To empirically validate this, we calculate the probability that $\Pr[\text{RHS of (8) is negative}]$. This measure quantifies how much the proportion of the generated DP synthetic demonstration helps in the successful inference on the ground-truth concept, since if RHS of (8) is negative, then the LLM accurately infers the ground-truth concept. As shown in Table 3, (P1) consistently achieves better values than the baselines. This result validates that PTA enlarges the divergence between $p(\tilde{O}|\theta^*)$ and $p(\tilde{O}|\theta)$, enhancing LLM's ability to infer the ground-truth concept. The alignment of these empirical findings with the theoretical results strongly supports PTA's effectiveness.

**Demystifying example of the effect of PTA.** We present a specific example of a next-token probability distribution to illustrate how PTA effectively amplifies the distinctive tokens conditioned on private demonstrations, as shown in

Figure 2. The top row in Figure 2 shows next-token probability distribution computed by using (2) and (13), while the bottom row shows the probability distribution after noise addition. Focusing on the tokens "Prince" and "foreign", (P1) correctly preserves the top-1 token "Prince" even after noise is added, maintaining task relevance. In contrast, (B1) fails to retain the top-1 token, instead selecting a token "foreign" with a lower likelihood due to noise. This example highlights PTA's robustness in retaining critical tokens under high noise, ensuring task accuracy. Other examples can be found in Appendix F.8

## 6. Conclusion

We proposed PTA, a novel method for generating DP synthetic demonstrations, which highlights tokens that distinctly represent ground-truth concepts underlying the original demonstrations. By interpreting ICL as implicit Bayesian inference, we not only theoretically demonstrate that limiting vocabulary space, empirically incorporated in (Tang et al., 2024), mitigates the negative impact of noise on concept inference but also introduced a refined method for modifying next-token probabilities for accurate inference on the ground-truth concept. We integrate PTA with vocabulary space limitation to ensure that DP synthetic demonstrations remain aligned with ground-truth concepts. Numerical experiments demonstrated the effectiveness of PTA, showing accuracy improvements, particularly in high-privacy regimes (e.g., $\varepsilon = 1$).

## Impact statement

This paper addresses the challenge of privacy-preserving machine learning, particularly for LLM applications. To safeguard individual data used as input to an LLM, we propose Plausible Token Amplification (PTA) for generating synthetic demonstrations that satisfy DP guarantees. By leveraging DP, our approach lowers the risk of an individual being inferred from an LLM's output, thereby offering rigorous privacy guarantees and fostering safer data collaboration. As with other DP-based methods, unintended leakage can arise without careful design of the accumulation of privacy parameters and pre-processing steps (e.g. de-duplication of data). These potential risks are consistent with issues commonly encountered in broader DP applications. Nevertheless, our work demonstrates that incorporating DP into LLM workflows can promote both privacy protection and fair, efficient data usage.

## Acknowledgements

We thank Ayato Nakagawa for his engineering support in numerical experiments.

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

# A. Limitations and future work

We briefly discuss the limitations and future directions of this study.

First, the introduced HMM-based prompt formulation may not fully capture the auto-regressive nature of generating DP synthetic demonstrations, where generated tokens recursively serve as input for next-token generation. The introduced HMMs overlook the effect of noise addition on transitions to the next hidden states. Addressing this could provide a more realistic representation of the DP synthetic prompt generation process.

Second, our framework could support efforts to adaptively noise variance reduction by estimating the maximum changes in the next-token probability of LLM for individual demonstrations, as proposed in (Gao et al., 2025), which complements our likelihood ratio-driven improvements in PTA. Integrating the proposed PTA with such extensions could further enhance its effectiveness by highlighting task-specific information and tailored noise variance, marking a fruitful direction for future work.

Third, Theorem 2 may suggest a reformulation in which the vocabulary space is limited to directly maximizing the divergence between private and public next-token probability distributions, rather than being fixed in advance based on public information as in our current compositional design. Specifically, selecting the vocabulary to maximize this divergence could further enhance alignment with the ground-truth concept while mitigating the noise error term in Theorem 2. However, this approach introduces challenges, such as the need for more complex DP mechanisms to account for vocabulary selection based on private information. We leave this as an extension of our current PTA.

# B. Related works

In this section, we will outline the methods that are related to this research and discuss their positioning in relation to it. To improve the accuracy of DP-ICL, the proposed PTA uses DP synthetic demonstrations. In generating DP synthetic demonstrations, the next-token probability is modified to improve the likelihood of the generated tokens. The following discussion is organized around three key perspectives: (i) DP-ICL methods, (ii) DP synthetic text generation techniques using LLMs, and (iii) methods for modifying the next-token probability distribution in LLMs.

## (i) DP-ICL methods.

(Duan et al., 2023) first highlighted the leakage risks of demonstrations provided in prompts for ICL. This motivates a series of works on applying DP-ICL. Many DP-ICL methods have been widely explored to mitigate such leakage risks (Tang et al., 2024; Wu et al., 2024; Gao et al., 2025).

A key approach in DP-ICL is generating DP synthetic demonstrations as alternatives for original ones in prompts. (Tang et al., 2024) first introduced this approach by adding noise to a next-token probability distribution conditioned on the original demonstrations. Their method also limits the vocabulary space to stabilize the accuracy, and its effectiveness has been empirically confirmed. The generated DP synthetic demonstrations are then used in prompts, leveraging DP's post-processing immunity for privacy preservation in query response.

While our proposed PTA indeed follows this work, it fundamentally differs in two key ways. i) Instead of relying solely on empirical validation, we provide theoretical support for the effectiveness of limiting vocabulary space by interpreting ICL as implicit Bayesian inference on the shared concept among demonstrations. ii) PTA goes beyond Tang et al.'s vocabulary space limitation by explicitly aligning the generated DP synthetic demonstrations with the ground-truth concept. Specifically, PTA highlights tokens that distinctly represent the task, ensuring that the LLM infers the ground-truth concept more accurately — a crucial aspect yet not considered in (Tang et al., 2024).

Concurrently with our work on generating DP synthetic demonstrations, (Gao et al., 2025) proposed the first data-adaptive DP-ICL. Their method adaptively reduces the noise variance $\sigma^2$ on the basis of the dynamic estimation of the sensitivity of the next-token probability distribution. This approach complements our PTA framework: while they focus on mitigating the negative noise impact by reducing its variance, PTA emphasizes tokens that distinctly represent the ICL task to enhance the likelihood of generated demonstrations on the ground-truth concept. Integrating their adaptive method with our PTA could potentially further improve the accuracy of DP-ICL, representing a promising direction for future research.

In contrast to synthetic demonstration approaches, (Wu et al., 2024) introduced a DP-ICL framework that applies the Gaussian mechanism directly to aggregated next-token probability distribution for classification. For generation tasks, it uses embedding-based and keyword-based aggregation. Although effective for various tasks, this approach leads to an increase in the noise variance $\sigma^2$ as the number of queries grows, subsequently degrading its accuracy. Due to this, we have opted not to adopt this method. In contrast, our PTA utilizes DP synthetic demonstrations in the prompt, which avoids the addition of noise to every query response. This strategy allows us to have an unlimited number of queries.

Another line of research in DP-ICL does not rely on the Gaussian mechanism. For example, (Hong et al., 2024) (DP-OPT) and (Amin et al., 2024) adopt the Exponential Mechanism (McSherry & Talwar, 2007), leveraging random sampling instead of adding noise to the next-token probability distribution. In DP-OPT, a Limited-Domain Mechanism (Durfee & Rogers, 2019) selects tokens from the next-token probability distribution while mitigating the curse of dimensionality caused by large vocabularies. Similarly, (Amin et al., 2024) employ the Exponential Mechanism for next-token selection by exploiting the equivalence between softmax sampling in log space and the Exponential Mechanism, further improving efficiency via the Sparse Vector Technique (Dwork et al., 2009).

In contrast to these sampling-based approaches that select tokens through uncertain sampling, our PTA will explicitly highlight tokens that distinctly represent the ICL task. This ensures that generated DP synthetic demonstrations assist an LLM in accurately inferring the ground-truth concept. Nevertheless, both sampling-based methods and noise-based methods can be analyzed within an extended Bayesian framework for ICL, offering a fair comparison of how different DP-ICL techniques improve accuracy under privacy constraints.

## (ii) DP synthetic text generation.

Our work is part of the broader category of DP synthetic text generation, which aims to generate text satisfying the target DP guarantee while maintaining practical utility. Traditionally, this has involved using DP-Stochastic Gradient Descent

(DP-SGD) (Abadi et al., 2016), which incorporates DP mechanisms into the training of generative model parameters using private data. The resulting model then produces DP synthetic text (Yue et al., 2023; Mattern et al., 2022; Flemings & Annavaram, 2024). However, the training of LLMs using DP-SGD has become impractical due to the significant computational resources required and the unavailability of model parameters in closed-source settings.

To address these issues, (Xie et al., 2024) introduced an API-based method that leverages LLM inference APIs to generate candidate synthetic texts through random sampling, which are then refined by using a selection mechanism. The selection mechanism involves private data, as it compares synthetic samples with private data to retain those that best represent the original text distribution. Since this step poses a risk of privacy leakage, noise is added to the selection mechanism, ensuring DP.

Compared to these existing DP synthetic text generation approaches, our PTA offers a tailored approach to improve the accuracy of DP-ICL. PTA, especially aligns the generated DP synthetic demonstrations with the ground-truth concept, thereby improving ICL accuracy from a Bayesian perspective. Furthermore, PTA can be seamlessly integrated into LLM inference processes, even in closed-source or resource-limited scenarios assumed in (Xie et al., 2024), as it does not require direct access to LLM parameters.

**(iii) Modification of the next-token probability of LLMs.** PTA leverages next-token modification to highlight tokens that enable an LLM to accurately infer the ground-truth concept. Beyond DP applications, next-token modification is widely used for various purposes, such as enhancing diversity and coherence in open-ended text generation (Li et al., 2023), improving factual consistency (Shi et al., 2024), and preventing toxic or harmful outputs (Xu et al., 2024).

Unlike these approaches, our PTA uniquely applies next-token modification to optimize DP synthetic demonstrations, ensuring they are both privacy-preserving and highly informative for LLM inference.

## C. Missing proof of Bayesian analysis of DP-ICL

In this section, we provide the details missing from Section 3. First, we provide the HMM-based prompt formulation (Appendix C.1) and assumptions (Appendix C.2) used in (Xie et al., 2022) and this paper. Then, we provide the simple transformation of (5) in Appendix C.4. In Appendix C.3, we list the useful mathematical tools used in the subsequent proof. Finally, we provide the detailed proof of Lemma 1 and Theorem 2 in Appendices C.5–C.8.

### C.1. Prompt formulation

To formalize the HMM-based prompt formulation described in Section 2.2, we let $\Theta$ be a set of concepts, where each concept $\theta \in \Theta$ defines the transition probability matrix of an HMM. The prompt is formulated as follows:

$$[S_n, x_{\text{query}}] = [O_1, o_1^{\text{delim}}, \ldots, O_n, o_n^{\text{delim}}, x_{\text{query}}] \sim p_{\text{prompt}}, \tag{14a}$$

$$O_i = [o_{i,1}, \ldots, o_{i,T}], \quad o_{i,j} \sim p(o_{i,j}|h_{i,j}), \tag{14b}$$

$$h_{i,j} \sim p(h_{i,j}|h_{i,j-1}, \theta^*), \forall j > 1, \quad h_{i,1} \sim p_{\text{prompt}}, \tag{14c}$$

$$o_i^{\text{delim}} \sim p(o_i^{\text{delim}}|h_i^{\text{delim}}, \theta^*), \tag{14d}$$

where $n$ denotes the number of demonstrations in the prompt and $T$ is the length of each demonstration. In (14a), the $i$-th demonstration and delimiter token like "\n" are denoted by $O_i$ and $o_i^{\text{delim}}$, respectively, and $x_{\text{query}}$ represents a query for ICL. According to the Markov property, each token $o_{i,j}$ is sampled based on its emission probability $p(o_{i,j}|h_{i,j})$ in (14b). As depicted in (14c), the transition between hidden states follows the HMM defined by the concept $\theta^*$ underlying the demonstrations, and the initial hidden state of each demonstration follows $p_{\text{prompt}}$. To concatenate these demonstrations into a prompt, a delimiter token is typically inserted after each demonstration, acting as a boundary between them. The transition to this token is also governed by the HMM, as shown in (14d).

To distinguish the different domains that probability distributions consider, we use the following notation as needed for clarity: $p_{\text{prompt}|\mathcal{V}}$ for the token emission probability distribution and $p_{\text{prompt}|\mathcal{V}^T}$ for the demonstration distribution.

As shown in (6), we introduce the perturbed prompt distribution to model the noise addition instead of using (14b) to model the DP synthetic demonstration. To ensure it remains a valid probability distribution, we have formally implemented the following formulation of the DP synthetic prompt distribution.

$$[\tilde{S}_n, x_{\text{query}}] = [\tilde{O}_1, o_1^{\text{delim}}, \ldots, \tilde{O}_n, o_n^{\text{delim}}, x_{\text{query}}] \sim \tilde{p}_{\text{prompt}} = \mathbb{E}_{\boldsymbol{\zeta}}\left[\hat{p}_{\text{prompt}}\right], \tag{15a}$$

$$\hat{p}_{\text{prompt}}(o_{i,j} = v|h_{i,j}) = \frac{\max\{0, p_{\text{prompt}}(o_{i,j} = v|h_{i,j}) + \zeta_v\}}{\sum_{v \in \mathcal{V}} \max\{0, p_{\text{prompt}}(o_{i,j} = v|h_{i,j}) + \zeta_v\}}, \quad \boldsymbol{\zeta} = (\zeta_v)_{v \in \mathcal{V}} \sim \mathcal{N}(0, \sigma^2 I_{|\mathcal{V}|}). \tag{15b}$$

Consequently, our HMM-based formulation for the DP synthetic demonstrations follows (14c), (14d), (15a) and (15b).

### C.2. Assumptions

Following (Xie et al., 2022), we detail the assumption regarding transition probabilities and emission probabilities of HMMs used in the prompt formulation. Under these assumptions, we quantify the mismatch between the prompt distribution and demonstrations in the prompt.

**Assumption 1** (Delimiter hidden states). Let the delimiter hidden states $\mathcal{D}$ be a subset of $\mathcal{H}$. For any $h^{\text{delim}} \in \mathcal{D}$ and $\theta \in \Theta$, $p(o^{\text{delim}}|h^{\text{delim}}, \theta^*) = 1$ and for any $h \notin \mathcal{D}$, $p(o^{\text{delim}}|h, \theta^*) = 0$.

**Assumption 2** (Bound on delimiter transitions). For any delimiter state $h^{\text{delim}} \in \mathcal{D}$ and any hidden state $h \in \mathcal{H}$, the probability of transitioning to a delimiter hidden state under $\theta$ is upper bounded $p(h^{\text{delim}}|h, \theta) < c_2$ for any $\theta \in \Theta \setminus \{\theta^*\}$, and is lower bounded $p(h^{\text{delim}}|h, \theta^*) > c_1 > 0$ for $\theta^*$. Additionally, the start hidden state distribution for delimiter hidden states is bounded as $p(h^{\text{delim}}|\theta) \in [c_3, c_4]$.

**Assumption 3** (Well-specification). The prompt concept $\theta^*$ is in the concept set $\Theta$.

**Assumption 4** (Regularity). The pretraining distribution $p$ satisfies: 1) Lower bound on transition probability for the prompt concept $\theta^*$: for any pair of hidden states $h, h' \in \mathcal{H}$, $p(h|h', \theta^*) > c_5 > 0$. 2) Start hidden state is lower bounded: for any $h \in \mathcal{H}$, $p(h|\theta^*) \geq c_8 > 0$. 3) All tokens can be emitted: for every symbol $o$, there is some hidden state $h \in \mathcal{H}$ such that $p(o|h, \theta^*) > c_6 > 0$. 4) The prior $p(\theta)$ has support over the entire concept family $\Theta$ and is bounded above everywhere.

## C.3. Technical definitions and lemmas

In this section, we list the mathematical tools to provide the proof in this paper.

**Definition 1** (Total Variation(TV) distance). Given two distributions $p, q$ over a finite domain $\mathcal{X}$, total variation(TV) distance is defined as follows:

$$d_{\mathrm{TV}}(p, q) := \frac{1}{2} \sum_{x \in \mathcal{X}} |p(x) - q(x)|.$$

**Lemma 2** (Moments of rectified Gaussian distribution (Beauchamp, 2018)). *Suppose $X$ follows $\mathcal{N}(\mu, \sigma^2)$. Then, random variable $\hat{X} = \max\{0, X\}$ is said to follow the rectified Gaussian distribution. Its first and second order moments are given by:*

$$\mathbb{E}\left[\hat{X}\right] = \mu \left(1 - \Phi(-\frac{\mu}{\sigma})\right) + \sigma \phi(-\frac{\mu}{\sigma}),$$
$$\mathbb{E}\left[\hat{X}^2\right] = (\mu^2 + \sigma^2) \left(1 - \Phi(-\frac{\mu}{\sigma})\right) + \mu\sigma \phi(-\frac{\mu}{\sigma}).$$

**Lemma 3** (Csiszar inequality (Wilde, 2011)). *Given two distributions $p, q$ over a finite domain $\mathcal{X}$, then we have the following*

$$|H(p) - H(q)| \leq d_{\mathrm{TV}}(p, q) \log |\mathcal{X}| + h(d_{\mathrm{TV}}(p, q)),$$

*where $H(\cdot)$ denotes the Shannon Entropy, $d_{\mathrm{TV}}(\cdot, \cdot)$ denotes the TV distance and $h(\cdot)$ represents the binary entropy function.*

**Lemma 4** (Bretagnolle–Huber bound (Bretagnolle & Huber, 1979)). *Given two distributions $p, q$ over a finite domain $\mathcal{X}$, then we have:*

$$d_{\mathrm{TV}}(p, q) \leq \sqrt{1 - \exp\left(-D_{\mathrm{KL}}(p\|q)\right)}.$$

**Lemma 5** (Reverse Pinsker's inequality (Sason & Verdú, 2016)). *Given two distributions $p, q$ over a finite domain $\mathcal{X}$, then we have:*

$$D_{\mathrm{KL}}(p\|q) \leq \frac{\log 2}{\min_{x \in \mathcal{X}} q(x)} d_{\mathrm{TV}}(p, q)^2.$$

**Lemma 6** (Hoeffding's inequality (Hoeffding, 1994)). *Let $Z_1, \ldots, Z_n$ be independent bounded random variables with $Z_i \in [a, b]$ for all $i$, where $-\infty < a \leq b < \infty$. Then, we have:*

$$P\left\{\frac{1}{n} \sum_{i=1}^{n} (Z_i - \mathbb{E}_{Z_i}[Z_i]) \geq t\right\} \leq \exp\left(-\frac{2nt^2}{(b-a)^2}\right),$$
$$P\left\{\frac{1}{n} \sum_{i=1}^{n} (Z_i - \mathbb{E}_{Z_i}[Z_i]) \leq -t\right\} \leq \exp\left(-\frac{2nt^2}{(b-a)^2}\right),$$

*for all $t \geq 0$.*

### C.4. Preliminary results and transformation of (5)

Here, we begin by recalling the original statement from (Xie et al., 2022).

**Theorem 3.** *Suppose Assumptions 1,2,3, and 4 hold. Then, we have $\lim_{n\to\infty} \exp{(n \cdot r_n(\theta))} = 0$, if $\theta^*$ is the ground-truth concept of the prompt and satisfies the following condition for any other concept $\theta \in \Theta$:*

$$\sum_{j=1}^{T} \mathbb{E}\left[D_{KL}(p_{prompt}(O_j|O_{1:j-1})\|p(O_j|O_{1:j-1},\theta))\right] > C^{start} + C^{delim},$$

*where $p_{prompt}$ and $p$ represent the prompt distribution and pre-training distribution, respectively. The expectation is taken over $O \sim p_{prompt}$. $C^{delim}$ is a positive constant.*

By using a straightforward transformation, we derive Theorem 3 from (5). To do so, we first transform the LHS of (5) as follows:

$$\mathbb{E}_{O\sim p_{\text{prompt}}}\left[\log \frac{p(O|\theta^*)}{p(O|\theta)}\right] = D_{\text{KL}}\left(p_{\text{prompt}}(O)\|p(O|\theta)\right) - D_{\text{KL}}\left(p_{\text{prompt}}(O)\|p(O|\theta^*)\right).$$

On the basis of the HMMs formulation, we can decompose the second KL-term into:

$$\begin{aligned}
D_{\text{KL}}\left(p_{\text{prompt}}(O)\|p(O|\theta^*)\right) &= \sum_{\boldsymbol{v}\in\mathcal{V}} p_{\text{prompt}}(O=\boldsymbol{v}) \log \frac{\sum_{H\in\mathcal{H}^T} p_{\text{prompt}}(h_1)p(h_2,\ldots,h_T|\theta^*)\prod_{j=1}^{T}p(o_j=v_j|h_j)}{\sum_{H\in\mathcal{H}^T} p(h_1|\theta^*)p(h_2,\ldots,h_T|\theta^*)\prod_{j=1}^{T}p(o_j=v_j|h_j)} \\
&\leq \sum_{\boldsymbol{v}\in\mathcal{V}} p_{\text{prompt}}(O=\boldsymbol{v}) \log \frac{1}{c_8} = \log \frac{1}{c_8}.
\end{aligned} \tag{16}$$

The inequality holds because applying Assumption 4 gives the following:

$$\forall h \in \mathcal{H}, \quad \frac{p_{\text{prompt}}(h)}{p(h|\theta^*)} \leq \frac{1}{c_8}.$$

Next, we rewrite the first KL term.

$$\begin{aligned}
D_{\text{KL}}\left(p_{\text{prompt}}(O)\|p(O|\theta)\right) &= \sum_{\boldsymbol{v}\in\mathcal{V}^T} p_{\text{prompt}}(O=\boldsymbol{v}) \log \prod_{j=1}^{T} \frac{p_{\text{prompt}}(O_j|O_{1:j-1})}{p(O_j|O_{1:j-1},\theta)} \\
&= \sum_{\boldsymbol{v}\in\mathcal{V}^T} \sum_{j=1}^{T} p_{\text{prompt}}(O=\boldsymbol{v}) \log \frac{p_{\text{prompt}}(O_j|O_{1:j-1})}{p(O_j|O_{1:j-1},\theta)} \\
&= \sum_{j=1}^{T} \sum_{\boldsymbol{v}\in\mathcal{V}^T} p_{\text{prompt}}(O=\boldsymbol{v}) \log \frac{p_{\text{prompt}}(O_j|O_{1:j-1})}{p(O_j|O_{1:j-1},\theta)} \\
&= \sum_{j=1}^{T} \mathbb{E}\left[D_{\text{KL}}(p_{\text{prompt}}(O_j|O_{1:j-1})\|p(O_j|O_{1:j-1},\theta))\right].
\end{aligned}$$

By using the transformed KL-term and (16) into (5), then we obtain another condition for the convergence of $r(\theta)$ as shown in Theorem 3.

## C.5. Proof of Lemma 1

The following lemma and its proof are nearly identical to Lemma 8 in (Xie et al., 2022) but we present it for a self-contained purpose.

**Lemma.** *Suppose Assumptions 1,2,3, and 4 hold. Then, we have*

$$\tilde{r}_n(\theta) \leq -\frac{1}{n}\sum_{i=1}^{n}\log\frac{p(\tilde{O}_i|\theta^*)}{p(\tilde{O}_i|\theta)} + C^{delim} + O(n^{-1}).$$

*Proof.* First we transform $p(\tilde{S}_n, x_{\text{query}}|\theta), \forall\theta\in\Theta$ as follows:

$$p(\tilde{S}_n, x_{\text{query}}|\theta) = p(x_{\text{query}}|\tilde{S}_n, \theta)p(\tilde{S}_n|\theta) = p(\tilde{S}_n|\theta)\left[\sum_{h_{\text{query}}^{\text{start}}\in\mathcal{H}} p(x_{\text{query}}|h_{\text{query}}^{\text{start}}, \theta)p(h_{\text{query}}^{\text{start}}|\tilde{S}_n, \theta)\right],$$

where for the last equation we used the fact that $x_{\text{query}}$ and $\tilde{S}_n$ are conditionally independent given $h_{\text{query}}^{\text{start}}$.

Let $\tilde{O}_i^{\text{ex}} = [o_{i-1}^{\text{delim}}, \tilde{O}_i]$ be the $i-1$-th delimiter token followed by the $i$-th observation sequence. For $i=1$ we define $\tilde{O}_1^{\text{ex}} = \tilde{O}_1$. Accordingly, we can decompose $p(\tilde{S}_n|\theta)$ as follows:

$$\begin{aligned}
p(\tilde{S}_n|\theta) =& p(o_n^{\text{delim}}, \tilde{O}_{1:n}^{\text{ex}}|\theta)\\
=& p(o_n^{\text{delim}}|\tilde{O}_{1:n}^{\text{ex}}, \theta)\prod_{i=1}^{n}p(\tilde{O}_i^{\text{ex}}|\tilde{O}_{1:i-1}^{\text{ex}}, \theta)\\
=& \sum_{h_n^{\text{delim}}\in\mathcal{D}} p(o_n^{\text{delim}}|h_n^{\text{delim}}, \theta)p(h_n^{\text{delim}}|\tilde{O}_{1:n}^{\text{ex}}, \theta)\prod_{i=1}^{n}\sum_{h_{i-1}^{\text{delim}}\in\mathcal{D}} p(\tilde{O}_i^{\text{ex}}|h_{i-1}^{\text{delim}}, \theta)p(h_{i-1}^{\text{delim}}|\tilde{O}_{1:i-1}^{\text{ex}}, \theta)\\
=& \prod_{i=1}^{n}\sum_{h_{i-1}^{\text{delim}}\in\mathcal{D}} p(\tilde{O}_i^{\text{ex}}|h_{i-1}^{\text{delim}}, \theta)p(h_{i-1}^{\text{delim}}|\tilde{O}_{1:i-1}^{\text{ex}}, \theta),
\end{aligned}$$

where for the third line we used the fact that $o_n^{\text{delim}}$ and $\tilde{O}_{1:n}^{\text{ex}}$ are conditionally independent given $h_n^{\text{delim}}$. Similarly, $\tilde{O}_{1:i}^{\text{ex}}$ and $\tilde{O}_{1:i-1}^{\text{ex}}$ are conditionally independent given $h_{i-1}^{\text{delim}}$ for $i = 1, \ldots, n$. We used total probability and $\forall i, p(o_i^{\text{delim}}|h_i^{\text{delim}}) = 1$ (Assumption 1) in the last line.

For $\theta \neq \theta^*$, we have the following upper bound:

$$\begin{aligned}
&\sum_{h_{i-1}^{\text{delim}}\in\mathcal{D}} p(\tilde{O}_i|h_{i-1}^{\text{delim}}, \theta)p(h_{i-1}^{\text{delim}}|\tilde{O}_{1:i-1}^{\text{ex}}, \theta)\\
\leq& c_2\sum_{h_{i-1}^{\text{delim}}\in\mathcal{D}} p(\tilde{O}_i|h_{i-1}^{\text{delim}}, \theta)\\
=& c_2\sum_{h_{i-1}^{\text{delim}}\in\mathcal{D}}\sum_{h_i^{\text{start}}\in\mathcal{H}} p(\tilde{O}_i|h_i^{\text{start}}, \theta)p(h_i^{\text{start}}|h_{i-1}^{\text{delim}}, \theta)\\
=& c_2\sum_{h_{i-1}^{\text{delim}}\in\mathcal{D}}\sum_{h_i^{\text{start}}\in\mathcal{H}} p(\tilde{O}_i|h_i^{\text{start}}, \theta)p(h_i^{\text{start}}|\theta)\frac{p(h_i^{\text{start}}|h_{i-1}^{\text{delim}}, \theta)}{p(h_i^{\text{start}}|\theta)}\\
=& c_2\sum_{h_{i-1}^{\text{delim}}\in\mathcal{D}}\sum_{h_i^{\text{start}}\in\mathcal{H}} p(\tilde{O}_i|h_i^{\text{start}}, \theta)p(h_i^{\text{start}}|\theta)\frac{p(h_i^{\text{start}}, h_{i-1}^{\text{delim}}, \theta)}{p(h_{i-1}^{\text{delim}}, \theta)}\frac{p(\theta)}{p(h_i^{\text{start}}, \theta)}\\
=& c_2\sum_{h_i^{\text{start}}\in\mathcal{H}} p(\tilde{O}_i|h_i^{\text{start}}, \theta)p(h_i^{\text{start}}|\theta)\sum_{h_{i-1}^{\text{delim}}\in\mathcal{D}}\frac{p(h_{i-1}^{\text{delim}}|h_i^{\text{delim}}, \theta)}{p(h_{i-1}^{\text{delim}}|\theta)}\\
\leq& c_2\sum_{h_i^{\text{start}}\in\mathcal{H}} p(\tilde{O}_i|h_i^{\text{start}}, \theta)p(h_i^{\text{start}}|\theta)\sum_{h_{i-1}^{\text{delim}}\in\mathcal{D}}\frac{p(h_{i-1}^{\text{delim}}|h_i^{\text{delim}}, \theta)}{c_3} = \frac{c_2}{c_3}p(\tilde{O}_i|\theta).
\end{aligned}$$

In the first inequality, we used $p(h_{i-1}^{\text{delim}}|O_{1:i-1}^{\text{ex}}\theta) < c_2$ (Assumption 2). We also used $p(h^{\text{delim}}|\theta) \in [c_3, c_4]$ (Assumption 2) in the second inequality. Subsequently, we marginalized out $h_{i-1}^{\text{delim}}, h_i^{\text{start}}$ in the last equality.

Symmetrically, we have the following lower bound for $\theta = \theta^*$:

$$\sum_{h_{i-1}^{\text{delim}}\in\mathcal{D}} p(O_i|h_{i-1}^{\text{delim}},\theta^*)p(h_{i-1}^{\text{delim}}|O_{1:i-1}^{\text{ex}},\theta^*) \geq c_1 \sum_{h_{i-1}^{\text{delim}}\in\mathcal{D}} p(O_i|h_{i-1}^{\text{delim}},\theta^*) \geq \frac{c_1}{c_4}p(O_i|\theta^*).$$

To derive the upper bound of $\tilde{r}_n(\theta)$, we rewrite it as follows:

$$\tilde{r}_n(\theta) = \frac{1}{n} \log \frac{p(\tilde{S}_n, x_{\text{query}}|\theta)}{p(\tilde{S}_n, x_{\text{query}}|\theta^*)}$$

$$= \frac{1}{n} \left[ \log \frac{\sum_{h_{\text{query}}^{\text{start}}\in\mathcal{H}} p(x_{\text{query}}|h_{\text{query}}^{\text{start}},\theta)p(h_{\text{query}}^{\text{start}}|\tilde{S}_n,\theta)}{\sum_{h_{\text{query}}^{\text{start}}\in\mathcal{H}} p(x_{\text{query}}|h_{\text{query}}^{\text{start}},\theta^*)p(h_{\text{query}}^{\text{start}}|\tilde{S}_n,\theta^*)} + \sum_{i=1}^{n} \log \frac{\sum_{h_{i-1}^{\text{delim}}\in\mathcal{D}} p(\tilde{O}_i^{\text{ex}}|h_{i-1}^{\text{delim}},\theta)p(h_{i-1}^{\text{delim}}|\tilde{O}_{1:i-1}^{\text{ex}},\theta)}{\sum_{h_{i-1}^{\text{delim}}\in\mathcal{D}} p(\tilde{O}_i^{\text{ex}}|h_{i-1}^{\text{delim}},\theta^*)p(h_{i-1}^{\text{delim}}|\tilde{O}_{1:i-1}^{\text{ex}},\theta^*)} \right].$$

Since we have already bounded the second term, we now focus on the first term. The denominator of this term involves:

$$\sum_{h_{\text{query}}^{\text{start}}} p(x_{\text{query}}|h_{\text{query}}^{\text{start}},\theta^*)p(h_{\text{query}}^{\text{start}}|\tilde{S}_n,\theta^*).$$

To ensure the entire expression is bounded, it suffices to lower bound each conditional likelihood term $p(x_{\text{query}}|h_{\text{query}}^{\text{start}},\theta^*)$. This is guaranteed by the following result (adapted from Proposition 2 in (Xie et al., 2022)):

**Proposition 1** ((Xie et al., 2022))**.** *The probability of an example is lower bounded for $\theta^*$: there is some $c_7 > 0$ such that $p(O_i|h_i^{start}, h_{j,l}, \theta^*) > c_7$ for all $i$ and future hidden states $h_{j,l}$, for any $l$ and $j > i$.*

This ensures that $p(x_{\text{query}}|h_{\text{query}}^{\text{start}},\theta^*)$ is uniformly lower bounded, and therefore the full denominator in the first log term is also bounded below by a constant depending on $c_7$. Applying this result along with earlier bounds, we obtain:

$$\tilde{r}_n(\theta) \leq \frac{1}{n} \left[ -\log c_7 + 2n\log\frac{c_2}{c_1} + n\log\frac{c_4}{c_3} + \sum_{i=1}^{n} \log\frac{p(\tilde{O}_i|\theta)}{p(\tilde{O}_i|\theta^*)} \right] = -\sum_{i=1}^{n} \log\frac{p(\tilde{O}_i|\theta^*)}{p(\tilde{O}_i|\theta)} + C^{\text{delim}} + \frac{1}{n}\log\frac{1}{c_7},$$

where we set $C^{\text{delim}} = 2\log\frac{c_2}{c_1} + \log\frac{c_4}{c_3}$. This completes the proof. $\qquad\square$

### C.6. Proof of Theorem 2

Here, we restate Theorem 2 and provide the proof strategy in this section.

**Theorem.** *Suppose Assumptions 1,2,3, and 4 hold. Then, we have* $\lim_{n\to\infty} \exp\left(n \cdot \tilde{r}_n(\theta)\right) = 0$ *with* $(1-\gamma)(1-\gamma'), \gamma, \gamma' \in (0,1)$ *probability, if* $\theta^*$ *is the underlying concept of the prompt and satisfies the following condition for any other competing concept* $\theta \in \Theta$:

$$\mathbb{E}_O\left[\log \frac{p(O|\theta^*)}{p(O|\theta)}\right] > \left(\frac{2T}{\log 2}\log|\mathcal{V}| + D_1 + D_2\right)TG(\sigma) + \left(e^{D_1}\log 2\right)T^2 G(\sigma)^2 + 2h(TG(\sigma))$$

$$+ \sum_{j=1}^{T} c_j(\theta^*)\sqrt{\frac{1}{2n}\log\frac{2}{\gamma}} + \sum_{j=1}^{T} c_j(\theta)\sqrt{\frac{1}{2n}\log\frac{2}{\gamma'}} + C^{delim} + 2C^{start},$$

*where* $C^{start} := \log\frac{1}{c_8}$, $D_1 := \min_{\boldsymbol{v}\in\mathcal{V}:p_{prompt}(O=\boldsymbol{v})>0}|\log p_{prompt}(O=\boldsymbol{v})|$, $D_2 = \min_{\boldsymbol{v}\in\mathcal{V}:p(O=\boldsymbol{v}|\theta)>0}|\log p(O=\boldsymbol{v}|\theta)|$, $h(\cdot)$ *is a binary entropy function and* $c_j(\theta^*) = \left|\min_{v_j\in\{v\in\mathcal{V}|p(\tilde{o}_{ij}=v|\theta^*)>0\}}\log p(\tilde{o}_{ij}=v_j|\tilde{o}_{1:j-1},\theta^*)\right|$ *and* $c_j(\theta) = \left|\min_{v_j\in\{v\in\mathcal{V}|p(\tilde{o}_{ij}=v|\theta)>0\}}\log p(\tilde{o}_{ij}=v_j|\tilde{o}_{1:j-1},\theta)\right|$.

*Proof.* To derive the condition of the RHS of (8) being negative, we focus on the first term of RHS of(8). The proof strategy involves reformulating this term to isolate the two additional terms (i) noise error and (ii) estimation error. Specifically, we reformulate the term as follows:

$$-\frac{1}{n}\sum_{i=1}^{n}\log\frac{p(\tilde{O}_i|\theta^*)}{p(\tilde{O}_i|\theta)}$$

$$= -\frac{1}{n}\sum_{i=1}^{n}\log\frac{p(\tilde{O}_i|\theta^*)}{p(\tilde{O}_i|\theta)} - \mathbb{E}_O\left[\log\frac{p(O|\theta^*)}{p(O|\theta)}\right] - \mathbb{E}_{\tilde{O}}\left[\log\frac{p(\tilde{O}|\theta^*)}{p(\tilde{O}|\theta)}\right] + \mathbb{E}_O\left[\log\frac{p(O|\theta^*)}{p(O|\theta)}\right] + \mathbb{E}_{\tilde{O}}\left[\log\frac{p(\tilde{O}|\theta^*)}{p(\tilde{O}|\theta)}\right]$$

$$= -\mathbb{E}_O\left[\log\frac{p(O|\theta^*)}{p(O|\theta)}\right] + \left(\mathbb{E}_O\left[\log p(O|\theta^*)\right] - \mathbb{E}_{\tilde{O}}\left[\log p(\tilde{O}|\theta^*)\right]\right) + \left(\mathbb{E}_{\tilde{O}}\left[\log p(\tilde{O}|\theta)\right] - \mathbb{E}_O\left[\log p(O|\theta)\right]\right)$$

$$+ \left(-\frac{1}{n}\sum_{i=1}^{n}\log p(\tilde{O}_i|\theta^*) + \mathbb{E}_{\tilde{O}}\left[\log p(\tilde{O}|\theta^*)\right]\right) + \left(\frac{1}{n}\sum_{i=1}^{n}\log p(\tilde{O}_i|\theta) - \mathbb{E}_{\tilde{O}}\left[\log p(\tilde{O}|\theta)\right]\right)$$

$$\leq -\mathbb{E}_O\left[\log\frac{p(O|\theta^*)}{p(O|\theta)}\right] + \underbrace{\left|\mathbb{E}_O\left[\log p(O|\theta^*)\right] - \mathbb{E}_{\tilde{O}}\left[\log p(\tilde{O}|\theta^*)\right]\right| + \left|\mathbb{E}_{\tilde{O}}\left[\log p(\tilde{O}|\theta)\right] - \mathbb{E}_O\left[\log p(O|\theta)\right]\right|}_{\text{(i) Noise error}}$$

$$+ \underbrace{\left|-\frac{1}{n}\sum_{i=1}^{n}\log p(\tilde{O}_i|\theta^*) + \mathbb{E}_{\tilde{O}}\left[\log p(\tilde{O}|\theta^*)\right] + \frac{1}{n}\sum_{i=1}^{n}\log p(\tilde{O}_i|\theta) - \mathbb{E}_{\tilde{O}}\left[\log p(\tilde{O}|\theta)\right]\right|}_{\text{(ii)Estimation error}}.$$

Here, (i) noise error quantifies the effect of noise addition to the prompt distribution by measuring the divergence between noisy prompt distribution and ground-truth prompt distribution. Meanwhile, (ii) estimation error arises from estimating the expectation of log-likelihood of the noisy demonstrations under $\theta^*, \theta$ with a finite sample mean of those.

To derive a bound for (i), we proceed as follows. First, we express the error in terms of the total variation (TV) distance between the noisy and original prompt distributions (see Lemma 7). Next, we decompose the TV distance between the demonstration distributions over the $\mathcal{V}^T$ into the TV distances of individual token distributions over $\mathcal{V}$ (see Lemma 8). Finally, we explicitly isolate the noise-dependent term to quantify the effect of noise on the TV distance between the token emission probability distributions, as defined in (14c) and (6) (see Lemma 9). Specifically, combining Lemma 8 and Lemma 9 gives:

$$d_{\text{TV}}\left(\tilde{p}_{\text{prompt}|\mathcal{V}^T}, p_{\text{prompt}|\mathcal{V}^T}\right) \leq \sum_{h_1,\ldots,h_T} \tilde{p}_{\text{prompt}}(h_1,\ldots,h_T)\sum_{j=1}^{T}G(\sigma) = TG(\sigma),$$

where $G(\sigma) \in [0,1]$ is continuous function of $\sigma$, satisfying $\lim_{\sigma\to 0}G(\sigma) = 0$. The behavior of $G(\sigma)$ naturally depends on the next-token probability distribution of the involved LLM, as it captures how noise perturbs the model's next-token

probabilities (see (18)). Nonetheless, we emphasize that its derivation follows from a tight and general analysis that does not assume any specific form of this distribution, ensuring broad applicability across LLM architectures and tasks of the derived bound.

By substituting the obtained bound on the total variation into Lemma 7, then we have the following bound:

$$\text{(i) Noise error} \leq \left( \frac{2T}{\log 2} \log |\mathcal{V}| + D_1 + D_2 \right) TG(\sigma) + \left( e^{D_1} \log 2 \right) T^2 G(\sigma)^2 + 2h(TG(\sigma)) + 2C^{\text{start}},$$

Taking the limit as $\sigma \to 0$ results in the derived bound converging to $2C^{\text{start}}$, which addresses the mismatch between the start distribution of the hidden state and is a negligible term in practice.

To bound (ii), we apply a concentration inequality for random variables, which leads to a high-probability bound (see Lemma 10). Specifically, we have at least $(1 - \gamma)(1 - \gamma')$ probability:

$$\text{(ii) Estimation error} \leq \sum_{j=1}^{T} c_j(\theta^*) \sqrt{\frac{1}{2n} \log \frac{2}{\gamma}} + \sum_{j=1}^{T} c_j(\theta) \sqrt{\frac{1}{2n} \log \frac{2}{\gamma'}}.$$

Combining these bounds and Lemma 1 yield Theorem 2. $\qquad\square$

### C.7. Lemmas to bound the noise error

In this section, we provide the proof to derive the bounds on the noise error.

**Lemma 7** (Noise error). *Suppose Assumption 4 holds. Given the HMM-based prompt formulations* (14c), (14d), (15a) *and* (15b)*, we have the following:*

$$\left| \mathbb{E}_O \left[ \log p(O|\theta^*) \right] - \mathbb{E}_{\tilde{O}} \left[ \log p(\tilde{O}|\theta^*) \right] \right| + \left| \mathbb{E}_{\tilde{O}} \left[ \log p(\tilde{O}|\theta) \right] - \mathbb{E}_O \left[ \log p(O|\theta) \right] \right|$$

$$\leq \left( \frac{2T}{\log 2} \log |\mathcal{V}| + D_1 + D_2 \right) d_{TV} \left( \tilde{p}_{prompt|\mathcal{V}^T}, p_{prompt|\mathcal{V}^T} \right) + \left( e^{D_1} \log 2 \right) d_{TV} \left( \tilde{p}_{prompt|\mathcal{V}^T}, p_{prompt|\mathcal{V}^T} \right)^2$$

$$+ 2h(d_{TV} \left( \tilde{p}_{prompt|\mathcal{V}^T}, p_{prompt|\mathcal{V}^T} \right)) + 2C^{start},$$

*where $C^{start} := \log \frac{1}{c_8}$, $D_1 := \min_{v \in \mathcal{V}: p_{prompt}(O=v)>0} |\log p_{prompt}(O = v)|$, $D_2 = \min_{v \in \mathcal{V}: p(O=v|\theta)>0} |\log p(O = v|\theta)|$ and $h(\cdot)$ is a binary entropy function.*

*Proof.* We transform the first term in the LHS of Lemma 7 as follows:

$$\left| \mathbb{E}_{\tilde{O} \sim \tilde{p}_{prompt}} \left[ \log p(\tilde{O}|\theta^*) \right] - \mathbb{E}_{O \sim p_{prompt}} \left[ \log p(O|\theta^*) \right] \right|$$

$$\leq \underbrace{\left| \mathbb{E}_{\tilde{O} \sim \tilde{p}_{prompt}} \left[ \log \tilde{p}_{prompt}(\tilde{O}) \right] - \mathbb{E}_{O \sim p_{prompt}} \left[ \log p_{prompt}(O) \right] \right|}_{:=T_1} + \underbrace{\left| D_{KL} \left( (p_{prompt|\mathcal{V}^T} \| p(O|\theta^*)) \right) - D_{KL} \left( (\tilde{p}_{prompt|\mathcal{V}^T} \| p(O|\theta^*)) \right) \right|}_{:=T_2}.$$

Applying Lemma 3 to $T_1$ gives:

$$T_1 \leq \frac{T \log |\mathcal{V}|}{\log 2} d_{TV} \left( \tilde{p}_{prompt|\mathcal{V}^T}, p_{prompt|\mathcal{V}^T} \right) + h(d_{TV} \left( \tilde{p}_{prompt|\mathcal{V}^T}, p_{prompt|\mathcal{V}^T} \right)).$$

Next, we have the bound on $T_2$ as follows:

$$T_2 \leq \left| D_{KL} \left( p_{prompt|\mathcal{V}^T} \| p(O|\theta^*) \right) \right| + \left| D_{KL} \left( \tilde{p}_{prompt|\mathcal{V}^T} \| p(O|\theta^*) \right) \right| \leq 2 \log \frac{1}{c_8}.$$

This holds because $p_{prompt}$ and $p(\cdot|\theta^*)$ only differs the start distribution of $h_1$ and similarly $p_{prompt}$ and $p(\cdot|\theta^*)$ also only differs the start distribution of $h_1$. See its derivation as shown in (16).

Similarly by isolating out the difference between entropy of two distributions, we rearrange the second term in the LHS of Lemma 7 as follows:

$$\left| \mathbb{E}_{\tilde{O}} \left[ \log p(\tilde{O}|\theta) \right] - \mathbb{E}_O \left[ \log p(O|\theta) \right] \right| \leq T_1 + \underbrace{\left| D_{KL} \left( (p_{prompt|\mathcal{V}^T} \| \| p(\cdot|\theta)) \right) \right| - D_{KL} \left( (\tilde{p}_{prompt|\mathcal{V}^T} \| p(\cdot|\theta)) \right)}_{:=T_3}.$$

Now, we bound $T_3$:

$$T_3 = \left| \sum_{v \in \mathcal{V}^T} \left( p_{prompt}(O = v) \log \frac{p_{prompt}(O = v)}{p(O = v|\theta)} - \tilde{p}_{prompt}(\tilde{O} = v) \log \frac{\tilde{p}_{prompt}(\tilde{O} = v)}{p(\tilde{O} = v|\theta)} \right) \right|$$

$$= \left| \sum_{v \in \mathcal{V}^T} (p_{prompt}(O = v) - \tilde{p}_{prompt}(O = v)) \log \frac{p_{prompt}(O = v)}{p(O = v|\theta)} - \tilde{p}_{prompt}(\tilde{O} = v) \log \frac{\tilde{p}_{prompt}(\tilde{O} = v)}{p_{prompt}(\tilde{O} = v|\theta)} \right|$$

$$\leq \sum_{v \in \mathcal{V}^T} \left| (p_{prompt}(O = v) - \tilde{p}_{prompt}(O = v)) \log \frac{p_{prompt}(O = v)}{p(O = v|\theta)} \right| + \left| D_{KL} \left( \tilde{p}_{prompt|\mathcal{V}^T} \| p_{prompt|\mathcal{V}^T} \right) \right|.$$

Note that fixing $v$ yields $p(O = v|\theta) = p(\tilde{O} = v|\theta)$. The inequality follows from the triangle inequality.

To bound the log-ratio term, we have:

$$\left| \log \frac{p_{prompt}(O = v)}{p(O = v|\theta)} \right| \leq \max \{ \min_{v \in \mathcal{V}: p_{prompt}(O=v)>0} |\log p_{prompt}(O = v)|, \min_{v \in \mathcal{V}: p(O=v|\theta)>0} |\log p(O = v|\theta)| \} \leq D_1 + D_2,$$

where we define $D_1 := \min_{\boldsymbol{v} \in \mathcal{V}: p_{\text{prompt}}(O=\boldsymbol{v})>0} |\log p_{\text{prompt}}(O = \boldsymbol{v})|$ and $D_2 = \min_{\boldsymbol{v} \in \mathcal{V}: p(O=\boldsymbol{v}|\theta)>0} |\log p(O = \boldsymbol{v}|\theta)|$. This gives the following bound:

$$
\sum_{\boldsymbol{v} \in \mathcal{V}^T} \left| (p_{\text{prompt}}(O = \boldsymbol{v}) - \tilde{p}_{\text{prompt}}(O = \boldsymbol{v})) \log \frac{p_{\text{prompt}}(O = \boldsymbol{v})}{p(O = \boldsymbol{v}|\theta)} \right| \leq (D_1 + D_2) \sum_{\boldsymbol{v} \in \mathcal{V}^T} |(p_{\text{prompt}}(O = \boldsymbol{v}) - \tilde{p}_{\text{prompt}}(O = \boldsymbol{v}))|
$$

$$
= (D_1 + D_2) d_{\text{TV}} \left( \tilde{p}_{\text{prompt}|\mathcal{V}^T}, p_{\text{prompt}|\mathcal{V}^T} \right).
$$

To bound the KL term, applying Lemma 5 gives:

$$
\left| D_{\text{KL}} \left( \tilde{p}_{\text{prompt}|\mathcal{V}^T} \| p_{\text{prompt}|\mathcal{V}^T} \right) \right| = D_{\text{KL}} \left( \tilde{p}_{\text{prompt}|\mathcal{V}^T} \| p_{\text{prompt}|\mathcal{V}^T} \right) \leq \frac{\log 2}{e^{-D_1}} d_{\text{TV}} \left( \tilde{p}_{\text{prompt}|\mathcal{V}^T}, p_{\text{prompt}|\mathcal{V}^T} \right)^2,
$$

where $e^{-D_1} = \min_{\boldsymbol{v} \in \mathcal{V}: p_{\text{prompt}}(O=\boldsymbol{v})>0} p_{\text{prompt}}(O = \boldsymbol{v})$.

By combining the bounds on $T_1, T_2$ and $T_3$, we complete the proof. $\qquad \square$

**Lemma 8.** *Given the HMM-based prompt formulations* (14c), (14d), (15a) *and* (15b), *we have the following:*

$$d_{TV}(\tilde{p}_{prompt|\mathcal{V}^T}, p_{prompt|\mathcal{V}^T}) \leq \sum_{h_1,\ldots,h_T} \tilde{p}_{prompt}(h_1,\ldots,h_T) \sum_{j=1}^{T} d_{TV}\left(\tilde{p}_{prompt}(o_j|h_j), \tilde{p}_{prompt}(o_j|h_j)\right).$$

*Proof.* On the basis of the HMM-based prompt formulation given by (14c), (14d), (15a) and (15b), we rewrite as follows:

$$d_{TV}(\tilde{p}_{prompt|\mathcal{V}^T}, p_{prompt|\mathcal{V}^T}) = \sum_{\boldsymbol{v} \in \mathcal{V}^T} |\tilde{p}_{prompt}(O = \boldsymbol{v}) - p_{prompt}(O = \boldsymbol{v})|$$

$$= \sum_{\boldsymbol{v} \in \mathcal{V}^T} \left| \sum_{h_1,\ldots,h_T} \tilde{p}_{prompt}(h_1,\ldots,h_T) \prod_{j=1}^{T} (\tilde{p}_{prompt}(o_j = v_j|h_j) - p_{prompt}(o_j = v_j|h_j)) \right|.$$

Using the triangle inequality, we bound the absolute difference:

$$\sum_{\boldsymbol{v} \in \mathcal{V}^T} \sum_{h_1,\ldots,h_T} \tilde{p}_{prompt}(h_1,\ldots,h_T) \left| \prod_{j=1}^{T} (\tilde{p}_{prompt}(o_j = v_j|h_j) - p_{prompt}(o_j = v_j|h_j)) \right|$$

$$\leq \sum_{\boldsymbol{v} \in \mathcal{V}^T} \sum_{h_1,\ldots,h_T} \tilde{p}_{prompt}(h_1,\ldots,h_T) \sum_{j=1}^{T} |\tilde{p}_{prompt}(o_j = v_j|h_j) - p_{prompt}(o_j = v_j|h_j)|.$$

Here, we applied the general triangle inequality for product differences:

$$\left| \prod_{j=1}^{T} a_j - \prod_{j=1}^{T} b_j \right| \leq \sum_{j=1}^{T} |a_j - b_j| \left( \prod_{j=1}^{t-1} a_j \right) \left( \prod_{j=T}^{t+1} b_j \right),$$

where $a_j = \tilde{p}_{prompt}(o_j = v_j|h_j)$ and $b_j = p_{prompt}(o_j = v_j|h_j)$, noting that both probabilities lie in $[0, 1]$.

Rearranging the summation order, we obtain:

$$d_{TV}(\tilde{p}_{prompt|\mathcal{V}^T}, p_{prompt|\mathcal{V}^T}) = \sum_{h_1,\ldots,h_T} \tilde{p}_{prompt}(h_1,\ldots,h_T) \sum_{j=1}^{T} \sum_{v_j \in \mathcal{V}} |\tilde{p}_{prompt}(o_j = v_j|h_j) - p_{prompt}(o_j = v_j|h_j)|$$

$$= \sum_{h_1,\ldots,h_T} \tilde{p}_{prompt}(h_1,\ldots,h_T) \sum_{j=1}^{T} d_{TV}\left(\tilde{p}_{prompt}(o_j|h_j), p_{prompt}(o_j|h_j)\right).$$

This establishes the desired bound. □

**Lemma 9.** *Given the HMM-based prompt formulations* (14c), (14d), (15a) *and* (15b), *we have the following:*

$$d_{TV}\left(\tilde{p}_{prompt}(o_j = v_j|h_j)\right), p_{prompt}(o_j = v_j|h_j)) \leq G(\sigma),$$

*where* $G(\sigma) \in [0, 1]$ *is continuous function of* $\sigma$ *and* $\lim_{\sigma \to 0} G(\sigma) = 0$.

*Proof.* To establish bound, we begin by applying Lemma 4:

$$d_{TV}\left(\tilde{p}_{prompt|\mathcal{V}}, p_{prompt|\mathcal{V}}\right) = d_{TV}\left(p_{prompt|\mathcal{V}}, \tilde{p}_{prompt|\mathcal{V}}\right) \leq \sqrt{1 - \exp\left(-D_{KL}\left(p_{prompt|\mathcal{V}} \| \tilde{p}_{prompt|\mathcal{V}}\right)\right)}.$$

Next, we express the KL-term using definition the noise addition as shown in (15a) and (15b). Rewriting the KL-term gives:

$$D_{KL}\left(p_{prompt|\mathcal{V}} \| \mathbb{E}_{\boldsymbol{\varsigma}}\left[\hat{p}_{prompt|\mathcal{V}}\right]\right) = \sum_{v \in \mathcal{V}} \left( p_v \log p_v - p_v \log \mathbb{E}_{\boldsymbol{\varsigma}}\left[\frac{p_v + Z_v}{1 + \sum_{u \in \mathcal{V}} Z_u}\right] \right), \tag{17}$$

where we introduce $p_v = p_{\text{prompt}}(o = v | h)$ and $Z_v = \max\{0, p_v + \zeta_v\} - p_v, \boldsymbol{\zeta} = (\zeta_v)_{v \in \mathcal{V}} \sim \mathcal{N}(0, \sigma^2 I_{|\mathcal{V}|})$ for notational simplicity.

Now, we aim to bound the expectation term of the RHS of (17). By using the Jensen inequality, which holds due to the convexity of $f(x) = -\log x$, then we have:

$$-\log \mathbb{E}_{\boldsymbol{\zeta}}\left[\frac{p_v + Z_v}{1 + \sum_{u \in \mathcal{V}} Z_u}\right] \leq \mathbb{E}_{\boldsymbol{\zeta}}\left[-\log \frac{p_v + Z_v}{1 + \sum_{u \in \mathcal{V}} Z_u}\right] \leq \underbrace{\log \mathbb{E}_{\boldsymbol{\zeta}}\left[\left(1 + \sum_{u \in \mathcal{V}} Z_u\right)\right] - \mathbb{E}_{\boldsymbol{\zeta}}\left[\log\left(p_v + Z_v\right)\right]}_{:=F(\sigma, p_v)}.$$

By defining $F(\sigma, p_v)$, we obtain an explicit dependency on $\sigma$ of the bound. Since $Z_v + p_v = \max\{0, p_v + \zeta_v\}$ follows the rectified Gaussian distribution, thus applying Lemma 2 yields:

$$\mathbb{E}_{\boldsymbol{\zeta}}\left[\left(1 + \sum_{u \in \mathcal{V}} Z_u\right)\right] = \log\left(1 + \sum_{u \in \mathcal{V}} -p_u \Phi(-\frac{p_u}{\sigma}) + \sigma\phi\left(-\frac{p_u}{\sigma}\right)\right).$$

Thus, we obtain the following upper bound:

$$d_{\text{TV}}\left(\tilde{p}_{\text{prompt}|\mathcal{V}}, p_{\text{prompt}|\mathcal{V}}\right) \leq \underbrace{\sqrt{1 - \exp\left(-\left(\sum_{v \in \mathcal{V}} p_v \log p_v + p_v F(\sigma, p_v)\right)\right)}}_{:=G(\sigma)}. \tag{18}$$

To validate this bound, we analyze the asymptotic behavior of $F(\sigma, p_v)$ as $\sigma \to 0$, rather than deriving its closed formula. Specifically, we evaluate:

$$\lim_{\sigma \to 0} F(\sigma, p_v) = 0 - \lim_{\sigma \to 0} \mathbb{E}_{\boldsymbol{\zeta}}\left[\log\left(p_v + Z_v\right)\right] = -\int_{-\infty}^{\infty} \log\max(0, \zeta_v + p_v) \lim_{\sigma \to 0} \frac{1}{\sqrt{2\pi}\sigma} \exp\left(-\frac{\zeta_v^2}{\sigma^2}\right) d\zeta_v$$

$$= -\int_{\infty}^{\infty} \log\max\left(0, \zeta_v\right)\delta(\zeta_v - p_v) d\zeta_v = \log p_v,$$

where $\delta(\cdot)$ represents the Dirac delta function. This result implies:

$$\lim_{\sigma \to 0} d_{\text{TV}}\left(\tilde{p}_{\text{prompt}|\mathcal{V}}, p_{\text{prompt}|\mathcal{V}}\right) = 0.$$

This asymptotic behavior is consistent with the definition of $\tilde{p}_{\text{prompt}}$ and $\tilde{p}_{\text{prompt}}$, confirming that $\tilde{p}_{\text{prompt}}$ converges to $p_{\text{prompt}}$ when $\sigma \to 0$ (i.e. noise-free setting) from its definition.

Consequently, we have established a bound in the form:

$$d_{\text{TV}}\left(\tilde{p}_{\text{prompt}|\mathcal{V}}, p_{\text{prompt}|\mathcal{V}}\right) \leq G(\sigma),$$

where $G(\sigma)$ is a continuous function that satisfies:

$$G(\sigma) \in [0, 1], \quad \lim_{\sigma \to 0} G(\sigma) = 0.$$

This completes the proof. $\qquad\square$

## C.8. Lemmas to bound the estimation error

**Lemma 10** (Estimation Error with Finite Demonstrations). *Given the HMM-based prompt formulation in* (14c), (14d), (15a) *and* (15b). *Then, we have the following at least* $(1 - \gamma)(1 - \gamma')$ *probability for any* $\theta \in \Theta$:

$$\left| \mathbb{E}_{\tilde{O}}\left[\log p(\tilde{O}|\theta)\right] - \frac{1}{n}\sum_{i=1}^{n}\log p(\tilde{O}_i|\theta) + \frac{1}{n}\sum_{i=1}^{n}\log p(\tilde{O}_i|\theta) - \mathbb{E}_{\tilde{O}}\left[\log p(\tilde{O}|\theta)\right] \right|$$

$$\leq \sum_{j=1}^{T} c_j(\theta^*)\sqrt{\frac{1}{2n}\log\frac{2}{\gamma}} + \sum_{j=1}^{T} c_j(\theta)\sqrt{\frac{1}{2n}\log\frac{2}{\gamma'}},$$

*where*

$$c_j(\theta^*) = \left| \min_{v_j \in \{v \in \mathcal{V} | p(\tilde{o}_{ij}=v|\theta^*)>0\}} \log p(\tilde{o}_{ij} = v_j|\tilde{o}_{1:j-1}, \theta^*) \right|,$$

$$c_j(\theta) = \left| \min_{v_j \in \{v \in \mathcal{V} | p(\tilde{o}_{ij}=v|\theta)>0\}} \log p(\tilde{o}_{ij} = v_j|\tilde{o}_{1:j-1}, \theta) \right|.$$

*Proof.* First, we aim to bound the following:

$$\left| -\frac{1}{n}\sum_{i=1}^{n}\log p(\tilde{O}_i|\theta^*) + \mathbb{E}_{\tilde{O}}\left[\log p(\tilde{O}|\theta^*)\right] \right|.$$

Since $\tilde{O}_1, \ldots, \tilde{O}_n$ are independent, thus applying the Lemma 6 gives the following bound:

$$\Pr\left[ \left| -\frac{1}{n}\sum_{i=1}^{n}\log p(\tilde{O}_i|\theta^*) + \mathbb{E}_{\tilde{O}}\left[\log p(\tilde{O}|\theta^*)\right] \right| \geq t \right] \leq 2\exp\left(-\frac{2nt^2}{(b-a)^2}\right),$$

where $b, a$ is upper and lower bound of $\log p(\tilde{O}_i|\theta^*)$. Specifically, to derive these bound we consider the following:

$$|b - a| \leq \left| \min_{\boldsymbol{v} \in \{\boldsymbol{v} \in \mathcal{V}^T | p(\tilde{O}_i=\boldsymbol{v}|\theta^*)>0\}} \log p(\tilde{O}_i = \boldsymbol{v}|\theta^*) \right| = \left| \min_{\boldsymbol{v} \in \{\boldsymbol{v} \in \mathcal{V}^T | p(\tilde{O}_i=\boldsymbol{v}|\theta^*)>0\}} \sum_{j=1}^{T}\log p(\tilde{o}_{ij} = v_j|\tilde{o}_{1:j-1}, \theta^*) \right|$$

$$\leq \sum_{j=1}^{T} \underbrace{\left| \min_{v_j \in \{v \in \mathcal{V} | p(\tilde{o}_{ij}=v|\theta^*)>0\}} \log p(\tilde{o}_{ij} = v_j|\tilde{o}_{1:j-1}, \theta^*) \right|}_{:=c_j(\theta^*)}.$$

By rearranging for probability bound $1 - \gamma$:

$$\left| -\frac{1}{n}\sum_{i=1}^{n}\log p(\tilde{O}_i|\theta^*) + \mathbb{E}_{\tilde{O}}\left[\log p(\tilde{O}|\theta^*)\right] \right| \leq \sum_{j=1}^{T} c_j(\theta^*)\sqrt{\frac{1}{2n}\log\frac{2}{\gamma}}.$$

Similarly, we have the following bound for $\theta \in \Theta$ at least $1 - \gamma'$ probability:

$$\left| -\frac{1}{n}\sum_{i=1}^{n}\log p(\tilde{O}_i|\theta) + \mathbb{E}_{\tilde{O}}\left[\log p(\tilde{O}|\theta)\right] \right| \leq \sum_{j=1}^{T} c_j(\theta)\sqrt{\frac{1}{2n}\log\frac{2}{\gamma'}}.$$

where $c_j(\theta) = \left|\min_{v_j \in \{v \in \mathcal{V} | p(\tilde{o}_{ij}=v|\theta)>0\}} \log p(\tilde{o}_{ij} = v_j|\tilde{o}_{1:j-1}, \theta)\right|$. This completes the proof.

$\square$

# D. Privacy analysis

This section presents the privacy analysis ensuring the proposed Algorithm 1 satisfies $(\varepsilon, \delta)$-DP. In Appendix D.1, we introduce key notions and relevant theorems from the literature that form the basis of our analysis. Appendix D.2 provides a detailed proof that Algorithm 1 satisfies $(\varepsilon, \delta)$-DP. Finally, we describe the numerical calculation of noise variance $\sigma^2$ to ensure $(\varepsilon, \delta)$-DP in Appendix D.3, based on the method proposed in (Gopi et al., 2021).

## D.1. Preliminaries

**Basics of differential privacy and its composition** Differential Privacy (DP), introduced by (Dwork et al., 2006a), provides rigorous privacy guarantees for datasets used in statistical queries. Formally, DP is defined as follows:

**Definition 2** (($\varepsilon, \delta$)-Differential Privacy (DP) (Dwork et al., 2006a)). A randomized mechanism $\mathcal{M}$ is $(\varepsilon, \delta)$-differentially private if for any two neighboring datasets $D, D' \in \mathcal{D}$ that differ by at most one element, and for any subset of outputs $\mathcal{S} \subseteq \mathrm{Range}(\mathcal{M})$, the following holds:

$$\Pr\left[\mathcal{M}(D) \in \mathcal{S}\right] \leq e^\epsilon \Pr\left[\mathcal{M}(D') \in \mathcal{S}\right] + \delta. \tag{19}$$

As described in Section 2.1, $\epsilon > 0$ bounds the distinguishbility of outputs between neighboring datasets and $\delta \in (0, 1)$ represents the allowable maximum failure probability of exceeding this bound $\varepsilon$. Smaller values of $\varepsilon, \delta$ indicate lower likelihood of privacy leakage. To achieve $(\varepsilon, \delta)$-DP, a Gaussian mechanism is widely used. The Gaussian mechanism is formally defined as follows:

**Definition 3** (Gaussian Mechanism). Let $f : \mathcal{X} \to \mathbb{R}^d$. The Gaussian mechanism is defined as follows:

$$\mathcal{M}(D) = f(D) + \mathcal{N}(0, \sigma^2 I_d),$$

where $I_d$ is the identity matrix.

The privacy guarantees of the Gaussian mechanism depend on the noise variance $\sigma$. To determine the noise variance, the sensitivity of $f$ must be considered, which quantifies how much the outputs of $f$ can change between neighboring datasets. For the Gaussian mechanism, the sensitivity is measured by using the $\ell_2$-norm, also known as $\ell_2$-sensitivity, defined as follows:

$$\Delta = \max_{D, D'} \|f(D) - f(D')\|_2.$$

By using $\ell_2$-sensitivity, the noise variance $\sigma^2$ is calibrated to ensure $(\varepsilon, \delta)$-DP. This relationship is formalized as follows:

**Theorem 4** (Noise variance of Gaussian Mechanism (Dwork & Roth, 2014)). *Let $f : \mathcal{X} \to \mathbb{R}^d$ and $\mathcal{M}$ be the Gaussian mechanism adding noise to the output of $f$. Further, let $\Delta$ be $\ell_2$-sensitivity of $f$. For any $\varepsilon \in (0, 1), \delta \in (0, 1)$, the Gaussian mechanism with $\sigma = \Delta \sqrt{2 \log 1.25/\delta}/\varepsilon$ is $(\varepsilon, \delta)$-DP.*

Thus, the Gaussian mechanism ensures privacy by using the tuned noise variance. When the Gaussian mechanism is applied to randomly sampled subsets of a dataset, the privacy guarantees improve due to sub-sampling amplification. Sub-sampling amplification formalizes the observation that accessing only a subset of the datasets reduces the risk of privacy leakage:

**Theorem 5** (Sub-sampling amplification (Balle et al., 2018)). *If $\mathcal{M}$ is $(\varepsilon, \delta)$-DP, then the sub-sampled mechanism with sampling rate $q$ obeys $(\varepsilon', \delta')$-DP with privacy parameters:*

$$\varepsilon' = \log\left(1 + q(e^\varepsilon - 1)\right), \quad \delta' = q\delta.$$

While the Gaussian mechanism or the sub-sampled Gaussian mechanism ensures privacy for a single iteration, repeated or sequential applications to the same dataset increase the risk of privacy leakage as small leaks accumulate. To quantify the overall DP guarantees across multiple iterations, the privacy parameters $\varepsilon, \delta$ of each step must be combined through composition. Naive composition (Dwork & Lei, 2009), which adds $\varepsilon$ and $\delta$ linearly, provides a valid but overly conservative estimate of cumulative privacy guarantees. This overestimation leads to unnecessarily high noise variance, degrading the accuracy of DP mechanisms. These limitations have motivated advanced methods, including advanced composition theorems (Dwork et al., 2010), Renyi DP (Mironov, 2017), and other refined analyses (Dong et al., 2022), to achieve tighter bounds on cumulative privacy guarantees.

Among these approaches, we focus on numerical composition (Gopi et al., 2021), which leverages privacy curves and Privacy Loss Random Variables (PRVs) for precise and efficient analysis. Numerical composition is particularly effective in adaptive settings, where the output of one mechanism influences the input to subsequent mechanisms. For example, in generating DP synthetic token sequences (Tang et al., 2024), the Gaussian mechanism is applied iteratively, with each token generation relying on previously generated tokens. Numerical composition provides a robust framework for analyzing such scenarios while minimizing overestimation.

**Privacy curve and numerical composition using PRVs** A privacy curve provides a functional representation of a DP mechanism's guarantees by relating the distinguishability parameter $\varepsilon$ to the failure probability $\delta$. Formally, the privacy curve is defined as follows:

**Definition 4** (Privacy curve (Gopi et al., 2021)). Given two random variables $X, Y$ supported on some set $\Omega$, define $\delta(X\|Y) : \mathbb{R} \to [0, 1]$ as:
$$\delta(X\|Y)(\varepsilon) = \sup_{S \subset \Omega} \Pr\left[Y \in S\right] - e^{\varepsilon} \Pr\left[X \in S\right].$$

For a DP mechanism $\mathcal{M}$, the privacy curve ensures that $\mathcal{M}$ is $(\varepsilon, \delta)$-DP iff $\delta(\mathcal{M}(D)\|\mathcal{M}(D'))(\varepsilon) \leq \delta$ for all neighboring datasets $D, D'$. By composing the privacy curves corresponding to individual DP mechanisms, we can determine their cumulative privacy guarantees when multiple DP mechanisms are applied iteratively or adaptively. The composition of privacy curves is formally defined as follows:

**Definition 5** (Composition of privacy curves (Dong et al., 2022; Gopi et al., 2021)). Let $\delta_1 \equiv \delta(X_1\|Y_1)$ and $\delta_2 \equiv \delta(X_2\|Y_2)$ be any two privacy curves. The composition of the privacy curves, denoted by $\delta_1 \otimes \delta_2$, is defined as
$$\delta_1 \otimes \delta_2 \equiv \delta((X_1, X_2)\|(Y_1, Y_2)),$$
where $X_1, X_2$ are independently sampled and $Y_1, Y_2$ are independently sampled.

This composition operation combines the privacy guarantees of individual mechanisms into a single privacy curve that describes the cumulative guarantees of the sequence of DP mechanisms.

**Theorem 6** (Composition theorem (Dong et al., 2022; Gopi et al., 2021)). *Let $\mathcal{M}_1, \mathcal{M}_2, \ldots, \mathcal{M}_k$ be DP algorithms with privacy curves given by $\delta_1, \delta_2, \ldots, \delta_k$ respectively. The privacy curve of the adaptive composition $\mathcal{M}_k \circ \mathcal{M}_{k-1} \circ \cdots \circ \mathcal{M}_1$ is given by $\delta_1 \otimes \delta_2 \otimes \cdots \otimes \delta_k$.*

This result demonstrates that privacy curves provide a unified framework for analyzing cumulative privacy guarantees, ensuring the accurate overall privacy parameters $\varepsilon, \delta$ for adaptive compositions without unnecessary overestimation or underestimation of cumulative privacy guarantees.

To efficiently compute privacy curves and their compositions, RPVs were introduced in (Gopi et al., 2021). PRVs re-parametrize privacy curves by representing them as pairs of random variables $(X, Y)$, enabling efficient evaluation and composition. The precise definition of PRVs and their formal derivation are deferred to prior work (Gopi et al., 2021). Here, we highlight key properties of PRVs:

**Uniqueness:** For any DP mechanism's privacy curve, there exists a unique pair of PRVs $(X, Y)$ such that $\delta \equiv \delta(X\|Y)$ (Theorem 3.2 in (Gopi et al., 2021)).

**Explicit formula:** The privacy curve can be directly computed using the PRVs $(X, Y)$ as $\delta(\varepsilon) = \Pr\left[Y > \varepsilon\right] - e^{\varepsilon} \Pr\left[X > \varepsilon\right]$ (Theorem 3.3 in (Gopi et al., 2021)).

**Composition:** The composition of privacy curves $\delta_1 \equiv \delta(X_1\|Y_1), \delta_2 \equiv \delta(X_2\|Y_2)$ corresponds to summing their PRVs $\delta = \delta_1 \otimes \delta_2 \equiv \delta(X_1 + X_2\|Y_1 + Y_2)$ (Theorem 3.5 in (Gopi et al., 2021)).

By leveraging these properties, the numerical composition process proceeds as follows: (1) determine the PRVs $(X_i, Y_i)$ for each DP mechanism, (2) compute the PDF of the sum of the PRVs for all mechanisms in the composition by convolving the individual PDFs, and (3) evaluate the resulting privacy curve using the explicit formula. This convolution operation transforms the computation of cumulative privacy guarantees into a scalable and precise process, making PRVs a practical and powerful tool for analyzing iterative and adaptive DP mechanisms.

**D.2. Privacy proof**

Here, we provide the proof of the following theorem.

**Theorem 7.** *Algorithm 1 is $(\hat{\varepsilon}_{total}, \hat{\delta}_{total})$-DP.*

Specifically, we aim to show that the entire procedure satisfies $(\hat{\epsilon}_{\text{total}}, \hat{\delta}_{\text{total}})$-DP, where these cumulative privacy parameters are obtained by composing per-iteration guarantees. The definition and calculation of these parameters are provided in the following proof.

*Proof.* The PTA procedure in Algorithm 1 operates iteratively at most $T$, processing the private dataset $\mathcal{D}_{\text{priv}}$ to generate DP synthetic demonstrations of length $T$. Each iteration generates a token that depends on the previously generated tokens, forming part of the DP synthetic demonstration. Consequently, the cumulative privacy guarantees of Algorithm 1 must be analyzed by using adaptive composition across $T$ iterations, on the basis of Theorem 6. This proof demonstrates that Algorithm 1 satisfies $(\hat{\varepsilon}_{\text{total}}, \hat{\delta}_{\text{total}})$-DP by composing the privacy guarantees of the sub-sampled Gaussian mechanism through numerical composition.

First, we identify the Lines in Algorithm 1 involving private datasets $\mathcal{D}_{\text{priv}}$ at $j$-th iteration. Algorithm 1 processes private dataset $\mathcal{D}_{\text{priv}}$ in the following steps:

- **Line 3:** The subroutine $\text{GenPrompt}(\mathcal{D}_{\text{priv}}, M, N, \tilde{y}, \tilde{O})$ generates private prompts $S_{\text{priv}}^{(1)}, \ldots, S_{\text{priv}}^{(M)}$ by sampling subsets of $\mathcal{D}_{\text{priv}}$. This step introduces sub-sampling amplification of the subsequent mechanism, as shown in Theorem 5.

- **Line 4–8:** Private prompts $S_{\text{priv}}^{(i)}$ are used to compute the next-token probabilities $p(o^{\text{LLM}} = v|S_{\text{priv}}^{(i)})$, which are then modified by PTA in Line 5 and 6. Finally, the modified next-token probabilities are aggregated across $M$ subsets. This modification and aggregation involves $\mathcal{D}_{\text{priv}}$ and requires a DP mechanism to ensure privacy.

To ensure that the above lines satisfy $(\hat{\varepsilon}_j, \hat{\delta}_j)$-DP, Algorithm 1 introduces the sub-sampled Gaussian mechanisms whose privacy curve $\delta_j$ accounts for

- **Line 3:** Sub-sampling amplification with sampling rate $q = \frac{MN}{|\mathcal{D}_{\text{priv}}|}$ as shown in Theorem 5.

- **Line 4–8:** The Gaussian mechanism adds the noise to the next-token probabilities modified by PTA. As shown in Line 5 and 6, PTA modifies the next-token probabilities, as follows:

$$p_v^{(i)} \leftarrow p_{\text{base}}(o^{\text{LLM}} = v) \left( \frac{p(o^{\text{LLM}} = v|S_{\text{priv}}^{(i)})}{p(o^{\text{LLM}} = v|S_{\text{pub}})} \right)^{\alpha},$$

$$p_v^{(i)} \leftarrow \frac{p_v^{(i)}}{\sum_{v' \in \mathcal{V}} p_{v'}^{(i)}}.$$

$\ell_2$-sensitivity of the above steps is at most $\sqrt{2}$, since as the token probabilities are necessarily projected onto the probability simplex. In Line 8, the obtained next-token probabilities $p_v^{(i)}$ are aggregated across $M$ subsets followed by the addition of the Gaussian noise:

$$\hat{p}_v \leftarrow \frac{1}{M} \sum_{i=1}^{M} p_v^{(i)} + \mathcal{N}(0, 2\sigma^2).$$

By combining Theorem 4 and 5, these steps satisfy $(\hat{\varepsilon}_j, \hat{\delta}_j)$-DP with the tuned noise variance $\sigma^2$. Therefore, there exist unique PRVs $(X_j, Y_j)$ such that the privacy curve of the sub-sampled Gaussian mechanisms is $\delta_j \equiv \delta(X_j, Y_j)$. Furthermore, due to the post-processing immunity of DP(Dwork et al., 2006b), subsequent operation on $\hat{p}_v$ in Line 9–11 never increases the risk of leakage of $\mathcal{D}_{\text{priv}}$. The overall $j$-th iteration satisfies $(\hat{\varepsilon}_j, \hat{\delta}_j)$-DP.

In the following discussion, we assume that the noise variance $\sigma^2$ has been appropriately determined to satisfy the per-iteration privacy guarantees $\delta_j(\hat{\varepsilon}_j) \leq \hat{\delta}_j$ given sampling rate $q$. The specific calculation of noise variance $\sigma^2$ is deferred to Appendix D.3, where it is determined through numerical composition(Gopi et al., 2021).

Since the sub-sampled Gaussian mechanism operates for $T$ iterations, the cumulative privacy guarantees are analyzed through adaptive composition. By using the composition theorem for privacy curves as shown in Theorem 6, the overall privacy guarantees are composed as:

$$\delta_{\text{total}} = \delta_1 \otimes \delta_2 \otimes \cdots \otimes \delta_T,$$

where $\delta_j \equiv \delta(X_j \| Y_j)$ represents the privacy curve of the sub-sampled Gaussian mechanism at the $j$-th iteration. This operation combines the privacy curves of individual iterations into a single curve representing the cumulative privacy guarantees.

As discussed in Appendix D.1, the cumulative privacy guarantees are efficiently computed by summing PRVs across all iterations. The composed privacy curve is given by:

$$\delta_{\text{total}} \equiv \delta\left(X_{\text{total}} \| Y_{\text{total}}\right), \quad (X_{\text{total}}, Y_{\text{total}}) = (\sum_{j=1}^{T} X_j, \sum_{j=1}^{T} Y_j).$$

Once the cumulative privacy curve $\delta_{\text{total}}$ is determined given noise variance $\sigma^2$, fixing one of the privacy parameters $\hat{\varepsilon}_{\text{total}}$ or $\hat{\delta}_{\text{total}}$ allows the other to be computed using the explicit relationship between the privacy curve and its PRVs.

In summary, at the end of $T$ iterations, the final output $\tilde{O}$ is guaranteed to satisfy $(\hat{\epsilon}_{\text{total}}, \hat{\delta}_{\text{total}})$-DP. The use of numerical composition ensures precise cumulative privacy guarantees, avoiding unnecessary overestimation or underestimation. $\square$

### D.3. Numerical calculation of $\sigma^2$

In Appendix D.2, we proved that Algorithm 1 satisfies $(\hat{\varepsilon}_{\text{total}}, \hat{\delta}_{\text{total}})$-DP. The noise variance $\sigma^2$ plays a crucial role in determining the specific values of $(\hat{\varepsilon}_{\text{total}}, \hat{\delta}_{\text{total}})$. Conversely, $\sigma^2$ can be calibrated to satisfy the pre-defined $(\hat{\varepsilon}_{\text{total}}, \hat{\delta}_{\text{total}})$. In this section, we describe the numerical calculation of $\sigma^2$, following the method proposed in (Gopi et al., 2021).

First, we present the privacy curve of the Gaussian mechanism. (Balle & Wang, 2018) show that the privacy curve of the Gaussian mechanism is given by:

$$\delta(\mathcal{N}(\Delta, \sigma^2) \| \mathcal{N}(0, \sigma^2))(\varepsilon).$$

To relate the noise variance of the Gaussian mechanism to its privacy curve, we derive the PRVs corresponding to the Gaussian mechanism, the same as (Gopi et al., 2021):

**Proposition 1** (PRVs of the Gaussian mechanism). *The PRVs $(X, Y)$ for $\delta(\mathcal{N}(\Delta, \sigma^2) \| \mathcal{N}(0, \sigma^2))$ are given by:*

$$X = \mathcal{N}\left(-\frac{\Delta^2}{2\sigma^2}, \frac{\Delta^2}{\sigma^2}\right), \quad Y = \mathcal{N}\left(\frac{\Delta^2}{2\sigma^2}, \frac{\Delta^2}{\sigma^2}\right). \tag{20}$$

The proof is nearly identical to Proposition B.1 in (Gopi et al., 2021).

*Proof.* Let $P = \mathcal{N}(\Delta, \sigma^2)$ and $Q = \mathcal{N}(0, \sigma^2)$. By Theorem 3.2 in (Gopi et al., 2021), the PRV $Y$ is defined as:

$$Y = \log\left(\frac{Q(t)}{P(t)}\right) \quad \text{where } t \sim Q = \mathcal{N}(0, \sigma^2).$$

Substituting the Gaussian distributions for $P$ and $Q$, we have:

$$Y = \log\left(\frac{\exp\left(-t^2/2\sigma^2\right)}{\exp\left(-(t-\Delta)^2/2\sigma^2\right)}\right) = \frac{(t-\Delta)^2}{2\sigma^2} - \frac{t^2}{2\sigma^2} = \frac{\Delta^2}{2\sigma^2} - \frac{\Delta}{\sigma^2}t \sim \mathcal{N}\left(\frac{\Delta^2}{2\sigma^2}, \frac{\Delta^2}{\sigma^2}\right).$$

A similar calculation shows that $X = \mathcal{N}\left(-\frac{\Delta^2}{2\sigma^2}, \frac{\Delta^2}{\sigma^2}\right)$. $\square$

By combining Proposition 1 and Proposition B.3 in (Gopi et al., 2021), we have the following:

**Theorem 8** (PRVs of the Sub-sampled Gaussian Mechanism (Gopi et al., 2021)). *Let $(X, Y)$ be the PRVs for the privacy curve of the Gaussian mechanism $\delta(\mathcal{N}(\Delta, \sigma^2) \| \mathcal{N}(0, \sigma^2))$, and let $q$ be the sub-sampling probability. Then the PRVs $(X_q, Y_q)$ for the privacy curve of the sub-sampled Gaussian mechanism are:*

$$X_q = \log\left(1 + q(e^X - 1)\right), \tag{21}$$

$$Y_q = \begin{cases} \log\left(1 + q(e^Y - 1)\right), & \text{w.p. } q, \\ \log\left(1 + q(e^X - 1)\right), & \text{w.p. } 1 - q. \end{cases} \tag{22}$$

We are ready to describe the steps to calculate noise variance $\sigma^2$.

**Steps for numerical calculation of $\sigma^2$:**

1. **Initialize and fix $\sigma^2$:** Start with an initial guess for the noise variance $\sigma^2$ within a feasible range (e.g., $\sigma \in [0.3, 3.0]$). Fix $\sigma^2$ and proceed to evaluate the privacy curve on the basis of this value.

2. **Compute PRVs for the sub-sampled Gaussian Mechanism:** For the given $\sigma^2$ and sub-sampling rate $q$, compute the PRVs $(X_q^{(j)}, Y_q^{(j)})$ for the $j$-th iteration, as defined in Theorem 8.

3. **Compose PRVs across $T$ iterations:** Sum the PRVs across all iterations to compose the corresponding privacy curve:

$$\hat{X}_{\text{total}} = \sum_{j=1}^{T} X_q^{(j)}, \quad \hat{Y}_{\text{total}} = \sum_{j=1}^{T} Y_q^{(j)}.$$

4. **Evaluate the privacy curve:** Use the PRVs $(\hat{X}_{\text{total}}, \hat{Y}_{\text{total}})$ to compute the privacy curve representing the cumulative privacy guarantees:

$$\delta_{\text{total}}(\epsilon) = \Pr\left[\hat{Y}_{\text{total}} > \epsilon\right] - e^\epsilon \Pr\left[\hat{X}_{\text{total}} > \epsilon\right].$$

5. **Check and adjust $\sigma^2$:** Compare the evaluated privacy curve $\delta_T(\hat{\epsilon})$ against the pre-defined privacy parameter $\hat{\delta}$:

   - If $\delta_{\text{total}}(\hat{\epsilon}) \leq \hat{\delta}$, the current $\sigma^2$ satisfies the privacy guarantees.
   - If $\delta_{\text{total}}(\hat{\epsilon}) > \hat{\delta}$, increase $\sigma^2$ and repeat steps 2–5.

   This iterative adjustment ensures that $\sigma^2$ is as small as possible while satisfying the pre-defined privacy parameters $(\hat{\varepsilon}, \hat{\delta})$.

6. **Finalize $\sigma^2$:** Once a $\sigma^2$ is found that satisfies the privacy parameters $(\hat{\epsilon}, \hat{\delta})$, terminate the search.

**Practical implementation:** The iterative process of adjusting $\sigma^2$ can be efficiently implemented using *binary search*. Start with a range for $\sigma$ (e.g., $[0.3, 3.0]$) and refine the range until $\delta_{\text{total}}(\hat{\epsilon})$ closely matches $\hat{\delta}$. Numerical evaluation involves discretizing and truncating the continuous PDFs of $\hat{X}_{\text{total}}$ and $\hat{Y}_{\text{total}}$, ensuring precise calculations with controllable errors $\epsilon_{\text{error}}$ and $\delta_{\text{error}}$, as detailed in (Gopi et al., 2021).

Table 4 summarizes the used specific values of noise variance $\sigma^2$ for each experimental configuration detailed in Appendix F. The noise variance $\sigma^2$ is determined within $[0.3, 3.0]$ by using a binary search with an interval of $0.01$. Note that the total number of compositions is computed as the product of $n_{\text{shot}}$, representing the number of the generated demonstrations associated with the same label $y$, and $T$, the length of each generated demonstration.

Table 4: The noise variance $\sigma^2$ for the experimental configuration detailed in Appendix F.

| Dataset | $\min_y \mathcal{D}_{\text{priv}}^y$ | $M$ | $N$ | $T$ | $n_{\text{shot}}$ per label $y$ | $\delta_{\text{error}}$ | $\varepsilon_{\text{error}}$ | $\delta$ | $\sqrt{\sigma^2}$ for $\varepsilon = 1, 2, 4, 8$ |
|---|---|---|---|---|---|---|---|---|---|
| GINC | 1600 | 5 | 4 | 10 | 4 | | | | $[0.70, 0.59, 0.47, 0.37]$ |
| AGNEWS | 30,000 | 10 | 2 | 100 | 1 | $10^{-10}$ | 0.01 | $\frac{1}{\min_y \left|\mathcal{D}_{\text{priv}}^y\right|}$ | $[0.51, 0.46, 0.39, 0.31]$ |
| DBPedia | 40,000 | 40 | 2 | 100 | 1 | | | | $[0.63, 0.54, 0.45, 0.36]$ |
| TREC | 835 | 80 | 1 | 15 | 1 | | | | $[1.33, 0.94, 0.69, 0.51]$ |

## E. Supplementary details of PTA

In this section, we provide the derivation of the proposed PTA from the KL-constrained problem defined in (11). Since this derivation relies on an estimated likelihood ratio based on observable prompts, we also explain the rationale behind this estimation and its implications, grounded in the Bayesian interpretation of ICL Finally, we describe the algorithmic implementation used in Algorithm 1 to generate DP synthetic demonstrations.

### E.1. Derivation of PTA

Here, we provide the derivation of the PTA, which modifies the next-token probabilities, as shown in (13). To derive the modified next-token probabilities in PTA, we aim to find the valid probability vector $\hat{p} := (\hat{p}_v)_{v \in \mathcal{V}_{\text{pub}}}$ that maximizes the objective in (11), using the likelihood ratio estimate in (12):

$$\max_{\hat{p}} \sum_{v \in \mathcal{V}_{\text{pub}}} \hat{p}_v \log \frac{p(o^{\text{LLM}} = v | S_{\text{priv}}^{(i)})}{p(o^{\text{LLM}} = v | S_{\text{pub}})} - \frac{1}{\alpha} D_{\text{KL}} \left( \hat{p} \| p_{\text{base}}(o^{\text{LLM}}) \right), \quad \text{s.t.} \quad \sum_{v \in \mathcal{V}_{\text{pub}}} \hat{p}_v = 1, \quad \hat{p}_v \geq 0 \quad \forall v \in \mathcal{V}_{\text{pub}}. \quad (23)$$

We transform the objective as follows:

$$\arg\max_{\hat{p}_v} \sum_{v \in \mathcal{V}_{\text{pub}}} \hat{p}_v \log \frac{p(o^{\text{LLM}} = v | S_{\text{priv}}^{(i)})}{p(o^{\text{LLM}} = v | S_{\text{pub}})} - \frac{1}{\alpha} D_{\text{KL}} \left( \hat{p} \| p_{\text{base}}(o^{\text{LLM}}) \right)$$

$$= \arg\max_{\hat{p}_v} \sum_{v \in \mathcal{V}_{\text{pub}}} \hat{p}_v \left( \log \frac{p(o^{\text{LLM}} = v | S_{\text{priv}}^{(i)})}{p(o^{\text{LLM}} = v | S_{\text{pub}})} - \frac{1}{\alpha} \log \frac{\hat{p}_v}{p_{\text{base}}(o^{\text{LLM}} = v)} \right)$$

$$= \arg\min_{\hat{p}_v} \sum_{v \in \mathcal{V}_{\text{pub}}} \hat{p}_v \left( \log \frac{\hat{p}_v}{p_{\text{base}}(o^{\text{LLM}} = v)} - \alpha \log \frac{p(o^{\text{LLM}} = v | S_{\text{priv}}^{(i)})}{p(o^{\text{LLM}} = v | S_{\text{pub}})} \right)$$

$$= \arg\min_{\hat{p}_v} \sum_{v \in \mathcal{V}_{\text{pub}}} \hat{p}_v \left( \log \frac{\hat{p}_v}{\frac{1}{Z} p_{\text{base}}(o^{\text{LLM}} = v) \exp \left( \alpha \log \frac{p(o^{\text{LLM}} = v | S_{\text{priv}}^{(i)})}{p(o^{\text{LLM}} = v | S_{\text{pub}})} \right)} - \log Z \right)$$

$$= \arg\min_{\hat{p}_v} D_{\text{KL}} \left( \hat{p} \| \hat{p}^* \right). \quad (24)$$

For the second and fourth equalities, $D_{\text{KL}} (\cdot \| \cdot)$ is evaluated over the limited vocabulary space $\mathcal{V}_{\text{pub}}$. Furthermore, in the third equality, we introduce the following definitions:

$$Z = \sum_{v \in \mathcal{V}_{\text{pub}}} p_{\text{base}}(o^{\text{LLM}} = v) \exp \left( \alpha \log \frac{p(o^{\text{LLM}} = v | S_{\text{priv}}^{(i)})}{p(o^{\text{LLM}} = v | S_{\text{pub}})} \right), \quad \hat{p}_v^* = \frac{1}{Z} p_{\text{base}}(o^{\text{LLM}} = v) \exp \left( \alpha \log \frac{p(o^{\text{LLM}} = v | S_{\text{priv}}^{(i)})}{p(o^{\text{LLM}} = v | S_{\text{pub}})} \right).$$

These definitions ensure that $\hat{p}^* := (\hat{p}_v^*)_{v \in \mathcal{V}_{\text{pub}}}$ is a valid probability vector satisfying, $\sum_{v \in \mathcal{V}_{\text{pub}}} \hat{p}_v^* = 1, \hat{p}_v^* \geq 0$. On the basis of Gibbs' inequality, the transformed objective in (24) is minimized precisely when $\forall v \in \mathcal{V}_{\text{pub}}, \quad \hat{p}_v = \hat{p}_v^*$. This completes the derivation of PTA, showing that the modified distribution is given by:

$$\hat{p}_v \propto p_{\text{base}}(o^{\text{LLM}} = v) \left( \frac{p(o^{\text{LLM}} = v | S_{\text{priv}}^{(i)})}{p(o^{\text{LLM}} = v | S_{\text{pub}})} \right)^{\alpha}.$$

### E.2. Rationale and implications of the likelihood estimation in (12)

To solve the objective (11), PTA relies on the likelihood ratio estimation given in (12). For the clarity, we restate the estimation here:

$$\log \frac{p(o^{\text{LLM}} = v | \theta^*)}{p(o^{\text{LLM}} = v | \theta)} \approx \log \frac{p(o^{\text{LLM}} = v | S_{\text{priv}}^{(i)})}{p(o^{\text{LLM}} = v | S_{\text{pub}})}.$$

---

**Algorithm 2** VocabSpaceLimit

---

**Input:** $\mathcal{V}, k \leq |\mathcal{V}|, p \in [0, 1], p(o^{\text{LLM}} = v | S_{\text{pub}})$

1: $\mathcal{V}' = \underset{\mathcal{V}' \subseteq \mathcal{V}}{\arg\max} \sum_{v \in \mathcal{V}'} p(o^{\text{LLM}} = v | S_{\text{pub}}), \quad \text{s.t.} \quad |\mathcal{V}'| = k$

2: $\mathcal{V}'' = \underset{\mathcal{V}'' \subseteq \mathcal{V}}{\arg\min} |\mathcal{V}''|, \quad \text{s.t.} \quad \sum_{v \in \mathcal{V}''} p(o^{\text{LLM}} = v | S_{\text{pub}}) \geq p$

3: $\mathcal{V}_{\text{pub}} = \mathcal{V}' \cap \mathcal{V}''$

4: **return** $\mathcal{V}_{\text{pub}}$

---

As we discussed in Section 4.2, the LHS of (12) is intractable to compute because $\theta^*$ and $\theta$ are latent concepts — neither explicitly parameterized nor observable in practice — making it impossible to directly evaluate the likelihoods conditioned on $\theta$ and $\theta^*$. To address this, we estimate the objective using tractable next-token probability distributions conditioned on observable prompts.

This estimation is grounded in the Bayesian interpretation of ICL (Xie et al., 2022), which views prompting an LLM with demonstrations as inducing a posterior distribution concentrated around the ground-truth concept underlying those demonstrations. In this view, conditioning the LLM on an observable prompt $S$ approximates inference over a latent concept $p(\theta|S)$. When the prompt consists of demonstrations aligned with a ground-truth concept $\theta^*$, the resulting next-token probability distribution asymptotically converges to $p(o^{\text{LLM}} = v | \theta^*)$. Accordingly, the private prompt $S_{\text{priv}}^{(i)}$ serves as a surrogate for conditioning on $\theta^*$, while the instruction-only public prompt $S_{\text{pub}}$ serves as a generic reference, approximating conditioning on any other concept $\theta$.

To clarify how the estimation works, we decompose the LHS of (12) as follows:

$$\log \frac{p(o^{\text{LLM}} = v | \theta^*)}{p(o^{\text{LLM}} = v | \theta)} = \log \frac{p(o^{\text{LLM}} = v | \theta^*)}{p(o^{\text{LLM}} = v | S_{\text{pub}})} + \log \frac{p(o^{\text{LLM}} = v | S_{\text{pub}})}{p(o^{\text{LLM}} = v | \theta)}. \tag{25}$$

This decomposition reveals that the estimation in (12) corresponds to substituting the intractable first term on the RHS of (25) with tractable log likelihood ratio $\log \frac{p(o^{\text{LLM}} = v | S_{\text{priv}}^{(i)})}{p(o^{\text{LLM}} = v | S_{\text{pub}})}$, motivated by the Bayesian interpretation of ICL discussed above, while omitting the second term. The omitted second term in (25) becomes negligible compared to the first when the sets of high-probability tokens are well-separated across concepts, since in such cases tokens distinctive to the ground-truth concept $\theta^*$ are unlikely under both the public prompt and other concepts.

To encourage this separation, we design $S_{\text{pub}}$ to include only general task instructions and no concept-specific information. This neutral design helps promote divergence in token likelihoods across concepts relative to the likelihood conditioned on $S_{\text{pub}}$, thereby supporting the plausibility of omitting the second term in (25).

### E.3. Algorithmic implementation

Algorithm 2 outlines how we limit the vocabulary space to a plausible subset by applying both a static threshold $k$ (on the maximum number of tokens) and a dynamic threshold $p$ (on the cumulative probability of selected tokens). This dynamic adjustment adaptively decreases the negative noise impacts primarily influenced by the vocabulary size in (9).

In Algorithm 3, we detail how to generate prompts from randomly sampled demonstrations. This procedure is identical to that of (Tang et al., 2024), and the prompt construction functions $PB(\cdot)$ for each dataset appear in Appendix F.9.

---

**Algorithm 3** GenPrompt

---

**Input:** instruction, $\mathcal{D}_{\text{priv}}$, $M$, $N$, $\tilde{y}$, $\tilde{O}$
1: $\mathcal{D}'_{\text{priv}} \leftarrow$ randomly draw $MN$ samples from $\mathcal{D}_{\text{priv}}$ with label $\tilde{y}$
2: $S_{\text{pub}} \leftarrow PB(\text{instruction}, \tilde{y}, \tilde{O}))$ where $PB(\cdot)$ defined in Table 13
3: **for** $i = 1$ to $M$ **do**
4:     $\mathcal{D}^{(i)}_{\text{priv}} \leftarrow \mathcal{D}'_{\text{priv}}[(i-1)N : iN]$
5:     $S^{(i)}_{\text{priv}} \leftarrow PB(\text{instruction}, \mathcal{D}^{(i)}_{\text{priv}}, \tilde{y}, \tilde{O}))$
6: **end for**
7: **return:** $S_{\text{pub}}, S^{(1)}_{\text{priv}}, \ldots, S^{(M)}_{\text{priv}}$

---

# F. Additional Experiments

We first describe the computing resources used for GINC and text-classification tasks in Appendix F.1, followed by a detailed account of each dataset (AGNews, DBPedia, TREC, and GINC) in Appendix F.2. Appendix F.3 focuses on GINC, explaining its synthetic generation process via factorial HMMs, the corresponding GPT-2 pre-training procedures, and how they align with earlier work.

Next, we outline the implementation details in Appendix F.4, including the calibration techniques from (Zhao et al., 2021) for stabilizing DP-ICL, as well as methods for computing the log-likelihood of generated demonstrations in GINC. Then, we report the accuracy and distinguishability comparison across various configurations and privacy parameters, omitted in the main paper, resepectively in Appendices F.5 and F.6. Finally, we offer examples of DP synthetic demonstrations, illustrating how our method amplifies the distinctive tokens while preserving coherence.

## F.1. Computing resources

For GINC, we used a server that has 2 GPUs (NVIDIA GeForce RTX 3080) with 2 CPUs (Intel Xeon Gold 5218R, 2.1 GHz). For text-classification tasks, we used a server with 8 GPUs (NVIDIA A6000) and 2CPUs (Intel Xeon Gold 6346, 3.10 GHz) for text-classification datasets.

## F.2. Datasets

This section describes the datasets used in our experiments.

- **AGNews:** The AGNews dataset (Zhang et al., 2015) involves topic classification with labels categorized into four classes: World, Sports, Business, and Technology. It includes 30,000 training samples and 1,900 test samples per class. We utilized all 30,000 training samples and 1,000 test samples.

- **DBPedia**: The DBPedia ontology classification dataset (Zhang et al., 2015) involves topic classification with labels categorized into 14 classes: Company, School, Artist, Athlete, Politician, Transportation, Building, Nature, Village, Animal, Plant, Album, Film, and Book. The dataset includes 40,000 training samples and 5,000 test samples per class.

- **TREC:** The TREC (Voorhees & Tice, 2000) question classification dataset involves classifying questions into six labels: Number, Location, Person, Description, Entity, and Abbreviation. It comprises 5,500 training samples and 500 test samples, distributed non-uniformly across the labels.

- **GINC:** The GINC dataset (Xie et al., 2022) is a synthetic text-classification dataset generated from a uniform mixture of five factorial HMMs. Each HMM represents a latent concept, an underlying pattern that governs token transitions in sequences. Using the original pre-training documents from (Xie et al., 2022), we further created 1,600 training and 400 test samples per concept for evaluating accuracy of ICL, ensuring no duplication.[4] Unlike the datasets described above, which have fixed label sets, GINC uses dynamic labels. Each label corresponds to the most likely last token of each data, determined by the GINC distribution for its corresponding latent concept. Details on the GINC distribution hyperparameters are provided in Appendix F.3.

## F.3. Details on setup for GINC

**HMMs setup.** The GINC dataset, introduced in (Xie et al., 2022), is designed to study ICL in a controlled setup. It is generated from a uniform mixture of five factorial HMMs, each representing a distinct *latent concept* $\theta_i \in \Theta (i = 1, \ldots, 5)$. Each HMM emits a token $v$ from vocabulary $\mathcal{V}$, which is constructed by enumerating combinations of letters (e.g., "a" to "z," "aa" to "az") with backslash designed as the delimiter token.

To model how real documents are generated using the meaningless vocabulary, each HMM comprises two independent Markov chains: one representing transitions between entities $e_t \in \{1, \ldots, |\mathcal{E}|\}$ (e.g., Einstein, Gandhi, etc.), and the other capturing transitions between properties $s_t \in \{1, \ldots, |\mathcal{S}|\}$ (e.g., nationality, occupation etc.). The emission probabilities $p(o_t|h_t)$ depend on the combination of entities and properties, i.e., $p(o_t|h_t) = p(o_t|e_t, s_t)$. These two chains are crucial because real documents often involve interdependent patterns: entities (e.g., Einstein) are associated with properties (e.g., scientist). Modeling these two aspects independently allows the HMM to reflect how documents are structured in the real

---

[4]To guarantee the privacy parameters of $(\varepsilon, \delta)$-DP, unintended exact duplication in training data can underestimate leakage risk, violating the assumed $(\varepsilon, \delta)$-DP. De-duplication is therefore essential.

Table 5: Empirical evaluation of $C^{\text{delim}} + \frac{1}{n} \log \frac{1}{c_7}$ appearing in RHS of Lemma 1 when $n = 4$.

| $\theta^*$ | $\theta_1$ | $\theta_2$ | $\theta_3$ | $\theta_4$ | $\theta_5$ |
|---|---|---|---|---|---|
| $C^{\text{delim}}$ | 42.75 | 49.94 | 46.00 | 41.53 | 39.52 |

Table 6: Pre-training train validation loss and the accuracy of ICL on GINC with $|\mathcal{V}| = 150$ for GPT-2 models varying in size.

| Model | train loss | validation loss | Accuracy of ICL |
|---|---|---|---|
| (MA1) GPT-2 (4-layer) | 1.264 | 1.283 | 92.80 |
| (MA2) GPT-2 (12-layer) | 1.263 | 1.279 | 98.40 |
| (MA3) GPT-2 (16-layer) | 1.267 | 1.283 | 99.70 |

world. These HMMs are then used to generate pre-training documents, training samples, and test samples for ICL, using the scripts provided in their code[5].

Importantly, the defined HMMs satisfy Assumption 1–4, which are essential for deriving the conditions under which ICL can accurately infer the true latent concept, as established in Lemma 1 and Theorem 2. These assumptions enable the numerical calculation of the specific value $C^{\text{delim}} + \frac{1}{n} \log \frac{1}{c_7}$, a key parameter in Lemma 1 and Theorem 2. The calculated values are presented in Table 5.

**Pre-training of GPT-2 models.** We pre-train GPT-2 models by using the generated pre-training documents, following the settings detailed in Appendix F.2 of (Xie et al., 2022). Using pre-trained models, we evaluate the accuracy of ICL. The validation loss and accuracy of ICL for the models are summarized in Table 6. The results are mostly consistent with those reported in Figures 6 of (Xie et al., 2022), validating our pre-training setup of GINC.

### F.4. Implementation Details

In this section, we provide the detailed implementation used in the evaluation of the conducted experiments.

**Calibration of ICL** Following (Zhao et al., 2021), we implement calibration to stabilize the accuracy of DP-ICL when evaluating DP synthetic demonstrations in classification tasks. To estimate the model's bias toward each answer, we use a content-free test input (e.g., "N/A") alongside the training prompt and analyze the resulting predictions. Calibration parameters are then fitted to ensure uniform predictions across answers for the content-free input. The calibration is performed by using the following two methods:

$$\text{(Diagonal)} \qquad \hat{\mathbf{p}} = \text{Softmax}(\mathbf{W}\mathbf{p} + \mathbf{b}), \quad \mathbf{W} = \text{diag}(\mathbf{p}_{\text{cf}}^{-1}), \qquad (26)$$

$$\text{(Identity)} \qquad \hat{\mathbf{p}} = \text{Softmax}(\mathbf{p} - \mathbf{p}_{\text{cf}}), \qquad (27)$$

where $\mathbf{p}$ is a probability vector representing the model's predicted probabilities for each label in the classification task, normalized to sum to one. Its dimension corresponds to the number of possible labels in the task. This vector is obtained by conditioning an LLM on the prompt with demonstrations. Similarly, $\mathbf{p}_{\text{cf}}$ is the probability vector obtained from the content-free test input, also normalized to sum to one. By scaling $\mathbf{W}$ or subtracting $\mathbf{p}_{\text{cf}}$ the content-free probabilities, the model's bias toward each answer can be corrected, improving calibration for classification tasks.

For text-classification datasets, we use "N/A," an empty string (""), or "[MASK]" as the content-free input. For GINC, we use token sequences of the same length as the demonstrations, where each token is sampled independently and uniformly at random from the vocabulary. This approach serves as a content-free input because GINC's vocabulary lacks a suitable representation like "N/A" or "[MASK]."

**Evaluation of log-likelihood for GINC.** To validate PTA's effectiveness, we examine the condition established in Lemma 1. The controlled setup of GINC allows the precise numerical computation of the log-likelihood of generated DP synthetic demonstrations, leveraging the parameters of the HMM.

---

[5] https://github.com/p-lambda/incontext-learning

Given the generated sequences $(o_1, \ldots, o_T)$, the joint probability $p(o_1, \ldots, o_T, h_T)$ can be computed recursively using the forward message passing algorithm (Rabiner, 1989):

$$p(o_1, \ldots, o_T, h_T) = p(o_T|h_T) \sum_{h_{T-1}} p(o_1, \ldots, o_{T-1}, h_{T-1})p(h_T|h_{T-1}). \tag{28}$$

Starting from $p(o_1, h_1)$, where $h_t = (s_t, e_t)$ the total likelihood of the sequence is obtained by summing over all possible hidden states at the final time step:

$$p(o_1, \ldots, o_T) = \sum_{h_T} p(o_1, \ldots, o_T, h_T).$$

The initialization step incorporates the start distribution $p(h_1)$, which is set as a uniform distribution over all possible combinations of entity $e_t$ and property $s_t$.

### F.5. Accuracy Comparison

**Hyperparameters.** We select optimal hyperparameter settings by averaging accuracy across $\varepsilon \in \{1, 2, 4, 8\}$. For all methods, we explore $k \in \{10, 100\}$ and $p \in \{0.7, 0.8, 0.9\}$ (for (PA2) and (BA2)). We search $\alpha \in \{1.0, 1.5, 2.0\}$ for (P1), (PA2), and (PA3) on the text-classification tasks, and extend this range to $\{1.0, 1.5, 2.0, 5.0\}$ for GINC. Notably, $k = 10$ consistently outperforms $k = 100$. All other chosen hyperparameters are summarized in Table 7.

**Effects of varying model.** When comparing different model configurations, we observe that (PA2) outperforms (P1) when the non-private baseline (R0) reports lower accuracy, particularly when using (MA1) and (MB1). One possible explanation is that, without top-$p$ truncation when limiting the vocabulary space, (P1) may unintentionally amplify very low-probability tokens through PTA. For text-classification tasks, the quantized model (MB1) exhibits a higher standard deviation of accuracy, as it exhibits the worst accuracy of (R0).

**Ablation on the proposed PTA.** Complementing (P1), the PTA variant with Top-$p$ (PA2), outperforms (B1, BA2) across text-classification tasks, and often surpasses (P1), particularly in DBPedia ($\varepsilon = 1$). However, (PA2) exhibits significantly worse accuracy in GINC than (P1), indicating that top-$p$ truncation may involve trade-offs under certain conditions. (PA2) prevents LLM from retaining low-probability tokens, which are somehow critical with small vocabularies (e.g., $|\mathcal{V}| = 150$). Despite these drawbacks, both (P1) and (PA2) outperform the non-private baseline (R0) in TREC ($\varepsilon = 1$). This observation is consistent with (Tang et al., 2024), suggesting that private generation mechanisms can potentially improve generalization in small-scale datasets.

Moreover, (P1) consistently outperforms (PA3), which considers the optimization defined in (11) without KL-constraint, when using (MA3) and (MB3), which yield the highest non-private (R0) performance, while (PA3) performs better with (MA1) and (MB2). This indicates that, if the original model produces more coherent text, introducing KL regularization (as in (P1)) effectively preserves the natural flow of the generated demonstrations. Conversely, when the LLM performs worse without noise addition, ignoring regularization (as in (PA3)) can lead to notable accuracy gains for DP-ICL (AGNews with (MB1) and $\varepsilon = 1$ and TREC AGNews with (MB2) and $\varepsilon = 1$).

To further understand the behavior of (P1) and (PA3) under varying privacy paramter, we examine how accuracy changes as the privacy parameter $\varepsilon$ decreases. We plot accuracy across different $\varepsilon = \{1, 2, 4, 8, \infty\}$ values for models (MA1), (MA2) and (MA3) in Figure 3. Figure 3 illustrates a clear trend: accuracy decreases as privacy strengthens. We highlight (MA1) as it exhibits a distinct pattern across methods, unlike (MA2) and (MA3), where (P1) consistently outperforms (B1). In contrast, for (MA1), (P1) performs the worst, while (PA3) is comparable to (B1). This difference may arise from (MA1)'s limited zero-shot accuracy, which makes the regularization in (P1) – penalizing deviation from the base next-token distribution shown in (11) — less effective in improving accuracy. In this setting, the regularization keeps the next-token probability distribution close the model's original output, which performs poorly, thus leading to degraded accuracy. By omitting this constraint, (PA3) allows more flexibility and achieves better performance.

**Accuracy comparison when $\varepsilon = \infty$.** Using $\sigma = 0$ or synthetic demonstration generation does not satisfy DP (i.e. $\varepsilon = \infty$), but it allows us to verify whether the resulting demonstrations increase the divergence between the ground-truth concept and any alternative. The bottom row of Table 8 shows the accuracy with $\sigma = 0$. (P1) outperforms (R0) except for GINC and DBPedia and this indicates the effectiveness of PTA, in increasing the divergence between concepts.

Table 7: The distinctive hyperparameters $p$ and $\alpha$ are selected to suit the specific combination of datasets, models, and methods. The table reports the values used for the results presented in Tables 2 and 8.

(a) The selected amplification strength $\alpha$ introduced in PTA.

| Methods | GINC (MA1) | (MA2) | (MA3) | AGNews (MB1) | (MB2) | (MB3) | DBPedia (MB1) | (MB2) | (MB3) | TREC (MB1) | (MB2) | (MB3) |
|---|---|---|---|---|---|---|---|---|---|---|---|---|
| (P1) | 5.0 | 2.0 | 5.0 | 2.0 | 2.0 | 1.5 | 2.0 | 1.0 | 1.0 | 2.0 | 1.5 | 2.0 |
| (PA2) | 5.0 | 5.0 | 2.0 | 2.0 | 2.0 | 1.0 | 1.0 | 1.5 | 1.5 | 1.5 | 1.0 | 1.0 |
| (PA3) | 2.0 | 1.5 | 5.0 | 1.0 | 1.0 | 1.0 | 1.0 | 1.0 | 1.0 | 1.0 | 1.0 | 1.0 |

(b) The selected adaptive threshold $p$ for limiting vocabulary space.

| Methods | GINC (MA1) | (MA2) | (MA3) | AGNews (MB1) | (MB2) | (MB3) | DBPedia (MB1) | (MB2) | (MB3) | TREC (MB1) | (MB2) | (MB3) |
|---|---|---|---|---|---|---|---|---|---|---|---|---|
| (PA2) | 0.9 | 0.9 | 0.9 | 0.9 | 0.9 | 0.8 | 0.7 | 0.9 | 0.7 | 0.8 | 0.8 | 0.9 |
| (PA3) | 0.9 | 0.9 | 0.9 | 0.9 | 0.7 | 0.8 | 0.8 | 0.7 | 0.8 | 0.8 | 0.7 | 0.9 |
| (BA2) | 0.9 | 0.9 | 0.9 | 0.9 | 0.7 | 0.9 | 0.7 | 0.7 | 0.8 | 0.7 | 0.8 | 0.9 |

Table 8: 4-shot DP-ICL accuracy across four datasets and three models, averaged over five different seeds. The highest accuracy is bolded, and the second-highest is underlined.

| $\varepsilon$ | Methods | GINC (MA1) | (MA2) | (MA3) | AGNews (MB1) | (MB2) | (MB3) | DBPedia (MB1) | (MB2) | (MB3) | TREC (MB1) | (MB2) | (MB3) |
|---|---|---|---|---|---|---|---|---|---|---|---|---|---|
| 1 | (P1) | 81.13±1.00 | 93.65±1.84 | **93.99±1.35** | 77.54±5.62 | 82.22±4.22 | **87.88±1.11** | 78.74±4.41 | 80.54±1.06 | 82.06±3.61 | 70.20±7.52 | 75.04±3.81 | **84.80±1.66** |
| | (PA2) | **85.90±1.02** | 94.12±0.91 | 90.06±1.63 | 77.28±6.06 | 79.84±4.32 | 87.00±1.16 | 78.88±3.37 | **81.12±0.84** | 84.72±3.54 | 75.32±2.21 | 75.24±2.21 | 84.72±2.50 |
| | (PA3) | 85.40±1.39 | 92.58±1.92 | 90.14±0.95 | 78.18±3.38 | 81.16±3.40 | 83.24±2.94 | 79.50±1.38 | 79.60±3.25 | **86.56±1.75** | 74.48±4.93 | **77.48±2.38** | 84.34±1.28 |
| | (B1) | 81.53±1.90 | 92.91±1.53 | 90.80±2.79 | 76.86±6.82 | 78.76±5.29 | 83.74±1.90 | 79.22±2.25 | 79.80±2.02 | 85.72±1.44 | 73.00±5.15 | 77.32±2.05 | 83.56±5.09 |
| | (BA2) | 83.46±2.08 | 92.09±2.49 | 89.34±0.98 | **78.50±4.46** | 68.89±5.59 | 84.86±2.83 | **79.72±1.54** | 79.62±2.23 | 86.10±1.83 | **76.92±2.08** | 77.12±1.72 | 81.00±2.57 |
| 2 | (P1) | 82.26±0.68 | **95.36±0.74** | **95.17±0.82** | 81.82±3.00 | 85.52±2.72 | **87.24±1.37** | 80.06±2.69 | **81.30±0.85** | 84.50±1.99 | 70.80±4.89 | 74.88±2.47 | **84.76±1.42** |
| | (PA2) | 85.34±1.21 | 93.98±1.68 | 90.88±1.90 | 79.78±6.19 | 78.74±2.52 | 85.06±2.30 | 78.82±3.90 | 80.16±2.73 | 84.52±2.90 | **77.44±1.93** | 78.00±2.49 | 83.92±3.62 |
| | (PA3) | **86.09±1.02** | 93.43±1.72 | 90.41±1.48 | 80.50±2.76 | 79.96±4.36 | 81.48±5.29 | **80.32±2.46** | 80.60±2.21 | **86.00±1.14** | 71.52±2.11 | 77.76±1.52 | 82.86±1.95 |
| | (B1) | 83.30±1.47 | 93.91±1.31 | 91.41±1.66 | 77.02±5.73 | 76.60±4.65 | 83.28±2.67 | 79.90±1.41 | 80.60±1.25 | 84.28±1.99 | 73.20±7.92 | 75.52±2.58 | 83.04±2.52 |
| | (BA2) | 84.55±2.07 | 93.19±2.30 | 90.73±0.83 | 72.74±6.55 | 66.77±4.05 | 81.58±4.50 | 78.70±3.18 | 80.12±1.90 | 84.74±1.62 | 73.84±2.70 | **78.72±3.15** | 83.20±1.57 |
| 4 | (P1) | 81.41±0.58 | **96.20±0.27** | **96.76±0.91** | 80.80±5.80 | 82.16±7.39 | **87.68±1.28** | 78.64±2.17 | 80.08±0.70 | **86.06±1.97** | 71.20±4.34 | 73.16±4.20 | 83.88±1.27 |
| | (PA2) | 85.72±0.91 | 95.07±0.78 | 91.76±1.84 | 78.06±7.44 | 78.78±4.73 | 86.10±1.78 | 79.28±2.05 | 80.04±2.81 | 84.94±3.62 | 75.96±3.01 | 77.12±2.61 | **84.40±0.97** |
| | (PA3) | **86.75±0.62** | 94.56±0.96 | 91.64±1.60 | 76.40±1.79 | 79.88±5.20 | 81.04±9.03 | **80.44±1.88** | 80.34±1.86 | 84.04±2.94 | 74.52±3.10 | 77.62±3.35 | 83.10±1.87 |
| | (B1) | 85.93±1.77 | 95.16±1.32 | 93.27±1.39 | 78.74±2.49 | 77.64±3.29 | 86.10±1.81 | 79.96±2.75 | 80.16±2.11 | 85.04±2.69 | 74.28±4.61 | 76.72±1.89 | 82.48±3.42 |
| | (BA2) | 85.81±0.60 | 93.31±2.29 | 90.07±0.78 | 77.32±3.73 | 68.30±3.67 | 82.14±3.65 | 79.52±2.58 | 79.02±2.00 | 84.84±3.40 | **76.72±2.21** | **77.64±3.09** | 82.16±2.37 |
| 8 | (P1) | 82.38±0.77 | 95.62±0.67 | **96.61±1.40** | 78.50±6.29 | 85.32±1.91 | **86.48±1.67** | 80.06±1.55 | 79.80±0.38 | 84.78±1.55 | 70.16±2.18 | 75.04±1.56 | 84.24±2.33 |
| | (PA2) | 86.37±1.10 | 95.33±1.36 | 91.15±1.24 | 78.04±3.86 | 79.74±5.12 | 84.24±2.20 | 80.40±2.61 | 79.92±3.58 | 84.12±3.47 | 75.80±2.91 | **78.20±1.79** | 83.52±1.02 |
| | (PA3) | 86.46±0.99 | 95.18±1.00 | 90.79±1.16 | **81.58±2.99** | 78.56±4.87 | 84.92±1.75 | **80.44±1.78** | 80.04±1.57 | 84.60±2.73 | 72.28±2.84 | 78.02±1.69 | **84.54±1.93** |
| | (B1) | **87.94±0.85** | **95.84±0.81** | 94.63±0.55 | 79.50±5.21 | 80.09±2.79 | 83.62±3.08 | 77.82±2.02 | 78.90±1.54 | 84.40±2.39 | 72.48±4.17 | 75.96±2.69 | 82.64±2.79 |
| | (BA2) | 86.46±0.78 | 94.64±1.56 | 90.61±1.23 | 79.78±3.92 | 67.17±2.01 | 82.96±2.32 | 79.18±2.90 | 79.12±2.59 | **84.86±1.38** | **76.16±2.71** | 75.12±3.13 | 81.96±1.59 |
| $\infty$ | (P1) | 81.03±1.28 | 96.11±0.40 | 96.57±0.83 | 84.14±1.58 | **86.12±3.87** | **88.28±0.84** | 79.42±1.33 | 80.68±0.63 | 82.78±4.00 | 73.16±3.04 | 73.52±4.72 | 83.56±2.35 |
| | (PA2) | 86.67±1.22 | 96.18±0.78 | 92.46±1.32 | 78.24±4.51 | 80.08±5.87 | 83.86±3.23 | 78.76±1.76 | 80.58±0.60 | 84.46±3.10 | 75.88±2.37 | 77.72±1.42 | **86.04±1.01** |
| | (PA3) | 86.67±0.77 | 95.81±0.83 | 91.21±1.09 | 81.98±2.24 | 79.98±2.73 | 85.82±2.03 | 78.88±2.17 | 80.02±2.65 | 85.68±1.39 | **77.08±1.02** | **77.84±2.11** | 82.88±2.24 |
| | (B1) | 88.43±0.68 | **97.51±0.14** | 96.80±1.36 | 83.24±2.98 | 84.48±3.17 | 86.80±1.02 | 75.98±2.52 | 79.46±1.22 | 84.98±2.01 | 74.40±3.12 | 77.64±0.83 | **84.48±1.32** |
| | (BA2) | 87.62±1.11 | 96.63±1.02 | 92.34±1.33 | **84.46±1.76** | 80.96±2.53 | 82.08±2.22 | 77.32±3.68 | 80.00±2.19 | 85.48±1.82 | 74.28±3.47 | 77.44±1.79 | 83.76±1.51 |
| | (R0) | **93.52±0.29** | 97.13±0.15 | **99.02±0.28** | 82.06±1.05 | 83.82±1.76 | 87.82±1.22 | **80.58±2.69** | 81.60±1.55 | **87.38±1.30** | 71.28±3.19 | 73.28±2.33 | 82.24±1.70 |

Table 9: Proportion of generated DP synthetic demonstrations for GINC, using three models, where the log-likelihood ratio exceeds the derived threshold, appearing in the RHS of Lemma 1.

| $\varepsilon$ | Methods | (MA1) | (MA2) | (MA3) |
|---|---|---|---|---|
| | (P1) | 16.89 | **48.89** | **49.33** |
| | (PA2) | 26.22 | 47.55 | 40.89 |
| 1 | (PA3) | **29.34** | 42.22 | 38.22 |
| | (B1) | 20.00 | 44.89 | 44.89 |
| | (BA2) | 28.89 | 42.22 | 42.22 |
| | (P1) | 20.89 | **50.22** | **49.33** |
| | (PA2) | 25.78 | 44.44 | 44.00 |
| 2 | (PA3) | **29.33** | 44.00 | 41.33 |
| | (B1) | 24.89 | 49.78 | 48.44 |
| | (BA2) | 28.89 | 43.56 | 46.22 |
| | (P1) | 24.45 | **58.67** | **56.89** |
| | (PA2) | 28.89 | 50.67 | 44.44 |
| 4 | (PA3) | **33.33** | 48.89 | 38.66 |
| | (B1) | 23.11 | 50.67 | 51.11 |
| | (BA2) | 28.89 | 46.22 | 42.22 |
| | (P1) | 20.89 | 53.34 | **53.33** |
| | (PA2) | **31.56** | 51.11 | 41.33 |
| 8 | (PA3) | 28.89 | 48.00 | 42.67 |
| | (B1) | 28.00 | **55.11** | 52.44 |
| | (BA2) | 30.67 | 48.45 | 44.45 |
| | (P1) | 23.55 | 52.45 | 49.33 |
| | (PA2) | 32.44 | 55.11 | 44.89 |
| $\infty$ | (PA3) | 32.44 | 52.00 | 40.89 |
| | (B1) | 33.78 | 55.56 | **52.00** |
| | (BA2) | **35.55** | **57.33** | 46.22 |

## F.6. Distinguishability Comparison

**Effects of varying model.** Similar to the accuracy across various models, (P1) clearly enlarges the divergence between $p(\tilde{O}|\theta)$ and $p(\tilde{O}|\theta^*)$ when using the models (MA2) and (MA3), which yield the higher non-private (R0) performance, while (PA2) and (PA3) shows better performance when using (MA1).

## F.7. Comparison with existing work on DP-ICL

We additionally conducted empirical comparison with DP-OPT (Hong et al., 2024) using the same Vicuna-7B-v1.5 model (Zheng et al., 2023) and TREC dataset setup as used in their study. The comparison results are summarized in Table 10. To ensure fair comparison, we report both our replicated results and the original results for DP-OPT from Table 3 of their paper, formatted as (replicated / original). This is because our replication was limited understanding of DP-OPT's hyperparameters. We followed their appendix settings for $\varepsilon = 8$ and extended them to $\varepsilon = 1$ using $\varepsilon_0$ from Table 5 of their paper. Additionally, differences in prompt format remain. Therefore, this comparison may not fully reflect the method's optimal performance.

To further support the numerical results, we highlight a key behavioral differences between DP-OPT and our approach at

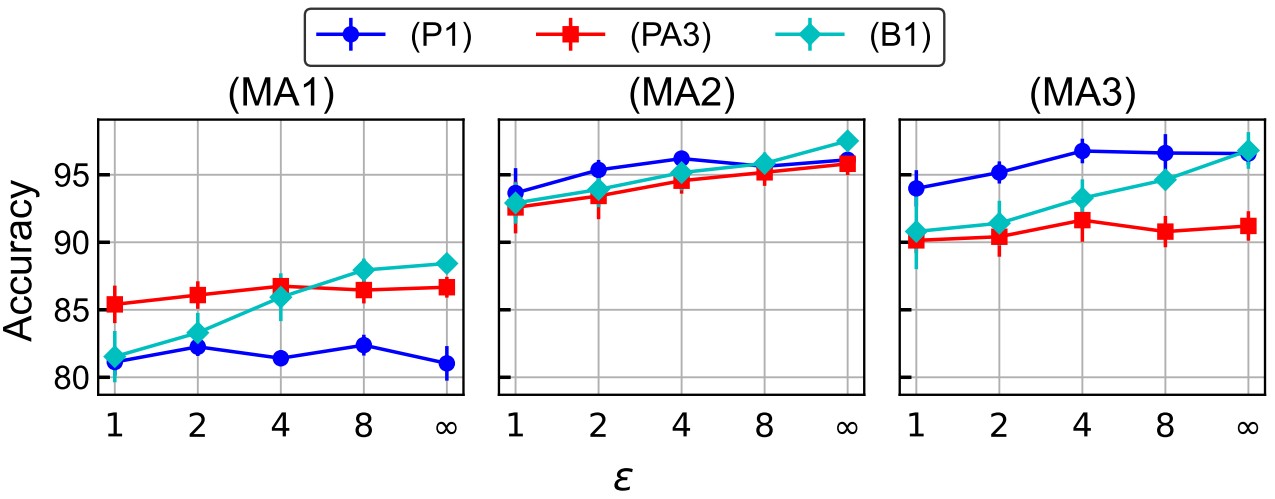

Figure 3: Effect of $\varepsilon$ on accuracy for DP-ICL methods and model variants

Table 10: Accuracy comparison with DP-OPT (Hong et al., 2024) on the TREC dataset using the Vicuna-7B-v1.5 model, averaged over five seeds. The highest accuracy is shown in bold, and the second-highest is underlined. For DP-OPT, we report original results from Table 3 of (Hong et al., 2024) and our replicated results, presented in the format (replicated / original).

| $\varepsilon$ | Methods | Vicuna-7B |
|---|---|---|
| 1 | (P1) | $\mathbf{77.84_{\pm 2.83}}$ |
| | (B1) | $\underline{75.60_{\pm 2.05}}$ |
| | DP-OPT (Hong et al., 2024) | $47.8_{\pm 0.0}$ / N.A. |
| 8 | (P1) | $\mathbf{76.28_{\pm 2.92}}$ |
| | (B1) | $\underline{73.84_{\pm 5.44}}$ |
| | DP-OPT (Hong et al., 2024) | $60.76_{\pm 1.27}$ / $65.3_{\pm 4.3}$ |

$\varepsilon = 1$. As discussed in Section 5.2 of their paper, DP-OPT tends to output only the instruction without demonstrations as the private prompt satisfying $(\varepsilon, \delta)$-DP guarantee with $\varepsilon = 1$. In such cases, the method method asymptotically converges to zero-shot prompting, limiting the practical utility of private demonstrations. In contrast, our method generates in-context demonstrations that, while potentially noisy, remain informative beyond the instruction even with $\varepsilon = 1$. This may allow our approach to consistently outperform zero-shot prompting, especially in high-privacy regimes where DP-OPT yields only marginal gains.

### F.8. Example of DP synthetic demonstrations

We provide examples of generated DP synthetic demonstrations in Table 11, highlighting how PTA influences token selection compared to other methods. Blue-highlighted tokens indicate cases where only PTA successfully selects the top-1 token after adding noise, while red-highlighted tokens represent cases where only PTA fails to do so.

Table 11: The DP synthetic demonstrations generated by PTA in the selected configurations.

| $\varepsilon$ Dataset | Label | Model | DP synthetic demonstrations |
|---|---|---|---|
| 1 AGNews | World | (MB1) | BREMER is back in court today to defend his role as the former governor of the bankrupt state of Alaska. The governor of the state of Alaska, Frank Murkowski, who appointed Bremer governor of the state after the bankruptcy, is expected to testify in support of Bremer, as is the current governor Tony Knowles. Bremer's former boss is also expected to defend. The trial was moved from federal court to a state court in |
| | | (MB2) | Syrian President Assad to Visit Iran in First Tory of 2009: Diplomat (AFP) Syrian President Bashar al-Assaad will visit Iran for the first time in 25 years later this month, in another sign of improving ties that have warmed since the two sides forged closer diplomatic links after the war in Lebanon. " 12.35 |
| | | (MB3) | Britain's Prince Charles, in his role as patron of the Prince's Youth Business Trust, has been in South Africa where he met young entrepreneurs. He met them at the University of Limpopo. He was impressed by the students' business ideas and their enthusiasm. The trust is working with the Department of Trade and Industry to promote entrepreneurship in South and southern Africa. It is also working with the African Development Bank to help young South Africans become self-employed |
| DBPedia | Album | (MB1) | The 64th Announcement is the second single released and fourth track from the album The Twilight Tapes, by the band The Twilight Sad. |
| | | (MB2) | The 64th Annual Grammy Awards were held on February 8, 2012, at the Staples Center in Los Angeles. The nomineesi were announced on October 3." |
| | | (MB3) | The album was recorded and mixed at the legendary Rockfield Studios in Wales, UK. The band recorded the album with producer Romesh Dodangoda (Motorhead, Bomb20, Funeral for a Friend, and Bullet for my Valentine). The album was mastered by Jason Mitchell (Young Guns, Bullet for my Valentine, Black Stone Cherry, and The Devil Wears Prada). The album was released on October 14th, 2 |
| TREC | Person | (MB1) | I have an interview on the phone tomorrow!" |
| | | (MB2) | I have an interview on Tuesday. I'm going to |
| | | (MB3) | Who was the first US president to fly in Air Force One ? Answer |

## F.9. Example of Prompts

In this section, we present the example of prompts during ICL and generating DP synthetic demonstrations in the text-classification datasets. For fair comparison, we used the same prompt format during ICL following (Tang et al., 2024), as shown in Table 12. In Table 13, we present the prompt construction functions used in Algorithm 3.

As for GINC, it's challenging to construct the instruction sentence due to its synthetic nature. When generating the DP synthetic demonstrations, we first insert a public demonstration and then append the private demonstrations. Each is formatted as a label-data pair.

Table 12: The prompts used during ICL for text-classification tasks, taken from Table 7 of Tang et al. (2024).

| Task | Prompt | Labels |
| --- | --- | --- |
| AGNews | Classify the news articles into the categories of World, Sports, Business, and Technology.

Article: USATODAY.com - Retail sales bounced back a bit in July, and new claims for jobless benefits fell last week, the government said Thursday, indicating the economy is improving from a midsummer slump.
Answer: Business

Article: New hard-drive based devices feature color screens, support for WMP 10.
Answer: | World, Sports, Business, Technology |
| DBPedia | Classify the documents based on whether they are about a Company, School, Artist, Athlete, Politician, Transportation, Building, Nature, Village, Animal, Plant, Album, Film, or Book.

Article: Geoffrey D. Falksen (born July 31 1982) is an American steampunk writer.
Answer: Artist

Article: The Perrin River is a 1.3-mile-long (2.1 km) tidal river in the U.S. state of Virginia. It is a small inlet on the north shore of the York River near that river's mouth at Chesapeake Bay.
Answer: | Company, School, Artist, Athlete, Politician, Transportation, Building, Nature, Village, Animal, Plant, Album, Film, Book |
| TREC | Classify the questions based on whether their answer type is a Number, Location, Person, Description, Entity, or Abbreviation.

Question: How did serfdom develop in and then leave Russia?
Answer Type: Description

Question: When was Ozzy Osbourne born?
Answer Type: | Number, Location, Person, Description, Entity, Abbreviation |

Table 13: Prompt construction function $PB(\cdot)$ used in Algorithm 3 for text-classification tasks, taken from Table 5 of Tang et al. (2024).

| Task | Prompt construction function $PB(\text{instruction}, \mathcal{D}, y)$ | Labels |
|---|---|---|
| AGNews | Given a label of news type, generate the chosen type of news accordingly.

News Type: World
Text: Australia boosts anti-terror measures at small airports SYDNEY: The Australian government announced a major security upgrade for nearly ...

News Type: World
Text: | World, Sports, Business, Technology |
| DBPedia | Given a label of document type, generate the chosen type of document accordingly.

Document Type: Company
Text: Cherry Lane Music was founded in 1960 by Milton Okun in the apartment above the Cherry Lane Theater in Greenwich Village of New York City...

Document Type: Company
Text: | Company, School, Artist, Athlete, Politician, Transportation, Building, Nature, Village, Animal, Plant, Album, Film, Book |
| TREC | Given a label of answer type, generate a question based on the given answer type accordingly.

Answer Type: Number
Text: How many people in the world speak French?

Answer Type: Number
Text: | Number, Location, Person, Description, Entity, Abbreviation |

