# OpenReview forum: "Plausible Token Amplification for Improving Accuracy of Differentially Private In-Context Learning Based on Implicit Bayesian Inference"
_ICML.cc/2025/Conference — ICML 2025 poster_

### Official Review · Reviewer_GXLu · 2025-03-13

**Overall Recommendation:** 3

**Summary:**

The paper introduces Plausible Token Amplification (PTA), a novel approach designed to improve the accuracy of Differentially Private In-Context Learning (DP-ICL) by refining the process of generating differentially private synthetic demonstrations. The paper also presents a theoretical guarantee of the distinguishibility of the latent concepts when noises are injected into the next-token-prediction distribution to ensure DP properties. The proposed PTA is aimed to make the latent concepts more distinguishable, thus improving the performance of generated DP demonstrations. Empirical effectiveness has been shown on several text classification datasets.

**Claims And Evidence:**

Most of the claims are well-supported. I have a few concerns about the experimental results, which aim to show the superiority of PTA:

1. DBPedia has significantly higher performance without the proposed KL divergence, which is an essential part of the proposed PTA. There is no explanation for this.

2. Only one DP-ICL baseline is included. Not sure if there could be more baselines to compare.

**Essential References Not Discussed:**

N/A

**Experimental Designs Or Analyses:**

I have a few concerns about the experimental results, which aim to show the superiority of PTA:

1. DBPedia has significantly higher performance without the proposed KL divergence, which is an essential part of the proposed PTA. There is no explanation for this.

2. Only one DP-ICL baseline is included. Not sure if there could be more baselines to compare.

**Methods And Evaluation Criteria:**

Yes. Though the datasets could be updated to more recent/relevant ones like MMLU.

**Other Comments Or Suggestions:**

N/A

**Other Strengths And Weaknesses:**

The paper is a nice combination of theoretical analysis and theory-inspired real-world algorithm.

**Questions For Authors:**

1. Why does DBPedia have significantly higher performance without the proposed KL divergence?

2. Are there any other DP-ICL baselines that you can compare PTA with?

3. Can you provide some analysis on Equation (12) and the effect of having this approximation?

**Relation To Broader Scientific Literature:**

The paper proposes a new DP in-context demonstration generation algorithm that performs better than the baseline.

**Theoretical Claims:**

The proofs are not fully/carefully checked.

The claim in Equation (12) is pretty hand-waving without any analysis of the closeness of the approximation. The whole proposed algorithm is built on this equation.

---

> ### Author Rebuttal · Authors · 2025-04-01
>
> # Interpretation of numerical experimental results}
>
> > Reviewer comments (Claims And Evidence, Questions For Authors):
> > 1. DBPedia has significantly higher performance without the proposed KL divergence, which is an essential part of the proposed PTA. There is no explanation for this.
> > 2. Why does DBPedia have significantly higher performance without the proposed KL divergence?
>
> Thank you for pointing this out. We agree that the performance degradation on DBPedia when using KL divergence requires clarification.
>
> As shown in Equation (11), PTA is fundamentally designed to amplify tokens distinctive to the ground-truth concept. The KL divergence term is introduced as an auxiliary regularizer to ensure that the modified token distribution does not deviate excessively from the original LLM next-token probability distribution. While not the core objective, this regularization can help stabilize training and preserve plausibility, contributing to improved performance on datasets such as AGNews and GINC.
>
> However, as with many regularization techniques, the KL term does not consistently improve accuracy. In some cases, it may act as a constraint that limits useful adaptation to the target distribution. We suspect this effect may be present in DBPedia, though further analysis would be needed to pinpoint the exact cause.
>
> We will add this clarification in the revised version.
>
> # Lacking baseline methods
>
> > Reviewer comments (Claims And Evidence, Questions For Authors):
> > 1. [...] Not sure if there could be more baselines to compare.
> > 2. [...] Are there any other DP-ICL baselines that you can compare PTA with?
>
> We agree with these comments. A similar suggestion was also raised by another reviewer (DiNj). In response, we conducted additional experiments during the rebuttal period using DP-OPT (Hong et al., ICLR 2024), which is conceptually similar to our approach in that it generates differentially private prompts once and reuses them for evaluating test queries.
>
> For a detailed discussion, please see our response to  Reviewer DiNj under “Missing comparison with related works.”
> We will include the new experimental results and their interpretation in the revised version of the paper.
>
> (Hong et al., ICLR 2024) DP-OPT: Make large language model your privacy-preserving prompt engineer.
>
> # Effect of approximation in Equation. (12)
> > Reviewer comments (Claims And Evidence, Theoretical evidence):
> > 1. The claim in Equation (12) is pretty hand-waving without any analysis of the closeness of the approximation. [...]
> > 2. Can you provide some analysis on Equation (12) and the effect of having this approximation?
>
> This is a good question.
> As you noted, PTA is based on the following approximation in Equation~(12):
> $$
> \log \frac{p(o = v \mid \theta^*)}{p(o = v \mid \theta)} \approx \log \frac{p(o = v \mid S_{\text{priv}}^{(i)})}{p(o = v \mid S_{\text{pub}})}. \hspace{10pt} (12)
> $$
>
> We approximate the LHS of Equation (12) since it is intractable to compute directly.
> This intractability arises from the fact that $\theta$ and $\theta^*$ are not directly observable as they are latent variables, making it impossible to evaluate the likelihoods $ p(o = v \mid \theta^*) $ and $ p(o = v \mid \theta) $ explicitly.
> To address this, we approximate the LHS using the LLM’s next-token probability distribution conditioned by observable prompts, which are designed to reflect their corresponding latent concepts.
>
> To explain how we approximate, we decompose the LHS of Equation (12) as follows:
> $$
>   \log \frac{p(o = v \mid \theta^*)}{p(o = v \mid \theta)} = \log \frac{p(o = v \mid \theta^*)}{p(o = v \mid S_{\text{pub}})} + \log \frac{p(o = v \mid S_{\text{pub}})}{p(o = v \mid \theta)}.
> $$
>
> First, the first term on the RHS is then approximated by $ \log \frac{p(o = v \mid S_{\text{priv}}^{(i)})}{p(o = v \mid S_{\text{pub}})} $, replacing the intractable $ p(o = v \mid \theta^*) $ with a quantity that can be computed using the LLM prompted by the private prompt $S_{\text{priv}}^{(i)}$.
> As detailed in Section 4.1, this is justified by the Bayesian interpretation of in-context learning from Xie et al., where such prompting induces a posterior concentrated around $\theta^*$.
>
> Second, we omit the second term in RHS to focus on amplifying tokens distinctive to $\theta^*$, without explicitly penalizing competing concepts. This omission is plausible when the second term is small. This occurs when different concepts assign high probability to distinct sets of tokens. In such cases, tokens indicative of $\theta^*$ are also unlikely under both $\theta$ and the reference $p(o = v \mid S_{\text{pub}})$, making the omitted term negligible. To support this in practice, we construct $S_{\text{pub}}$ using instruction-only prompts that avoid concept-specific content. This yields a neutral reference distribution and helps maintain separation across concept token distributions.
>
> We will revise the paper to clarify this approximation strategy.

---

### Official Review · Reviewer_DiNj · 2025-03-14

**Overall Recommendation:** 4

**Summary:**

This manuscript explores Differentially Private In-Context Learning (DP-ICL) to mitigate leakage risks in ICL. The authors first provide a theoretical analysis for a prior work (Tang et al., 2024) using a Bayesian analysis, where Tang et al. studied generating synthetic demonstrations by adding variance-tuned noise to the next-token probability obtained from an LLM prompted with the original demonstrations. Based on the authors' theory, they identify how the added noise to ensure DP affects the LLM’s ability to infer the ground-truth concept. Accoradingly, two insights for improving DP-ICL are derived: (i) Reducing the vocabulary size lowers the noise-dependent threshold and (ii) Increasing the divergence between concepts, by employing another next-token probability distribution that enlarges the gap between the ground-truth and any other concept. The authors also propose Plausible Token Amplification (PTA) to improve the performance of DP-ICL and empirically verify its effectiveness.

**Claims And Evidence:**

The claims made in this submission are supported by convincing evidence.

**Essential References Not Discussed:**

N/A

**Experimental Designs Or Analyses:**

The experimental designs and analyses are soundness.

**Methods And Evaluation Criteria:**

The proposed methods indeed make sense for the problem.

**Other Comments Or Suggestions:**

I have no more comments or suggestions.

**Other Strengths And Weaknesses:**

**Strengths**
1. This manuscript is overall well-written with clearly organized.
2. The authors provide theoretical evidence (based on Implicit Bayesian Inference) supporting Tang et al.’s empirical method in Differentially Private In-Context Learning.
3. This paper also introduce a refined method PTA for modifying next-token probability distribution accorading their theoretical insights.
4. Extensive experiments empirically demonstrate the effectiveness of PTA.

**Weaknesses**
1. The term "concept" in the introduction needs clearer demonstration. It can improve the readability of the introduction.
2. Missing emprical comparison with:\
      Wu et al. Privacypreserving in-context learning for large language models. ICLR 2024\
      Hong et al. DP-OPT: Make large language model your privacy-preserving prompt engineer. ICLR 2024
2. The authors verify their method on relatively small LLMs such as GPT-2, Llama2-7B. How will the PTA perform on modern LLMs such as GPT4?

**Questions For Authors:**

Please see the weaknesses.

**Relation To Broader Scientific Literature:**

N/A

**Theoretical Claims:**

The proofs for theoretical claims seems correct.

---

> ### Author Rebuttal · Authors · 2025-04-01
>
> # Unclear definition of "concept"
> >  Reviewer comment (Other Strengths And Weaknesses, Weakness):
> > 1. The term "concept" in the introduction needs clearer demonstration. It can improve the readability of the introduction.
>
> Thank you for pointing this out we had not noticed this readability issue. In our context, a concept refers to the latent rule or semantic mapping that connects inputs to outputs in demonstrations. This underlying concept governs the token transitions in the demonstrations and is implicitly inferred by the language model during In-Context Learning (ICL).
>
> In the revised version, we carefully clarify and introduce the term “concept” in Section. 1.
>
> # Missing comparison with related works
> > Reviewer comment (Other Strengths And Weaknesses, Weakness):
> > 1. Missing empirical comparison with: (1) Wu et al. (ICLR 2024) and (2) Hong et al. (ICLR 2024).
>
> Thank you for pointing out these relevant works.
>
> (1) Wu et al. (ICLR 2024)
>
> While their method is indeed related, it directly adds noise to the next-token probability distribution at test time for each classification query.
> As a result, the noise variance must scale with the number of test queries to ensure a fixed $(\varepsilon,\delta)$-DP.
> In contrast, the noise variance in our method (and Tang et al.'s) remains fixed—regardless of the number of queries thanks to the post-processing property of DP: once a prompt satisfying $(\varepsilon,\delta)$-DP is generated.
> This fundamental difference in design makes direct comparison challenging, thus we did not include their method in our evaluation.
>
> (2) Hong et al. (ICLR 2024)
>
> We agree that DP-OPT, proposed by Hong et al., is conceptually similar to our approach, as it generates differentially private prompts once and reuses them to evaluate test queries. During the rebuttal period, we conducted an additional empirical comparison using the same Vicuna-7B-v1.5 model and TREC dataset setup as used in their study. The comparison results are summarized in the table below.
>
> To ensure fair comparison, we report both our replicated results and the original values for DP-OPT from Table 3 of their paper, formatted as (replicated / original). This is because our replication was limited by computational constraints and a limited understanding of DP-OPT’s hyperparameters. We followed their appendix settings for $\varepsilon=8$ and extended them to $\varepsilon=1$ using $\epsilon_0 = 0.1$ from their Table 5. Additionally, differences in prompt format remain. Therefore, this comparison may not fully reflect the method’s optimal performance.
>
> |$\varepsilon$|Method|Accuracy|
> |-|-|-|
> |1|PTA (Ours)|$77.84_{\pm2.83}$|
> ||Tang et al.|$75.60_{\pm2.05}$|
> ||DP-OPT|$47.8_{\pm0.0}$ / N.A.|
> |8| PTA (Ours)|$76.28_{\pm2.92}$|
> ||Tang et al.|$73.84_{\pm5.44}$|
> ||DP-OPT|$60.76_{\pm1.27}$ / $65.3_{\pm4.3}$|
>
> We present below our interpretation of the experimental results:
>
> - Even under the same evaluation setup, both our proposed PTA and Tang et al.'s DP-ICL acheieve high accuracy. This confirms the competetive performance of PTA compared to a strong DP-ICL baseline.
>
> To further support the numerical results, we highlight a key behavioral differences between DP-OPT and our approach at $\varepsilon=1$.
>
> - As discussed in Section 5.2 of their paper, DP-OPT tends to output only the instruction without demonstrations as the private prompt satisfying $(\varepsilon,\delta)$-DP guarantee with $\varepsilon=1$. In such cases, the method method asymptotically converges to zero-shot prompting, limiting the practical utility of private demonstrations.
>
> - In contrast, our method generates in-context demonstrations that, while potentially noisy, remain informative beyond the instruction even with $\varepsilon=1$. This may allow our approach to consistently outperform zero-shot prompting, especially in high-privacy regimes where DP-OPT yields only marginal gains.
>
> We will include these numerical results and their interpretation in the revised paper.
>
> # Performance investigation using GPT4-o
> > Reviewer comment (Other Strengths And Weaknesses, Weakness):
> > 1. [...] How will the PTA perform on modern LLMs such as GPT4?
>
> We believe that applying our method (as well as that of Tang et al.) to models such as GPT-4 accessed via the OpenAI API poses challenges from a privacy evaluation perspective. At this moment, the API does not provide access to the full token probability distribution, instead returning only a limited set of top tokens. Since the selection of these tokens depends on the full (unobservable) distribution, it may lead to unintended privacy leakage.
> Moreover, as this selection process is a black box, it is practically infeasible to assess its differential privacy guarantees. Given the current API constraints, we consider rigorous privacy evaluation using GPT-4 constraints to be challenging at this time.

---

> > ### Comment · Reviewer_DiNj · 2025-04-09
> >
> > Thanks for the authors' rebuttal. My concerns have been addressed, and I will maintain my rating.

---

### Official Review · Reviewer_kLHj · 2025-03-23

**Overall Recommendation:** 4

**Summary:**

The paper presents an approximate Bayesian model of DP-ICL (differentially private in-context learning). DP-ICL in general combines a public LLM with a private LLM to generate a number of low-privacy-leakage prompts, then uses those prompts for in-context learning. A previous paper (Tang et al.) used a simple vocabulary restriction method to improve DP-ICL; this paper uses their Bayesian model to explain why vocabulary restriction works, then present an improved Plausible Token Amplification method which achieves better empirical results.

## Update after rebuttal

I am happy with the authors' rebuttal, and continue to think the paper should be accepted. The authors have said they'll clarify the theorem statements to show that C_delim and G depend on some things, and after that my only reservation is that I expect there is a future cleaner paper that will result in cleaner and better algorithms as well. However, the correct course of action is to accept this paper in hopes it in fact encourages that future cleaner paper.

**Claims And Evidence:**

(Caveat: Unfortunately I caught a cold over the weekend, and as a result have not had time to check the proofs. Therefore, this review is based on the main body of the paper and a quick skim of the proofs.)

# Theoretical evidence

1. The theoretical model seems plausibly correct given that assumptions: it is a reasonable generalisation of an existing Bayesian approximation to non-DP ICL. However, I have to admit I don't find it very elegant! Appendix C.2 has to introduce a whole pile of constants to approximate things about the Hidden Markov Model, so the C_delim "constant" in the main body theorem statement hides a bunch of things which are very far from constant, even in the approximate model. Similarly, G is also dependent on the LLMs involved.
2. Actually the non-constant natural of C_delim and G really do need to be called out in the main text. When I was reading the main text I thought they really were constants, which is not correct, as is the theorem statements mislead the reader.
3. My guess is that there are cleaner theorems hiding under the surface the theorems they stated, that might lead to improved methods. In particular, Theorem 2 in particular has a term involving the vocabulary size, but my guess is that this is artificial in order to present direct evidence for the success of Tang's method. If the prompts are sampled sufficiently closely to the public distribution on a reasonable metric, we should be able to get a bound that doesn't depend on the vocabulary size. The vocabulary size would play a role, but would not as directly show up in the theorem statement.
4. This vocabulary size appearance then contaminates the empirical method, which combines *both* vocabulary restriction and PTA. My guess is that there is a simpler, better method available if one more naturally combines the two features.

# Empirical evidence

1. The empirical evaluation is good, and do support the improvement of PTA over the baseline.
2. However, I would have preferred to see a graph showing the degradation in accuracy as the privacy budget is tightened: currently only two points along that curve are shown.

# Summary

Overall, I am happy with the paper! The fact that I think there is a better paper lurking under this one (with cleaner theory and empirics) still means this paper has good contributions, and I believe it should be accepted.

**Essential References Not Discussed:**

Due to catching a cold immediately prior to the deadline, I don't have capacity to do a literature search, and unfortunately I don't have significant existing knowledge of the DP literature. Thus, while the paper's internal discussion of the literature appears good, I am unable to check whether important things are missed.

**Experimental Designs Or Analyses:**

The experimental designs are simple. The main issue with any DP method is that bugs could easily break the guarantees and result in fake, better results, but this is hard to rule out with certainty.

**Methods And Evaluation Criteria:**

Yes, I am happy with the datasets chosen in section 5.

**Other Comments Or Suggestions:**

N/A

**Other Strengths And Weaknesses:**

Discussed above, but to summarise: both the theorems presented and the final method, while useful contributions, appear non-natural. I expect there are better versions of both theorems and methods, in particular by merging vocabulary restriction natively into some metric comparing public and private distributions.

**Questions For Authors:**

N/A

**Relation To Broader Scientific Literature:**

Due to catching a cold immediately prior to the deadline, I don't have capacity to do a literature search, and unfortunately I don't have significant existing knowledge of the DP literature. Thus, while the paper's internal discussion of the literature appears good, I am unable to check whether important things are missed.

**Theoretical Claims:**

See the Claims And Evidence section above.

---

> ### Author Rebuttal · Authors · 2025-04-01
>
> # Concerns in bounding constants
> > Reviewer comments (Claims And Evidence, Theoretical evidence)
> > 1. [...] introduce a whole pile of constants [...], so the C\_delim "constant" [...] hides a bunch of things which are very far from constant, [...]. Similarly, G is also dependent on the LLMs involved.
> > 2. [...] non-constant natural of C\_delim and G really do need to be called out in the main text. [...]
>
> This is a good question. We address $G(\sigma)$ and $C_{delim}$, separately.
>
> First, $G(\sigma)$ is a function, not a constant, as defined in Line 1339 (Appendix C.7), though this was not clearly stated in the main text. We did not intend to hide this point, thus we will include its definition and clarify its dependency on the LLM's next-token probability distribution in the revised paper. Notably, introducing the bound $G(\sigma)$ in Line 1339 is essential for a **tight** analysis of the noise error in Theorem 2. Under general settings without assuming any specific form of the LLM's next-token probability distribution, a tight analysis using $G(\sigma)$ in Line 1339 is established thanks to Bretagnolle–Huber bound (Lemma. 4, Appendix C.3).
>
> Next, we clarify that the term $C_{delim}$, which is defined in Line 1009, is also a function, not a constant.
> It originated from the formulation in Xie et al. (ICLR 2022) and depends on the ground-truth concept $\theta^*$ and the other concept $\theta$. We will also clarify this dependency in the revised paper.
>
> # Refinement of Theorem
> > Reviewer comments  (Claims And Evidence, Theoretical evidence):
> > 1. [...] cleaner theorems hiding under the surface the theorems [...]. Theorem 2 has a term involving the vocabulary size, [...] this is artificial in order to present direct evidence for the success of Tang's method.
> > 2. [...] If the prompts are sampled sufficiently closely to the public distribution [...], we should be able to get a bound that doesn't depend on the vocabulary size. [...]
>
> Thank you for the insightful comment. We agree more elegant theorems may arise under additional assumptions—e.g., when prompts are sampled closely from a public distribution and this is a promising direction for future work.
>
> However, Theorem 2 is designed to hold under general settings without assuming a specific form of next-token probability distribution. In practice, prompts often contain private demonstrations that may diverge from public distributions, making such assumptions unreliable.
>
> To address your concern, we clarify why the $\log |V|$ term appears in Theorem 2. Our analysis (Section 3) examines how the noise impacts latent concept inference in DP-ICL. Since next-token probability distributions of LLMs can vary widely, especially with private prompts, we aim to derive a general bound that holds without assuming any specific form of the distribution. Applying Csiszár’s inequality (Lemma 3, Appendix C.3) naturally introduces the $\log |V|$ term as a bound on the maximum distributional shift. Further, as discussed in the former section (Concerns ...), $G(\sigma)$ also results from a tight analysis. Together, $G(\sigma)\log |V|$ forms a general upper bound on noise impact.
>
> We will clarify this motivation in the revised version.
>
> # Refinement of empirical method}
> > Reviewer comments (Claims And Evidence, Empirical evidence, Other Strengths And Weaknesses):
> > 1. [...] there is a simpler, better method available if one more naturally combines the two features, vocabulary restriction and PTA.
> > 2. [...] by merging vocabulary restriction natively into some metric comparing public and private distributions.
>
> This is an insightful point. We acknowledge that metrics comparing public and private distributions can directly inform vocabulary restriction.
>
> Our goal, however, was to theoretically support Tang et al.’s empirical vocabulary restriction and refine it through PTA. Building on Theorem 2, we adopted a modular design to isolate each component’s role: vocabulary restriction narrows the output space, to reduce the noise impact, while PTA amplify token probabilities to increase divergence between the ground-truth and other concepts.
>
> We appreciate the suggestion and see it as a promising future direction.
>
> # Empirical evaluation
> > Reviewer comment (Claims And Evidence, Empirical evidence):
> > 1. [...] to see a graph showing the degradation in accuracy as the privacy budget is tightened [...]
>
> Due to space constraints, we showed results for $\varepsilon=1$ and $8$ in the main paper. We agree broader coverage is important.
>
> Table 8 (Appendix F.4) includes results for $\varepsilon=1,2,4,8,\infty$ (following Tang et al., ICLR 2024), giving a clearer tren of how accuracy changes as privacy strengthens. Below is a summary for PTA on GINC:
>
> |$\varepsilon=1$|$\varepsilon=2$|$\varepsilon=4$|$\varepsilon=8$|
> |-|-|-|-|
> |$94.55_{\pm 1.20}$|$95.17_{\pm 0.82}$|$ 96.76_{\pm 0.91}$|$96.85_{\pm 0.97}$|
>
> We will include accuracy curves in the revision to better illustrate this trend.

---

### Decision · Program_Chairs · 2025-05-01

**Decision:**

Accept (poster)

**Comment:**

This manuscript received favourable reviews from 3 reviewers. The reviews and discussions highlighted both strengths and weaknesses but the general tendency is in favour.

On the side of strengths, it was commented by a reviewer who is expert on the topic that the paper presents an approximate Bayesian model of DP-ICL (differentially private in-context learning). DP-ICL in general combines a public LLM with a private LLM to generate a number of low-privacy-leakage prompts, then uses those prompts for in-context learning. A previous paper (Tang et al.) used a simple vocabulary restriction method to improve DP-ICL; this paper uses their Bayesian model to explain why vocabulary restriction works, then present an improved Plausible Token Amplification method which achieves better empirical results. The other reviewers commented also that this manuscript is overall well-written with clearly organized, provides theoretical evidence (based on Implicit Bayesian Inference) supporting Tang et al.’s empirical method in Differentially Private In-Context Learning, introduces a refined method (PTA) for modifying next-token probability distribution accorading their theoretical insights, extensive experiments empirically demonstrate the effectiveness of PTA, and that the paper is a nice combination of theoretical analysis and theory-inspired real-world algorithm.

On the side of weaknesses, it was commented that it would be expected that there are better versions of both theorems and methods, in particular by merging vocabulary restriction natively into some metric comparing public and private distributions, that there were missing emprical comparison with some works, and the method could be tested on relatively small LLMs such as GPT-2, Llama2-7B.

The rebuttals alleviated some concerns and the reviewers were satisfied, reiterating their positive evaluation of this paper.